# Characteristics of Surface Physical and Biogeochemical Parameters within Mesoscale Eddies in the Southern Ocean

Qian Liu [1, 2], Yingjie Liu [1, 2], Xiaofeng Li [1, 2]

[1]CAS Key Laboratory of Ocean Circulation and Waves, Institute of Oceanology, Chinese Academy of Sciences, Qingdao 266071, China.
[2]University of Chinese Academy of Sciences, Beijing 100049, China.

*Correspondence to*: Yingjie Liu (yjliu@qdio.ac.cn)

**Abstract.** Using satellite sea surface temperature (SST) and chlorophyll-*a* (Chl-*a*) as well as observation-based reconstruction of dissolved inorganic carbon (DIC) and partial pressure of $CO_2$ ($pCO_2$) from 1996 to 2015, we investigate the modulation mechanisms of eddies on surface physical and biogeochemical parameters in the Southern Ocean (SO). About 1/4 of eddies are observed to be "abnormal" (cold anticyclonic and warm cyclonic eddies) in the SO, which show opposite SST signatures contrary to "normal" eddies (warm anticyclonic and cold cyclonic eddies). The study finds that the modification of "abnormal" eddies on physical and biogeochemical parameters is significant and differs from "normal" eddies due to the combined effects of eddy pumping and eddy-induced Ekman pumping. "Normal" and "abnormal" eddies have opposite DIC anomalies, contrary to the SST anomalies. Moreover, the contributions of "abnormal" eddies to $pCO_2$ are about 2.7 times higher than "normal" eddies in regions where "abnormal" eddies dominate. Although Chl-*a* anomalies in "normal" and "abnormal" eddies show similar patterns and signals, eddy-induced Ekman pumping attenuates the magnitudes of Chl-*a* anomalies within "abnormal" eddies. In addition to the variation of the same parameter within different eddies, the dominant eddy-driven mechanisms for different parameters within the same kind of eddies also vary. The strength of the eddy stirring effect on different parameters is the primary factor causing these differences, attributed to variations in the magnitudes of horizontal parameter gradients. Understanding the role of "abnormal" eddies and the complexity of eddy-driven processes is crucial for accurately estimating the influence of mesoscale eddies on physical and biogeochemical processes in the SO, which is essential for simulating and predicting biogeochemical dynamics and carbon cycling in the region.

## 1 Introduction

Mesoscale eddies are swirling water existing ubiquitously in the global ocean and can influence biogeochemical cycling through horizontal and vertical transport of water masses with physical and biogeochemical parameters (Altabet et al., 2012; Stramma et al., 2013; Dong et al., 2014; Song et al., 2016; Dawson et al., 2018). Eddy activity is particularly high in the Southern Ocean (SO), a critical area for global ocean dynamics (Marshall and Speer, 2012). The absorption of anthropogenic $CO_2$ by the SO accounts for approximately 40 % of the global ocean, and the strength of this carbon sink is variable and

sensitive to changes in climate (Le Quéré et al., 2007; Landschützer et al., 2015), highlighting the tremendous importance of SO in the global climate. Therefore, it is significant to comprehensively investigate the role of eddies in regulating physical and biogeochemical parameters in the SO.

Sea surface temperature (SST), chlorophyll-a (Chl-a), dissolved inorganic carbon (DIC), and partial pressure of $CO_2$
($pCO_2$) are crucial physical and biochemical parameters that are extensively utilized to investigate the impact of mesoscale eddies on the marine environment and carbon cycle (Mcgillicuddy and Robinson, 1997; Gaube et al., 2013; Frenger et al., 2015; Jones et al., 2017). Previous literature found eddies can deform the horizontal parameter gradient via eddy rotation (eddy stirring), trap and transport water masses (eddy trapping), and enhance or suppress local surface SST, Chl-a, DIC, and $pCO_2$ through the vertical velocity in eddy cores, such as eddy pumping, seasonal modulation of the mixed layer, and eddy-
induced Ekman pumping (Mcgillicuddy et al., 2007; Dufois et al., 2014; Gaube et al., 2014; Song et al., 2016; Dawson et al., 2018; Frenger et al., 2018). Eddy pumping refers to the vertical displacement of isopycnals during the formation, growth, and destruction of mesoscale eddies (Nencioli et al., 2010; Lasternas et al., 2013; Huang et al., 2017; Dawson et al., 2018; Xu et al., 2019). Typically, anticyclonic eddies (AEs) cause a deepening of isopycnals and downwelling with warm, unproductive, and low-DIC surface water. On the contrary, cyclonic eddies (CEs) lead to a doming of isopycnals and
upwelling with cold, productive, and DIC-rich deep water into the euphotic zone. The variation of $pCO_2$ was found to be positively correlated with SST and DIC but negatively correlated with Chl-a (Chen et al., 2007; Landschützer et al., 2015; Song et al., 2016; Fay et al., 2018; Jersild and Ito, 2020; Iida et al., 2021). The competing seasonal cycles of SST, Chl-a, and DIC would induce the seasonal variability of $pCO_2$ and the seasonal variation of $pCO_2$ within the eddies varies in different regions (Chen et al., 2007; Frenger et al., 2013; Jiang et al., 2014; Munro et al., 2015; Song et al., 2016; Jones et al., 2017;
Jersild and Ito, 2020). Therefore, the variation of $pCO_2$ within the eddies will be complex and necessitates discussion based on seasons and regions.

However, previous studies have reported that seasonal modulation of the mixed layer and eddy-induced Ekman pumping can cause eddy-induced anomalies contrary to those predicted due to unusual vertical transports in eddy cores (Mcgillicuddy et al., 2007; Gaube et al., 2013; Dufois et al., 2014; Gaube et al., 2014; Gille et al., 2014; Dawson et al., 2018). For example,
Dufois et al. (2014) suggested that in the South Indian Ocean between 20°S and 30°S, deeper mixing in winter AEs can elevate nutrient supply, while shallower mixing in CEs can reduce it, which could explain stronger positive Chl-a anomalies in AEs than in CEs. The influence of mixing on Chl-a anomalies within eddies is similar to the eddy-induced Ekman pumping, which is generated by the sea surface stress curl caused by surface differential currents associated with mesoscale eddies and surface wind fields (Mcgillicuddy et al., 2007; Gaube et al., 2015). This surface stress curl has an opposite
polarity to the vorticity of the eddy, causing Ekman upwelling in the cores of AEs and Ekman downwelling in the cores of CEs. Unlike the seasonal modulation of the mixed layer, eddy-induced Ekman pumping persists throughout the lifetime of the eddy, and its magnitude depends on eddy amplitude and ambient wind speed (Gaube et al., 2014).

Previous research has primarily utilized rotation direction and sea surface height anomaly (SSHA) to distinguish AEs and CEs and analyze their impacts on physical and biochemical parameters (Frenger et al., 2015; Song et al., 2016; Dawson et al.,

2018). However, recent studies found that AEs can be further divided into warm and cold anticyclonic eddies (WAEs and CAEs). Similarly, CEs can be divided into cold and warm cyclonic eddies (CCEs and WCEs) depending on SST (Leyba et al., 2017; Liu et al., 2020; Liu et al., 2021; Ni et al., 2021). WAEs and CCEs are considered "normal" eddies that align with conventional knowledge, and CAEs and WCEs are considered "abnormal" eddies. "Abnormal" eddies are ubiquitous in the ocean, constituting approximately 32 % of the total eddies in the global ocean, and "abnormal" eddies in the Antarctic

Circumpolar Current (ACC) account for 19.9 % of global "abnormal" eddies (Liu et al., 2021). Previous literature proposed that "abnormal" eddies may be induced by eddy-induced Ekman pumping (Gaube et al., 2013; Mcgillicuddy, 2015), instability during the eddy decay stage, eddy horizontal entrainment (Sun et al., 2019), and warm/cold background water (Leyba et al., 2017). The roles of "abnormal" eddies in ocean circulation (Shimizu et al., 2001), mass transportation (Pickart et al., 2005; Mathis et al., 2007; Everett et al., 2012), and air-sea interaction (Leyba et al., 2017; Liu et al., 2020) differ from

those of "normal" ones (Assassi et al., 2016; Dilmahamod et al., 2018). However, the specific impact of "abnormal" eddies on physical and biogeochemical parameters in the SO remains unclear. Moreover, previous studies have primarily focused on the basin-wide effects of eddies on Chl-$a$, while investigations into the basin-scale effects of SO eddies on DIC and $p$CO$_2$ are lacking. Given the potential interactions between different physical and biogeochemical parameters and the importance of the SO in global climate change, biological productivity, and carbon cycling, it is necessary to systematically study the

influence of eddies on SST, Chl-$a$, DIC, and $p$CO$_2$ in the SO.

We aim to extend SO eddy-induced anomalies studies and examine the influence of "abnormal" eddies on surface physical and biogeochemical parameters. Unlike traditional eddy detection methods based on satellite sea surface height (SSH) data (Chelton et al., 2011a; Faghmous et al., 2015), the eddy dataset we used is developed by a deep learning (DL) model based on the fusion of SSH and SST data (Liu et al., 2021), which can simultaneously detect eddy locations and

distinguish between "normal" and "abnormal" eddies with great accuracy and efficiency. Instead of using potential density and geostrophic current direction to identify "abnormal" eddies (Mcgillicuddy, 2015), we choose to use the SST signature for distinguishing between "normal" and "abnormal" eddies. Because compared to potential density, SST data can be obtained from satellite remote sensing with higher spatial and temporal resolutions, making it a convenient and reliable data source for identifying eddies (Castellani, 2006; Liu et al., 2021). Using satellite SST and Chl-$a$, observation-based

reconstruction of DIC and $p$CO$_2$, and eddy datasets from 1996 to 2015, we systematically analyze their seasonal and regional variations induced by "normal" and "abnormal" eddies and investigate the mechanisms driving these responses. The study is organized as follows (Fig. S1). First, Sects. 2 and 3 provide details about data and methods. Then, in Sect. 4, we present the spatial distributions of eddy parameters, as well as spatial distributions and composite maps of eddy-induced SST, Chl-$a$, DIC, and $p$CO$_2$ anomalies. Section 5 investigates the mechanisms driving the surface parameter responses to eddies. Section

6 discusses the cause of distinct dominant eddy-driven mechanisms for different parameters within the same kind of eddies and provides conclusions.

## 2 Data

### 2.1 Study region

The SO is the region between 30°S and 65°S (Fig. 1). The ACC in the SO is a global circulation that links the Pacific,
Atlantic, and Indian Oceans from west to east (Marshall and Speer, 2012). We use the positions of the northern Subantarctic
Front (SAF) and the Polar Front (PF) (Sallée et al., 2008) available from the Center for Topographic Studies of the Ocean
and Hydrosphere (CTOH; http://ctoh.legos.obs-mip.fr/applications/mesoscale/southern-ocean-fronts). We average the data
of the fronts over the eddy period (1996 to 2015) as boundaries for ACC major Fronts (black lines in Fig. 1).

### 2.2 SST, Chl-*a*, DIC, and *p*CO$_2$ datasets

Four datasets of sea surface parameters are used in the study, including SST, Chl-*a*, DIC, and *p*CO$_2$ from 1996 to 2015,
between 30°S and 65°S. Table S1 presents information about spatial and temporal resolutions and filtering methods
employed for these parameters. A brief description of each data is given below.

The daily SST dataset is the NOAA Optimum Interpolation (OI) SST product with 0.25° resolution, spanning from 1981
to the present (Reynolds et al., 2007). The SST dataset combines observations from different platforms on a regular global
grid, including Advanced Very High-Resolution Radiometer (AVHRR) satellite data, ships, buoys, and Argo floats with an
accuracy of about 0.1 °C daily.

The Chl-*a* dataset is provided by Copernicus Marine Environmental Monitoring Service (CMEMS), based on the
Copernicus-GlobColour processor that merges three algorithms (Gohin et al., 2002; Hu et al., 2012; Garnesson et al., 2019).
The Chl-*a* dataset combines observations from different sensors (SeaWiFS, MODIS Aqua, MODIS Terra, MERIS, VIIRS
NPP, VIIRS-JPSS1 OLCI-S3A, and S3B). The original 4 km resolution data was re-gridded to 0.25° with daily temporal
resolution. We log-transform Chl-*a* using the base 10 logarithm because Chl-*a* is lognormally distributed (Campbell, 1995).

The *p*CO$_2$ and DIC datasets are from the Japan Meteorological Agency (JMA) Ocean CO$_2$ Map dataset with monthly 1° ×
1° gridded values on the global ocean from 1990 to 2020 (Iida et al., 2021). The DIC concentration is calculated from total
alkalinity (TA) values and CO$_2$ fugacity (*f*CO$_2$) data provided by the Surface Ocean CO$_2$ Atlas (SOCAT), containing data
from the 1950s to the present (Bakker et al., 2016). The DIC field is gap-filled by using a multi-linear regression (MLR)
method based on the DIC and satellite observation data, including SST, sea surface salinity (SSS), sea surface dynamic
height (SSDH), Chl-*a*, and surface mixed layer depth (MLD) (Iida et al., 2021).

$$nDIC = f \text{ (time, SST, SSS, SSDH, Chl-}a\text{, MLD)}, \qquad (1)$$

The globally averaged error in DIC is 6.1 μmol kg$^{-1}$, which is 5.4 μmol kg$^{-1}$ smaller than the error of Global Ocean Data
Analysis Project version 2 update 2019 (GLODAPv2.2019), a uniformly calibrated open ocean data product on inorganic
carbon and carbon-relevant variables (Olsen et al., 2019).

The $p$CO$_2$ field is calculated from TA, DIC, SST, and SSS based on seawater CO$_2$ chemistry (Iida et al., 2021). Firstly, the mean rates of regional $p$CO$_2$ and multiple regressions are used to derive the algorithms of $p$CO$_2$ expressed empirically as a function of in situ TA, DIC, SST, SSS, and the year. Then, the $p$CO$_2$ fields that filled both in space (1° × 1°) and time (monthly) are drawn by applying global data sets of TA, DIC, SST, and SSS to the variables in these empirical equations. The error in $p$CO$_2$ was 10.9 µatm, comparable with those estimated with other empirical methods, e.g., 14.4 µatm (Landschützer et al., 2014) and 15.73 µatm (Denvil-Sommer et al., 2019). This dataset developed by Iida et al. (2021) is widely used to study the relationship between the $p$CO$_2$ low-frequency variability and the recent global warming hiatus and the Interdecadal Pacific Oscillation, the net community production and ocean acidification (Hashihama et al., 2021; Qiu et al., 2021; Ono et al., 2023).

## 2.3 Eddy Database

"Normal" and "abnormal" eddies are from the eddy dataset developed by Liu et al. (2021), using a deep learning (DL) model based on the fusion of satellite sea surface height (SSH) and SST data. Based on the U-Net framework (Ronneberger et al., 2015; Falk et al., 2019), the model combines HyperDense-Net (Dolz et al., 2019) to fuse SSH and SST data. The SSH data used in the model is obtained from the Archiving, Validation, and Interpretation of Satellite Oceanographic (AVISO) dataset. The SST data refers to the NOAA SST product (Reynolds et al., 2007). The model simultaneously extracts SSH anomaly (SSHA) features to determine eddy locations and distinguish between AEs and CEs and extracts the mean SST anomaly (SSTA) within eddy boundaries to distinguish between "normal" and "abnormal" eddies. Specifically, WAEs are identified according to SSHA >0 and SSTA >0, CAEs are identified according to SSHA >0 and SSTA <0, CCEs are identified according to SSHA <0 and SSTA <0, and WCEs are identified according to SSHA <0 and SSTA >0. The dice loss (a cost function to calculate the difference between the predicted and true values) and accuracy of the model was about 14 % and 94 % when training with the ground truth data set, generated automatically using the SSH-based method (Liu et al., 2016). The eddy data set has a daily and 0.25° resolution, including the number, radius, amplitude, rotational speed, and eddy kinetic energy (EKE) in the global ocean from 1996 to 2015. Due to the limitations of the resolution capability of the SSHA data (Ducet et al., 2000), eddies with amplitudes < 2 cm and radii < 35 km were discarded in this work.

Compared to the AVISO eddy database (Pegliasco et al., 2022), our study utilizes a different eddy detection method (Liu et al., 2021). The reason why we use this method is that DL technology has unparalleled learning ability and the capability to model complex nonlinear relationships compared to traditional statistics and machine learning methods (Reichstein et al., 2019). Besides, our method achieves great accuracy and much higher efficiency than the traditional method that first detects the eddies and then uses the SST signature to classify them into "normal" and "abnormal" eddies (Liu et al., 2021). In addition, the method is able to detect eddies in regions where traditional methods may not be effective, such as in regions with weak eddies or regions with complex oceanic dynamics (Liu et al., 2021). Given its high accuracy and comprehensive information on eddy characteristics, we find this dataset particularly useful for our study. Considering that the changes in SSH, SST, Chl-$a$, and roughness caused by eddies can be recorded by altimeter, infrared, ocean color, and synthetic aperture

radar (SAR) remote sensing, respectively, and potential density and temperature recorded by Argo floats can also identify "abnormal" eddies, in future work, we will combine multiple remote sensing data with Argo profiles to evaluate the accuracies of "abnormal" eddy identification method.

## 3 Eddy Analysis Methodology

### 3.1 Composite Eddy-induced Anomalies

To extract the eddy-induced mesoscale features in sea surface variables, including SST, Chl-$a$, DIC, and $p$CO$_2$, we use temporal and spatial filters similar to those used in Villas Bôas et al. (2015) (Fig. S2). The temporal filter is a band-pass Butterworth window (Butterworth, 1930) applied to preserve the temporal signal between 7 and 90 days corresponding to the typical time scales of eddies. The SST and Chl-$a$ anomalies are computed using the 7-90 days band-pass filter to remove the seasonal signal. However, for DIC and $p$CO$_2$ datasets with the monthly temporal resolution, we subtract their climatological
averages. The spatial filter is a moving average Hann window (Stearns and Ahmed, 1976) designed to contain spatial signals smaller than 600 km. This filter removes large-scale variability unrelated to the mesoscale eddy influence.

Finally, we use the eddy-centric composite method to estimate the spatial pattern of the eddy-induced anomalies in sea surface variables. The positions of co-located SST, Chl-$a$, DIC, and $p$CO$_2$ observations are normalized by R, which defines the edge of an eddy as ±1 and the eddy core as 0. This allows us to construct composite averages from eddies of various sizes.
The specific method to calculate eddy-centric composite maps is demonstrated in Text S1. The composite maps are not rotated with the background variables gradient, as the large-scale background variables gradients in the SO are predominantly oriented north-south. Previous studies have indicated that rotating eddies to the large-scale variables gradient in the SO has a negligible impact on the results (Frenger et al., 2015). Therefore, the axes in each figure point north and east. This eddy-centric composite method is frequently used in studies of eddy tracer anomalies (Hausmann and Czaja, 2012;
Gaube et al., 2013; Gaube et al., 2014; Frenger et al., 2015; Gaube et al., 2015; Dawson et al., 2018). Its advantage lies in the ability to average over multiple eddies, which helps reduce noise and reveal persistent eddy structures (Melnichenko et al., 2017). This method is particularly useful when studying eddies in regions where eddy activity is highly variable, as it allows us to identify common patterns and trends in eddy-induced anomalies.

### 3.2 Eddy-induced Ekman pumping

At present, there are no explicit formulas to quantify eddy stirring, trapping, and pumping, but with the Ekman transport modified by the surface geostrophic vorticity $\zeta$ following Stern (1965), the total eddy-induced Ekman pumping $W_{tot}$ is

$$
\begin{aligned}
W_{tot} &= \frac{1}{\rho_o} \nabla \times \left[ \frac{\tau}{(f+\zeta)} \right] \\
&\approx \frac{\nabla \times \tau}{\rho_o (f+\zeta)} + \frac{1}{\rho_o (f+\zeta)^2} \left( \tau^x \frac{\partial \zeta}{\partial y} - \tau^y \frac{\partial \zeta}{\partial x} \right),
\end{aligned}
\tag{2}
$$

where $\rho_o$ = 1020 kg m$^{-3}$ is the (assumed constant) density of sea surface water, $f$ is the Coriolis parameter, and the surface stress $\tau$ has zonal and meridional components $\tau^x$ and $\tau^y$, respectively. The surface stress curl $\nabla \times \tau$ was computed by using finite centered differences of $\tau^x$ and $\tau^y$. The surface geostrophic vorticity $\zeta$ is calculated as

$$\zeta = \frac{\partial v}{\partial x} - \frac{\partial u}{\partial y}, \tag{3}$$

where $u$ and $v$ are the zonal and meridional components of geostrophic current velocity. The surface wind stress $\tau$ is calculated as

$$\tau = \rho_a C_D (U_a - U_o)|U_a - U_o|, \tag{4}$$

where $\rho_a$ = 1.2 kg m$^{-3}$ is the air density (assumed constant), $C_D$ is the drag coefficient, $U_a$ and $U_o$ are the wind and ocean current vectors, respectively. The above formulas to calculate the $W_{tot}$ are similar to those used in Gaube et al. (2015).

$U_a$ is a gridded Level 4 (L4) product with 0.25° resolution available every six hours from the Cross-Calibrated Multi-Platform (CCMP) ocean surface wind data set, produced by Remote Sensing Systems. The data set combines ocean surface (10m) wind retrievals from a reanalysis background field from the ERA-Interim reanalysis, multiple types of satellite microwave sensors, and observations from ships and buoys. The $U_o$ is a daily sea surface geostrophic current product with a spatial 0.25° resolution obtained from AVISO.

To extract the mesoscale features of $W_{tot}$, we use temporal and spatial filters similar to those used in Gaube et al. (2015). The $W_{tot}$ is temporally low-pass filtered with a half-power filter cutoff of 30 days and spatially high-pass filtered to contain spatial signals smaller than 600 km. Finally, we use the eddy-centric composite method to obtain the spatial pattern of eddy-induced Ekman pumping.

## 4 Results

### 4.1 Spatial Distributions of "Normal" and "Abnormal" Eddies

From 1996 to 2015, an average of 1991 eddies were identified daily in the SO (65°S–30°S), with "abnormal" eddies accounting for 26.3 % of the total. Figures 2a, b, d, and e show the spatial distribution of eddy number, defined as the frequency of eddy occurrence in each 1° × 1° latitude-longitude bin over the analyzed period 1996–2015. The eddy frequency represents the ratio of the number of days eddies appeared to the total number of observation days. Eddies disappear in the regions shallower than 2000m and the area near Antarctica (shown in gray in Fig. 2) because the bottom topography constrains the generation of eddies, and satellite altimetric cannot measure sea level beneath sea ice (Frenger et al., 2015). "Normal" and "abnormal" eddies are concentrated in strong currents regions, such as the ACC, Western Boundary Current (WBC), and Eastern Boundary Current (EBC) regions, as shown in Fig. 1. These findings are consistent with those findings reported by Frenger et al. (2015), which did not distinguish between "normal" and "abnormal" eddies. The

differences between AEs and CEs, i.e., the eddy polarity, are critical for understanding the physical and biogeochemical anomalies induced by eddies (Mcgillicuddy et al., 1998; Siegel et al., 2011). "Abnormal" eddies have the opposite polarity distribution to "normal" eddies in the continental boundary currents where more CCEs and CAEs occur. The most

significant difference in polarity distributions between "normal" and "abnormal" eddies is the dominance of WAEs and WCEs in the southwestern Australia (SWA) (Figs. 2c, f).

Despite the great differences in occurrence distributions, the amplitude distributions of the four types of eddies are similar. The eddy amplitude is greater in the Brazil Malvinas Confluence (BMC), Agulhas Return Current (ARC), ACC, SWA, and Tasman Sea (Figs. 2g, h, j, and k). One should note that the amplitudes of "abnormal" eddies are smaller than those of

"normal" ones (Table S2), which is consistent with previous studies (Liu et al., 2020; Liu et al., 2021). In addition, the spatial distributions of rotational speed and EKE correlate well with the patterns of eddy amplitude (Fig. S3).

We investigated the tracks of "normal" and "abnormal" eddies and found that they are consistent (Fig. 3 and Table S3). To accurately represent eddy propagation directions, we incorporated statistics that encompass a broader range of eddy lifetimes, including both short-lived and long-lived eddies (living longer than 1 year) (Table S3). Regardless of the lifespan, both AEs

and CEs propagate primarily westward and northward. By contrast, AEs and CEs with lifetimes longer than 1 year propagate primarily northward and southward, respectively, corresponding with the intrinsic meridional propagation of eddies (Cushman-Roisin and Beckers, 2011). Frenger et al. (2015) reported that only partial eddies follow this intrinsic meridional propagation in the SO, owing to the strong overcompensation by the background meridional deflections of the mean current. Figure 3 shows that between 30°S and the ACC, the major propagation direction of eddies is westward, with AEs

propagating north and CEs propagating south. However, most eddies in the ACC influence area propagate eastward, with AEs propagating south and CEs propagating north. These results are similar to those reported by Dawson et al. (2018).

## 4.2 Spatial Distributions of Eddy-induced SST, Chl-*a*, DIC, and *p*CO$_2$ Anomalies

Using the eddy-centric composite method (Fig. S2), we average the eddy-induced SST, Chl-*a*, DIC, and *p*CO$_2$ anomalies into $1° \times 1°$ longitude-latitude grid boxes. The maps of the climatological imprint of eddies on SST show that the distributions of

SST anomalies over "normal" eddies correlate well with the amplitude distributions, with stronger positive/negative SST anomalies (in WAEs/CCEs) concentrated in the BMC, ARC, ACC, SWA, and Tasman Sea (Figs. 4a, c). In contrast, in regions with larger amplitudes, CAEs/WCEs have weaker negative/positive SST anomalies (Figs. 4b, d).

Also, the distributions of Chl-*a* anomalies over both "normal" and "abnormal" eddies are similar to the amplitude distributions, with stronger negative/positive anomalies within AEs/CEs in regions with higher amplitudes (Figs. 4e–h).

However, in the south of the ACC, including the ACC, we find the patterns of Chl-*a* anomalies appear spotty, with average positive and negative Chl-*a* anomalies in AEs and CEs, respectively. As shown in Fig. S4, the correlation coefficients between amplitude and Chl-*a* anomalies have larger magnitudes in subtropical waters, with negative values in WAEs and CAEs and positive values in CCEs and WCEs. This result illustrates that in subtropical regions with higher amplitudes, such

as BMC, ARC, and Tasman Sea, WAEs and CAEs induced stronger negative Chl-*a* anomalies, while CCEs and WCEs induced stronger positive Chl-*a* anomalies.

The distributions of DIC anomalies differ significantly from those of SST and Chl-*a* anomalies, with uniform speckles featuring average negative DIC anomalies in WAEs and WCEs and positive DIC anomalies in CAEs and CCEs (Figs. 4i–l). In addition to the opposite DIC anomaly signals between "normal" and "abnormal" eddies, the magnitudes of DIC anomalies are generally larger in "normal" eddies than in "abnormal" eddies.

The patterns of eddy-induced $pCO_2$ anomalies are zonal. For AEs, $pCO_2$ anomalies are positive in the north of ACC and negative along the ACC, while the opposite is true for CEs (Figs. 4m–p). However, there are some distinctions between "normal" and "abnormal" eddies. For example, in the north of ACC (including SWA) with high SST (Fig. 5a1) and low DIC (Fig. 5b1), WAEs and WCEs have more positive speckles compared to CAEs and CCEs, respectively. Conversely, along the ACC (including ARC) with low SST (Fig. 5a1) and high DIC (Fig. 5b1), WAEs and WCEs have more negative speckles than CAEs and CCEs.

These findings indicate variability in the spatial distribution of physical and biogeochemical parameters induced by "normal" and "abnormal" eddies. To further quantify the effects of eddies on different parameters, we average all eddy-centric composite maps for SST, Chl-*a*, DIC, and $pCO_2$ anomalies over eddies to analyze the pattern characteristics of eddy-induced parameters. Due to seasonal variations in $pCO_2$, eddies' physical and biogeochemical characteristics are also synthesized in summer and winter.

### 4.3 Composite Maps of Eddy-induced SST, Chl-*a*, DIC, $pCO_2$ Anomalies

Using the eddy-centric composite method (Fig. S2), we investigate the seasonal variations of SST, Chl-*a*, DIC, and $pCO_2$ associated with "normal" and "abnormal" eddies in the SO (Fig. 6). Figures 6a1–a4 and e1–e4 show the composite SST anomalies within "normal" and "abnormal" eddies in winter and summer, respectively. There are no significant differences in the signals and spatial patterns of SST anomalies within the same kind of eddies between summer and winter. Composite SST anomalies over "normal" eddies show asymmetric monopole patterns, with positive/negative extrema slightly shifting westward and poleward (equatorward) relative to the WAEs/CCEs cores. In comparison, "abnormal" eddies also display monopole patterns but with opposite signals. Furthermore, the magnitudes of SST anomalies over "normal" eddies are larger than those over "abnormal" eddies.

The composite Chl-*a* anomalies within the same kind of eddies also show no obvious seasonality in the signals and spatial patterns (Figs. 6b1–b4 and f1–f4). Moreover, composite Chl-*a* anomalies have no significant difference between "normal" and "abnormal" eddies, with monopole negative signals in WAEs and CAEs and positive signals in CCEs and WCEs. The extrema of Chl-*a* anomalies slightly shift poleward relative to the cores of WAEs and CAEs and equatorward relative to the cores of CCEs and WCEs.

Regarding eddy-induced DIC anomalies, their composite maps within the same kind of eddies are similar in summer and winter, except that the magnitudes of DIC anomalies within eddies are slightly higher in winter (Figs. 6c1–c4 and g1–g4).

Moreover, the composite DIC anomalies within "normal" and "abnormal" eddies show dipole patterns dominated by opposite signals. WAEs are dominated by negative DIC anomalies, whereas CAEs are dominated by positive DIC anomalies. CCEs are dominated by positive DIC anomalies, whereas WCEs are dominated by negative DIC anomalies. The DIC anomalies have opposite signals to SST anomalies within the same kind of eddies.

Although $pCO_2$ is influenced by SST, Chl-$a$, and DIC, $pCO_2$ anomalies within eddies in winter are significantly different from summer, unlike SST, Chl-$a$, and DIC anomalies within eddies similar in summer and winter (Figs. 6d1–d4 and h1–h4). To determine which factor dominates the change in $pCO_2$, we calculate the structural similarity index (SSIM) in Eq. (5), which can quantify the similarity of the patterns between $pCO_2$ and other anomalies over eddies (Wang et al., 2004).

$$SSIM\,(X,Y) = \frac{(2\mu_X\mu_Y + D_1)(2\sigma_{XY} + D_2)}{(\mu_X^2 + \mu_Y^2 + D_1)(\sigma_X^2 + \sigma_Y^2 + D_2)}$$
$$D_1 = (k_1 L_p)^2 \tag{5}$$
$$D_2 = (k_2 L_p)^2$$

where $X$ and $Y$ denote composite averages of normalized $pCO_2$ and DIC (SST) anomalies, respectively. $\mu_X$ and $\mu_Y$ are the average values of $X$ and $Y$. $\sigma_X$ and $\sigma_Y$ are the standard deviations of $X$ and $Y$. $\sigma_{XY}$ is the covariance of $X$ and $Y$. $L_p$ is the dynamic range of values, $L_p = 2$. $k_1 = 0.01$ and $k_2 = 0.03$. SSIM ranges from $-1$ to $1$. The closer the SSIM value is to 1, the more similar the two patterns are. Because the Chl-$a$ negatively correlates with the $pCO_2$, its SSIMs are multiplied by $-1$. In winter, the SSIMs between $pCO_2$ and DIC anomalies are the largest ($>0.9$). The $pCO_2$ anomalies have similar patterns and signals with DIC anomalies, dominant by positive signals within CAEs and CCEs and negative signals within WAEs and WCEs (Figs. 6d1–d4). However, in summer, the SSIMs are negative between $pCO_2$ and DIC anomalies but positive between $pCO_2$ and SST (Chl-$a$) anomalies over eddies in summer ($\leq 0.35$). The patterns of $pCO_2$ anomalies differ from those of SST, Chl-$a$, and DIC within eddies in the SO (Figs. 6h1–h4).

## 5 Modulation Mechanisms of "Normal" and "Abnormal" Eddies to Physical and Biogeochemical Parameters

This section discusses how eddies affect SST, Chl-$a$, DIC, and $pCO_2$ through various mechanisms, including eddy stirring, trapping, pumping, and eddy-induced Ekman pumping (Fig. 7).

### 5.1 Mechanism Analysis of Eddy's Influence on SST

Composite SST anomalies over eddies show monopole patterns, with positive anomalies in WAEs and WCEs and negative anomalies in CCEs and CAEs (Figs. 6a1–a4, e1–e4, S5a and Table S4). First, we analyze the effect of eddy trapping on SST, which is determined by the directions of horizontal SST gradient and eddy propagation (Frenger et al., 2015; Frenger et al., 2018). The climatological and seasonal averages of SST reveal a zonal distribution with a southward decrease (Figs. 5a1–a3). Table S3 shows that the predominant propagation direction of eddies is westward and northward (Fig. 3). According to the

southward decreasing SST, northward propagating eddies would trap cold water and result in negative SST anomalies. However, this process contradicts the positive SST anomalies within WAEs and WCEs, indicating the weak effect of eddy trapping on SST.

The meridional and zonal phase shifts in "normal" eddies are proposed to be induced by the large-scale background SST gradient and eddy stirring (Hausmann and Czaja, 2012; Villas Bôas et al., 2015). Specifically, WAEs rotating counterclockwise through the SST gradient would advect warmer water from the north to the southeast, leading to positive extrema slightly shifting westward and poleward relative to the cores (Figs. 6a1, e1). Conversely, CCEs rotating clockwise through the SST gradient would advect cooler water from the south to the northwest, leading to negative extrema slightly shifting westward and equatorward relative to the cores (Figs. 6a3, e3). In summary, for the advective effects of eddies, the effect of eddy trapping on SST is not reflected, and eddy stirring contributes to the slight shift of SST anomalies extrema within "normal" eddies.

For the vertical effects of eddies, eddy pumping within AEs associated with downwelling induces positive SST anomalies, while eddy pumping within CEs associated with upwelling induces negative SST anomalies (Fig. 7b). This process is consistent with the observed SST anomalies within "normal" eddies (Figs. 6a1, a3, e1, and e3). On the other hand, eddy-induced Ekman pumping within AEs is associated with upwelling induces negative SST anomalies, while eddy-induced Ekman pumping within CEs is associated with downwelling induces positive SST anomalies (Fig. 7b) (Gaube et al., 2013; Dawson et al., 2018). This process is consistent with the observed SST anomalies within "abnormal" eddies (Figs. 6a2, a4, e2, and e4)Moreover, WAEs/CCEs have stronger positive/negative SST anomalies in the regions with larger amplitude, while CAEs/WCEs have weaker negative/positive SST anomalies (Figs. 4a–d). We further compared the quantitative relationship between SST anomalies and amplitudes over "normal" and "abnormal" eddies, as shown in Fig. S6. The SST anomalies in WAEs and CAEs are positively correlated with amplitudes, whereas the SST anomalies in CCEs and WCEs are negatively correlated with amplitudes. These findings indicate that the strength of eddy pumping is positively correlated with eddy amplitude, i.e., larger amplitude corresponds to stronger downwelling and upwelling in the cores of AEs and CEs, respectively. Table S2 shows that the amplitudes of "abnormal" eddies are smaller than "normal" eddies, indicating weaker eddy pumping in "abnormal" eddies. Hence, "abnormal" eddies are more likely to be influenced by eddy-induced Ekman pumping. In summary, within "normal" eddies, eddy pumping dominates the vertical heat advection, resulting in positive SST anomalies in WAEs and negative SST anomalies in CCEs (Figs. 6a1, a3, e1, and e3). However, within "abnormal" eddies, the effect of eddy-induced Ekman pumping becomes more prominent, resulting in negative SST anomalies in CAEs and positive SST anomalies in WCEs (Figs. 6a2, a4, e2, and e4).

**5.2 Mechanism Analysis of Eddy's Influence on Chl-*a***

The composite maps of eddy-induced Chl-*a* anomalies in the SO show asymmetric monopole patterns, with negative/positive extrema shifting poleward/equatorward relative to the AEs/CEs cores (Figs. 6b1–b4 and f1–f4). We calculate the climatological average gradient of Chl-*a* in the SO from 1996 to 2015, which is normalized before calculation.

The north-south gradient of Chl-*a* is −0.02 (north is the positive direction), and the east-west gradient of Chl-*a* is −0.04 (east is the positive direction). Due to the climatological Chl-*a* increasing southward (Figs. 5b1–b3), eddies propagating northward tend to trap high Chl-*a* into northern areas with low Chl-*a*. Likewise, due to the climatological Chl-*a* increasing westward, eddies propagating westward tend to trap low Chl-*a* into western areas with high Chl-*a*. However, the effect of eddy trapping on Chl-*a* cannot explain the opposite Chl-*a* anomalies between AEs and CEs (Figs. 6b1, b2, f1, and f2). Consequently, it can be inferred that the role of eddy trapping in influencing Chl-*a* distributions is limited.

Moreover, considering that the climatological Chl-*a* increases southward and westward, the counterclockwise rotation of AEs in the SO would advect low Chl-*a* from the northeast to the west and high Chl-*a* from the southwest to the east. The reverse is true for CEs. Previous works found that the dipole shapes arising from stirring tend to be asymmetric, with larger anomalies on the leading side compared to the trailing side of eddies (Figs. 7a, c) (Chelton et al., 2011b; Frenger et al., 2015; Dawson et al., 2018; Frenger et al., 2018). As the major propagation direction of eddies is westward (Table S3), the composite Chl-*a* anomalies in AEs/CEs show dominant negative/positive signals due to eddy stirring (Figs. 6b1–b4 and f1–f4). Like SST anomaly patterns, Chl-*a* anomaly patterns also show meridional and zonal extremum shifts. For meridional shifts, AEs rotating counterclockwise through the southward increasing Chl-*a* gradient would induce negative extrema slightly shifting poleward relative to the cores (Figs. 6b2, f1, and f2). The reverse is true for CEs (Figs. 6b3, b4, f3, and f4). For zonal shifts, AEs through the westward increasing Chl-*a* gradient would induce negative extrema slightly shifting westward relative to the cores (Figs. 6b2, f1, and f2). The reverse is true for CEs (Figs. 6b3, b4, f3, and f4).

Eddy pumping induces negative and positive Chl-*a* anomalies within AEs and CEs, respectively, on the contrary, eddy-induced Ekman pumping induces positive and negative Chl-*a* anomalies within AEs and CEs, respectively (Dawson et al., 2018). The eddy-centric composite maps of Chl-*a* anomalies show monopole negative signals in AEs and positive signals in CEs (Figs. 6b1–b4 and f1–f4). Besides, in regions of higher amplitude, the magnitudes of eddy-induced Chl-*a* anomalies are greater (Figs. 4e–h). These results reflect the dominant effect of eddy pumping on Chl-*a* anomalies within eddies. However, the more evident Ekman pumping mechanism of "abnormal" eddies resists eddy pumping and leads to lower Chl-*a* magnitude within "abnormal" eddies than "normal" eddies (Figs. 6b1–b4 and f1–f4). It is worth noting that in some regions with small amplitude, such as the south of ACC and the South Pacific Ocean, we observe positive Chl-*a* anomalies in AEs and negative Chl-*a* anomalies in CEs (Figs. 4e–h). Such a result may be caused by a more dominant effect of eddy-induced Ekman pumping on Chl-*a*. Overall, eddy stirring and eddy pumping are mainly responsible for the patterns of Chl-*a* anomalies within eddies in the SO, and eddy-induced Ekman pumping attenuates the magnitudes of Chl-*a* anomalies within "abnormal" eddies.

### 5.3 Mechanism Analysis of Eddy's Influence on DIC

The composite DIC anomalies within "normal" and "abnormal" eddies show dipole patterns with opposite dominant signals, negative in WAEs and WCEs and positive in CAEs and CCEs in the SO (Figs. 6c1–c4, g1–g4, S5c and Table S4). Due to the climatological DIC increasing southward, the counterclockwise rotation of AEs in the SO would advect low DIC from

375 the north to the southwest and high DIC from the south to the northeast. The reverse is true for CEs. As eddies migrate westward, negative and positive DIC anomalies in AEs and CEs on the western side would intensify, affected by asymmetric dipole shapes arising from eddy stirring (Figs. 7a, d). Under the condition of southward increasing DIC (Figs. 5c1–c3), eddies propagating northward tend to trap high DIC. Thus, the effect of eddy trapping may contribute to the positive signals of DIC anomalies within eddies. However, the opposite dominant signals between "normal" and "abnormal" eddies cannot

be explained solely by the advective effects of eddies.

Eddy pumping induces negative DIC anomalies within AEs through the downwelling of surface low-DIC waters and positive DIC anomalies within CEs through the upwelling of deep rich-DIC waters. The reverse is true for the effect of eddy-induced Ekman pumping on DIC anomalies. As mentioned in Section 5.1, the greater amplitude of eddies corresponds to stronger eddy pumping. The larger amplitude of "normal" eddies than "abnormal" eddies leads to higher magnitudes of

385 negative DIC anomalies in WAEs and positive anomalies in CCEs. Moreover, the Ekman pumping caused by WAEs is stronger than that caused by CAEs (Figs. 8a1, a2), resulting in stronger positive DIC anomalies within WAEs than CAEs (Figs. 6c1, c2, g1, and g2). Similarly, the Ekman pumping caused by WCEs is stronger than that caused by CCEs (Figs. 8a3, a4), resulting in stronger negative DIC anomalies within WCEs than CCEs (Figs. 6c3, c4, g3, and g4). Consequently, the combined effects of eddy pumping and eddy-induced Ekman pumping contribute to the dominant negative DIC anomalies

within WAEs and WCEs and positive DIC anomalies within CAEs and CCEs in the SO.

### 5.4 Mechanism Analysis of Eddy's Influence on $pCO_2$

In winter, the $pCO_2$ anomalies have similar patterns and signals with DIC anomalies, and the SSIMs between $pCO_2$ and DIC anomalies are the highest in the SO (Figs. 6d1–d4), suggesting that the effect of DIC on $pCO_2$ is stronger than the effects of SST and Chl-$a$. However, in summer, the patterns of $pCO_2$ anomalies differ significantly from the anomalies of SST, Chl-$a$,

and DIC within eddies in the SO with relatively lower SSIMs (the highest SSIM is 0.35) (Figs. 6h1–h4). This result may be caused by different processes affecting $pCO_2$ in different regions of the SO. To prove this hypothesis, we further examine the eddy-induced SST, Chl-$a$, DIC, and $pCO_2$ anomalies in the SWA (95°–115°E, 30°–40°S) and ARC (25°–75°E, 35°–45°S), where the eddy activity is strong, and the eddy amplitude and rotation speed are high (Figs. 2, S3, magenta rectangular box), leading to strong eddy stirring, trapping and pumping (Dawson et al., 2018; Frenger et al., 2018).

Similar to the SO, the SSIMs between $pCO_2$ and DIC anomalies are the highest in both the SWA and ARC during winter, indicating the dominant effect of DIC on $pCO_2$ (Figs. 9d1–d4 and 10d1–d4). However, unlike the SO, the SSIMs between $pCO_2$ and SST anomalies are the highest in the summer SWA (Figs. 9h1–h4). By contrast, the SSIMs between $pCO_2$ and DIC anomalies are the highest in summer ARC (Figs. 10h1–h4). These results suggest that in summer, the $pCO_2$ within eddies is dominated by the SST effect in the SWA and dominated by the DIC effect in the ARC. Despite similar magnitudes

of SST anomalies over eddies between the SWA and ARC, the magnitudes of DIC anomalies in the SWA are significantly lower than those in the ARC, which may cause different processes affecting $pCO_2$ in these two regions. Likewise, the $pCO_2$ anomalies over eddies are determined by the DIC anomalies in winter, which is also associated with the higher magnitudes

of DIC anomalies in winter compared to summer (Figs. 6c1–c4 and g1–g4). Such regional and seasonal magnitude variations of DIC anomalies are controlled by the complex interactions among processes such as biological activity (production/remineralization), vertical mixing, and air-sea gas exchanges (Racapé et al., 2010).

We further calculate the contributions of eddies to $pCO_2$ (Table S5). On average, the contributions of "abnormal" eddies to $pCO_2$ are generally smaller than those of "normal" eddies in the SO and ARC. Nevertheless, the contributions of "abnormal" eddies to $pCO_2$ surpass those of "normal" eddies in the SWA. These findings can be attributed to the dominance of "abnormal" eddies in the SWA, primarily driven by the more pronounced eddy-induced Ekman pumping observed in "abnormal" eddies as compared to "normal" eddies (Figs. 2, 8). This contrast in Ekman pumping between "abnormal" and "normal" eddies is more significant in the SWA than in the SO and ARC, as illustrated in Fig. 8. Additionally, the contributions of the ARC and SWA eddies to $pCO_2$ are higher than the SO eddies, which is caused by the regional cancellation effect in the SO (Figs. 4m–p). In the SO and ARC, the contributions of eddies to $pCO_2$ are higher in winter than in summer (except WCEs in ARC), with a maximum value of 2.64 % (WAEs in the SO) and 5.03 % (CCEs in the ARC). However, in the SWA, the contributions of eddies to $pCO_2$ in summer are higher than in winter, with a maximum value of 5.15 % in WCEs, which is about 2.7 times higher than that of CCE. In summary, the contributions of eddies to $pCO_2$ vary depending on the eddy type, region, and season.

## 6 Discussion and Conclusions

Section 5 reveals distinct influence mechanisms of eddies on SST, Chl-$a$, DIC, and $pCO_2$, which vary based on the inherent properties of each parameter and the complex interactions between eddies and the biogeochemical processes in the SO. As shown in Table 1, we compare the significance magnitudes of different effects, including eddy trapping, stirring, pumping, and eddy-induced Ekman pumping, on SST, Chl-$a$, and DIC. It should be noted that the seasonal modulation of the mixed layer is not discussed in our study due to the absence of significant seasonal variations in eddy-induced SST, Chl-$a$, and DIC anomalies (Fig. S7). Additionally, the variability of $pCO_2$ anomalies within eddies is controlled by the effects of SST, Chl-$a$, and DIC, therefore, the eddy-driven mechanisms on $pCO_2$ can be demonstrated by exploring the effects of eddies on SST, Chl-$a$, and DIC.

Compared to SST, eddy stirring plays a more significant role in Chl-$a$ and DIC anomalies within eddies. As eddy stirring redistributes physical and biogeochemical parameters spatially through horizontal advection, the larger the horizontal parameter gradient, the stronger the eddy stirring effect (Mcgillicuddy, 2016). We calculate the average gradients of normalized SST, Chl-$a$, and DIC in the SO from 1996 to 2015 and find their values are 0.05, 0.11, and 0.20, respectively. The specific method to obtain the gradients is demonstrated in Text S2 (Quarteroni et al., 2006). The small gradient of SST leads to a negligible effect of eddy stirring and results in more pronounced monopole patterns within eddies than other variables (Figs. 6a1–a4 and e1–e4).

By contrast, the average gradient of Chl-*a* is nearly two times higher than that of SST, thus, eddy stirring can cause a stronger effect on Chl-*a*. Both eddy stirring and eddy pumping contribute to the generation of negative/positive Chl-*a* anomalies within AEs/CEs. The combined effects of eddy stirring and eddy pumping dominate the similar patterns of Chl-*a* anomalies in "normal" and "abnormal" eddies. However, the effect of eddy-induced Ekman pumping on Chl-*a* is relatively small and contributes to attenuating the magnitudes of Chl-*a* anomalies within "abnormal" eddies (Figs. 6b1–b4 and f1–f4). The major limitation of marine Chl-*a* is the insufficient supplement of nutrients from depth into the euphotic zone (Mahadevan, 2016). The transport of nutrients enriched in deep seawater is mainly controlled by eddy pumping. By contrast, the variations of SST and DIC anomalies are prone to be influenced by heat and carbon exchange at the ocean-atmosphere interface (Gaube et al., 2015; Song et al., 2016), making them susceptible to eddy-induced Ekman pumping. Consequently, Chl-*a* anomalies in "normal" and "abnormal" eddies show similar patterns and signals, whereas SST and DIC anomalies in "normal" and "abnormal" eddies show opposite signals.

Such limited influence of eddy-induced Ekman pumping on Chl-*a* in the SO was also reported by Gaube et al. (2014), who plotted global maps of the cross correlation of Chl-*a* anomalies and SSH, as well as eddy-induced Ekman pumping, revealing a negative correlation between Chl-*a* anomalies and SSH and a negative correlation between Chl-*a* anomalies and eddy-induce Ekman pumping in most areas of the SO. These results indicate that AEs have negative Chl-*a* anomalies and CEs have positive Chl-*a* anomalies, and eddy-induced Ekman pumping does not dominate the variation of Chl-*a* anomalies within eddies. In addition, we obtain the composite averages for Chl-*a* anomalies in the BMC, defined by Gaube et al. (2014) as 305°–330°E and 34°–50°S (Fig. S8). The patterns are similar to those obtained by Gaube et al. (2014), with dominant monopole negative Chl-*a* anomalies within AEs and positive Chl-*a* anomalies within CEs. However, the magnitudes of Chl-*a* anomalies within "abnormal" eddies are lower than "normal" eddies, which is related to the more pronounced impact of "abnormal" eddies in countering eddy pumping through the mechanism of Ekman pumping.

The average gradient of DIC is four times higher than that of SST, indicating that eddy stirring will have a more pronounced impact on DIC than on SST. As a result, the composite DIC anomalies within eddies show dipole patterns (Figs. 6c1–c4 and g1–g4). In addition to the different impacts of eddy stirring on SST and DIC, both eddy pumping and eddy-induced Ekman pumping contribute to the variations in these parameters (Table 1). In "normal" eddies, eddy pumping dominates the vertical distribution of SST and DIC. Within CCEs, the upwelling of cold, DIC-rich deep water induces negative SST anomalies and positive DIC anomalies, whereas the reverse is true for WAEs. However, the influence of eddy-induced Ekman pumping becomes more prominent within "abnormal" eddies. Within WCEs, the downwelling of warm, low-DIC surface waters induces positive SST anomalies and negative DIC anomalies, whereas the reverse is true for CAEs.

The impact of eddies on $p$CO$_2$ anomalies varies by season and region, which arises from the combined effects of SST, Chl-*a*, and DIC. In winter, the dominant DIC-driven effect leads to negative $p$CO$_2$ anomalies in WAEs and WCEs and positive anomalies in CAEs and CCEs (Figs. 6d1–d4). However, in summer, the $p$CO$_2$ anomalies are dominated by the combined effects of SST, Chl-*a*, and DIC (Figs. 6h1–h4). Notably, the $p$CO$_2$ anomalies within eddies are dominated by SST

anomalies in the summer SWA, with smaller magnitudes of DIC anomalies (Fig. 9). In contrast, the $p$CO$_2$ anomalies within eddies are dominated by DIC anomalies in the ARC, with larger magnitudes of DIC anomalies (Fig. 10).

In conclusion, using the eddy-centric composite method, we investigate the effects of "normal" and "abnormal" eddies on the variability of SST, Chl-$a$, DIC, and $p$CO$_2$ in the SO from 1996 to 2015. The distinct modifications in physical and biogeochemical parameters of "abnormal" eddies compared to "normal" eddies stem from the effect of eddy-induced Ekman pumping. Figure S9 illustrates that in low-wind regions, specifically with wind speeds less than 6 m s$^{-1}$, the occurrence of "abnormal" eddies is scarce. Nevertheless, as wind speed progressively increases, the number of "abnormal" eddies generally increases. Considering that eddy-induced Ekman pumping is expected to exert a more pronounced influence in high-wind regions, this result indicates the effect of eddy-induced Ekman pumping on the generation of "abnormal" eddies. Specifically, in the SWA dominated by "abnormal" eddies, the contributions of "abnormal" eddies to $p$CO$_2$ are opposite to "normal" eddies and are about twice as high as "normal" eddies (Table S5). The current research commonly combines all the AEs or CEs and masks the presence of CAEs and WCEs with very different upper ocean properties. Given their abundance, considering the distinct role of "abnormal" eddies when investigating eddy-induced modulation in physical and biogeochemical parameters provides a more accurate estimation of the impact of mesoscale eddies.

The observational-based study of basin-wide surface physical and biogeochemical parameters within SO mesoscale eddies provides important insights into the SO ecosystem and carbon cycling. The spatial redistribution of Chl-$a$ concentrations through eddy stirring and eddy pumping indicates the potential for localized hotspots of productivity and nutrient supply within eddies. Moreover, the impacts of eddy pumping and eddy-induced Ekman pumping on DIC distributions highlight the role of eddies in transporting carbon-rich waters, which can significantly influence the regional carbon budget and oceanic carbon uptake. Understanding the complexity of eddy-driven processes in the SO is crucial for accurately simulating and predicting the biogeochemical dynamics of the SO and its role in the global carbon cycle. Further investigations focusing on the specific mechanisms driving the observed patterns and their consequences for larger-scale oceanic processes will provide valuable insights into the role of mesoscale eddies in the SO.

**Data availability**

All data used in the analysis are available in public repositories. The OISST data is available from https://www.ncei.noaa.gov/data/sea-surface-temperature-optimum-interpolation/v2.1/access/avhrr/. The Chl-$a$ product is available from https://data.marine.copernicus.eu/product/OCEANCOLOUR_GLO_BGC_L3_MY_009_103/services. The $p$CO$_2$ and DIC datasets are available from https://www.data.jma.go.jp/gmd/kaiyou/english/co2_flux/co2_flux_data_en.html. "Normal" and "abnormal" eddies datasets are available from https://figshare.com/s/3c3b03776d9862ac85bc for peer review only. The CCMP vector wind data is available from https://www.remss.com. The AVISO altimeter current product is available from https://data.marine.copernicus.eu/product/SEALEVEL_GLO_PHY_L4_MY_008_047/services. The positions

of the main ACC fronts (Polar Front and Subantarctic Front) are available from http://ctoh.legos.obs-mip.fr/data/southern-ocean-fronts-extraction-form.

**Author contributions**

QL, YL, and XL conceived the project. QL did the writing and original draft preparation. All authors provided feedback on the analysis and interpretation of results and contributed to reviewing and editing the manuscript. All authors have read and agreed to the published version of the manuscript.

**Competing interests**

The authors declare that they have no conflict of interest.

**Acknowledgements**

This work was supported by the Qingdao National Laboratory for Marine Science and Technology, the special fund of Shandong province (No. 2022QNLM050301-2), the Natural Science Foundation of Shandong Province (ZR2020MD083), the National Natural Science Foundation of China (U2006211 and 42306194), the Strategic Priority Research Program of the Chinese Academy of Sciences (XDA19060101, and  XDB42000000), Major scientific and technological innovation projects of Shandong Province (2019JZZY010102), and the CAS Program (Y9KY04101L).

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

① **Brazil Current**
② **Malvinas Current**
③ **Humboldt Current**
④ **East Australian Current**
⑤ **Leeuwin Current**
⑥ **West Australian Current**
⑦ **Agulhas Return Current**
⑧ **Agulhas Current**
⑨ **Benguela Current**

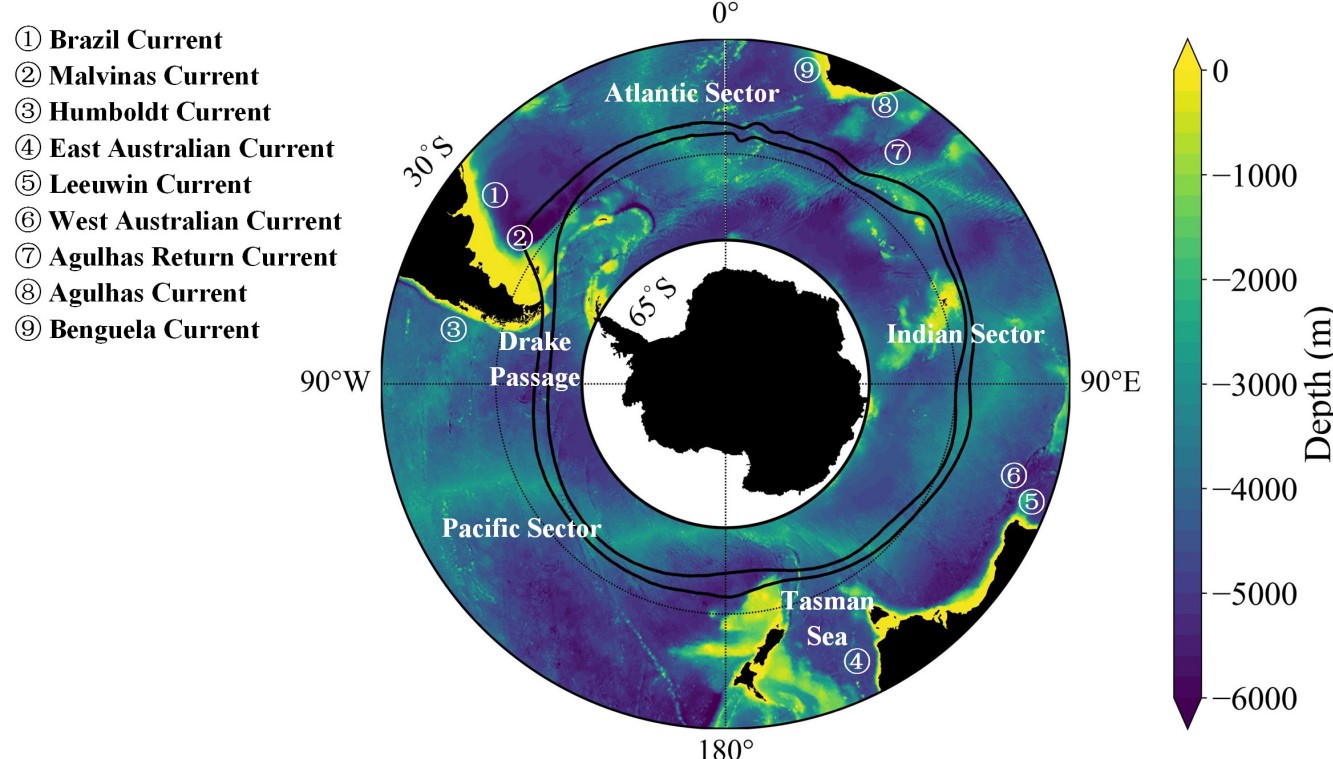

**Figure 1.** Southern Ocean topography and current. Black solid lines show the mean northern (SAF) and southern (PF) positions of the ACC major fronts (Sallée et al., 2008). The black dotted circle is 50° S.

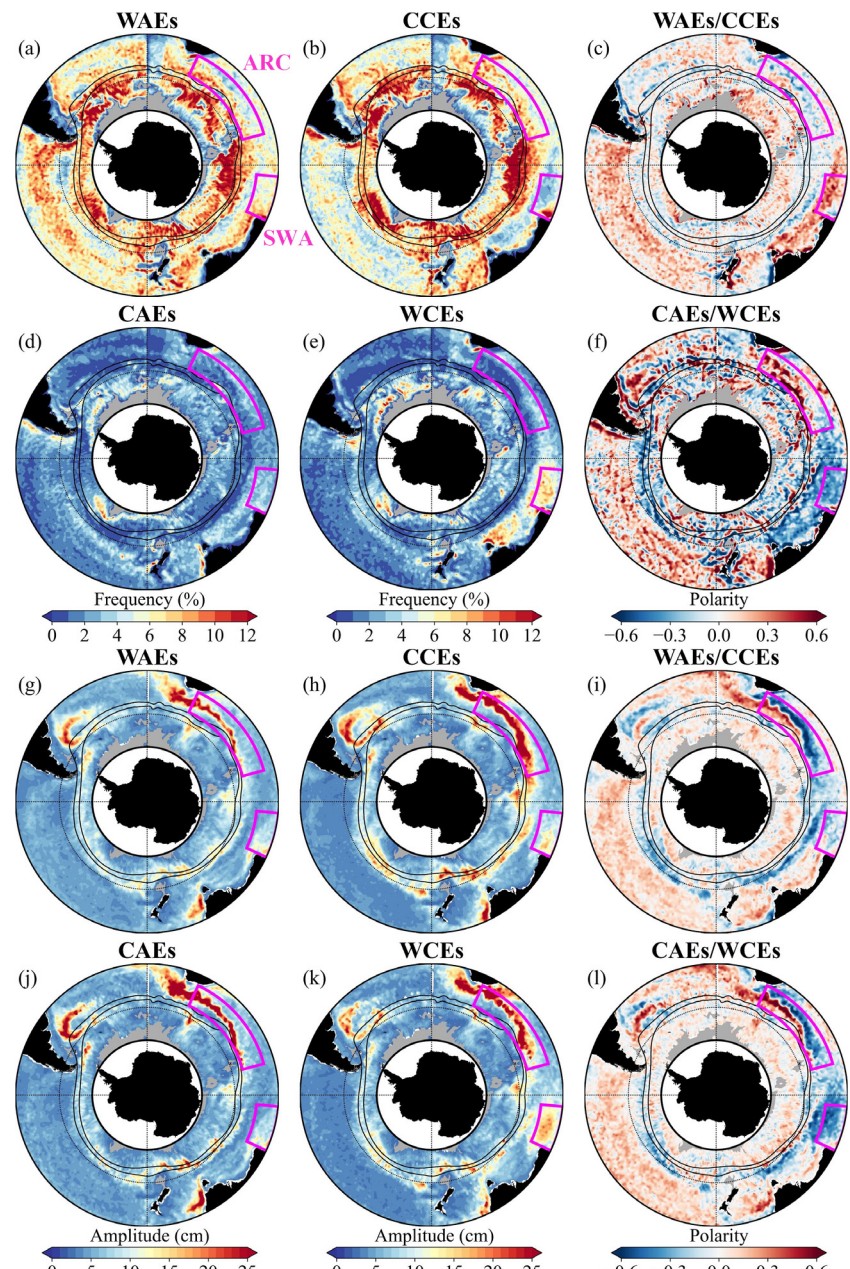

**Figure 2.** Spatial distribution of (a, b, d, and e) eddy frequency, (g, h, j, and k) eddy amplitude, and eddy polarity dominance in the SO from 1996 to 2015. (c, f) Ratio of the area occupied by WAEs/CAEs) over the area covered by CCEs/WCEs). (i, l) Ratio of amplitude for WAEs/CAEs) over CCEs/WCEs). For (c, f, i, l) eddy polarity, values >0 in red and <0 in blue mark the dominance of AEs and CEs, respectively. Black solid lines show the mean northern (SAF) and southern (PF) positions of the ACC major fronts (Sallée et al., 2008). The black dotted circle is 50° S. The magenta boxes represent ARC and SWA regions.

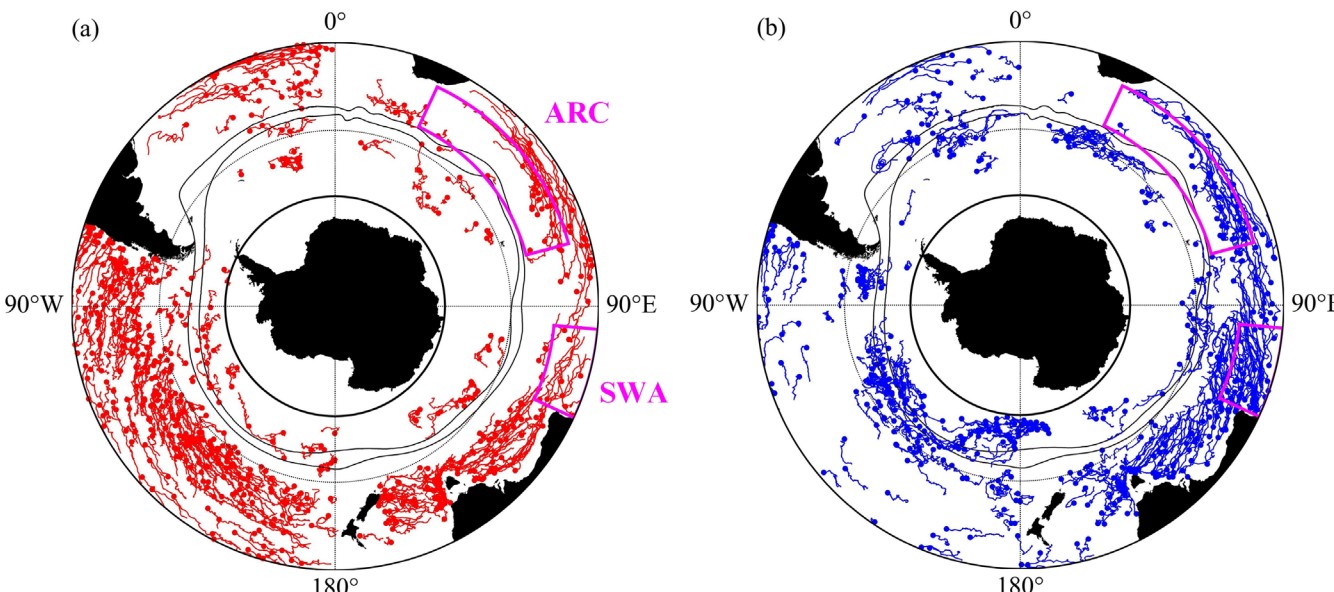

**Figure 3.** Trajectories of (a) AEs and (b) CEs in the SO during 1996–2015. Red (blue) dots and lines mark the AEs (CEs) birth locations and propagation paths. To show the eddies tracks more clearly, only eddies with a minimum lifetime of 1 year and one-third of the long-lived eddies in the south and southwest of Australia have been considered. Black solid lines show the mean northern (SAF) and southern (PF) positions of the ACC major fronts (Sallée et al., 2008). The black dotted circle is 50° S. The magenta boxes represent ARC and SWA regions.

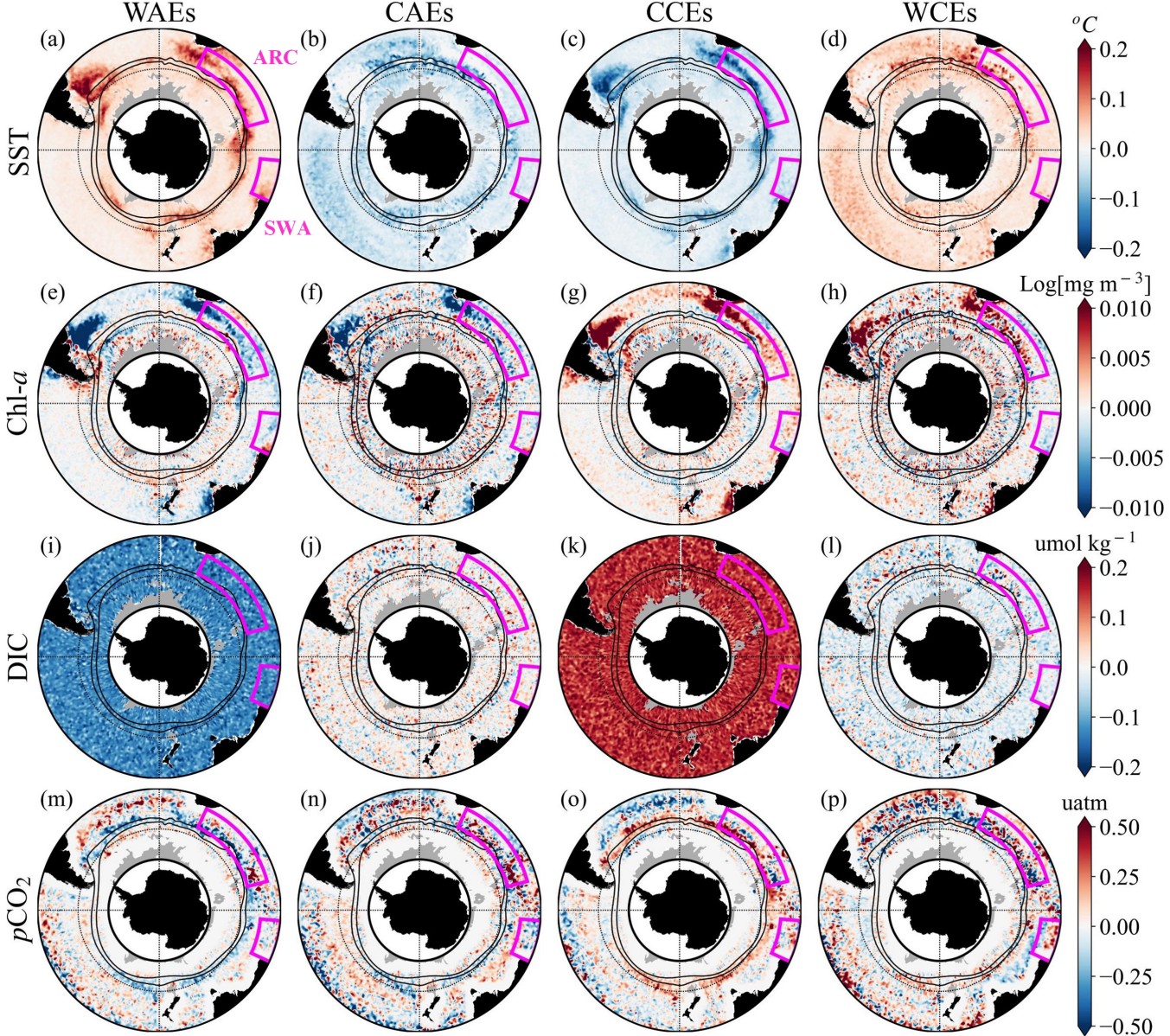

**Figure 4.** Spatial distribution of eddy-induced anomalies of (a–d) SST, (e–h) Chl-*a*, (i–l) DIC, and (m–p) *p*CO$_2$ in the Southern Ocean from 1996 to 2015. The anomalies within eddies are averaged in 1° × 1° longitude-latitude grid boxes. From left to right, columns represent four kinds of eddies. Black solid lines show the mean northern (SAF) and southern (PF) positions of the ACC major fronts (Sallée et al., 2008). The black dotted circle is 50° S. The magenta boxes represent ARC and SWA regions.

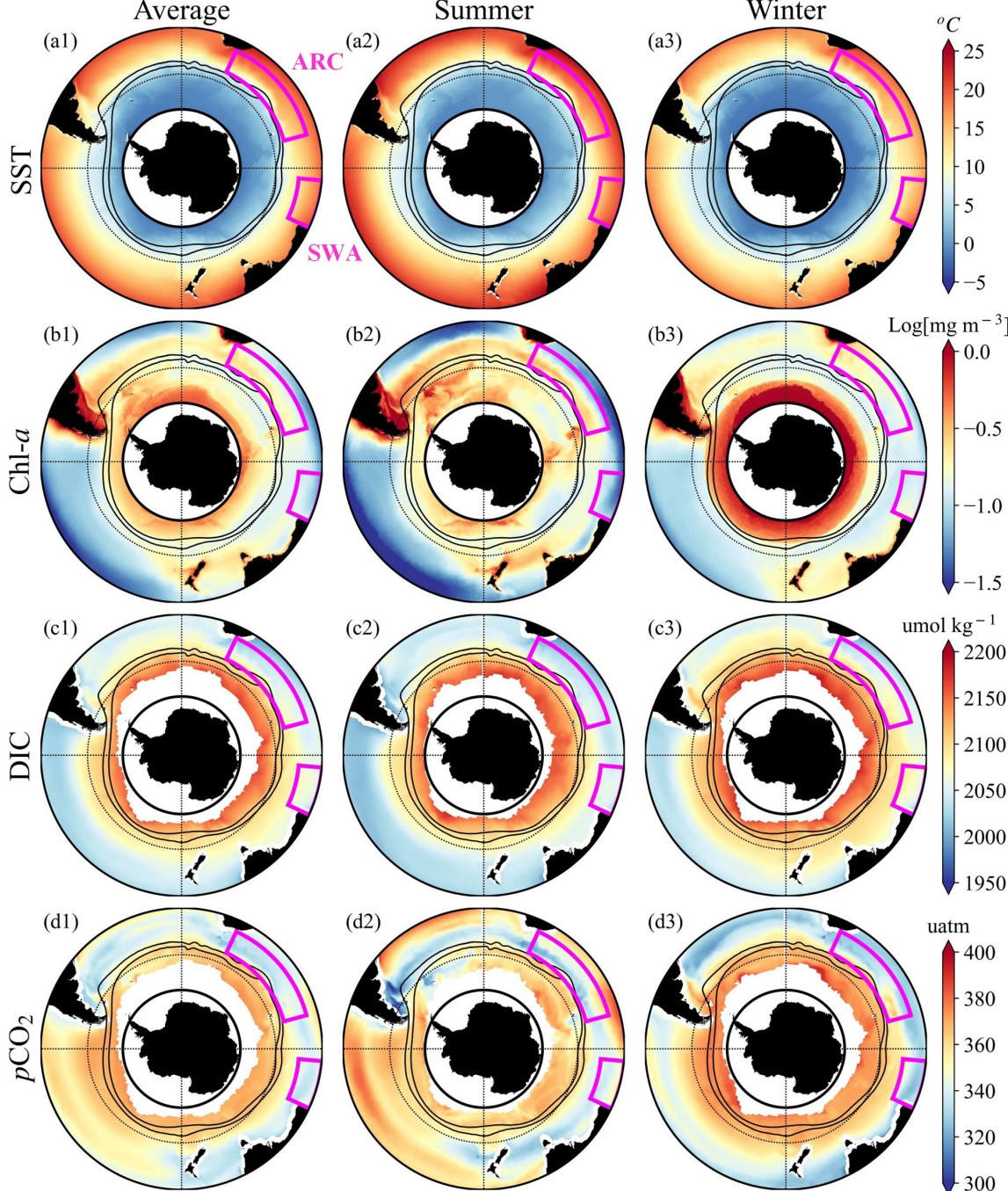

**Figure 5.** The climatological and seasonal averages of (a1–a3) SST, (b1–b3) Chl-*a*, (c1–c3) DIC, and (d1–d3) *p*CO₂ from 1996 to 2015 in the SO. Black solid lines show the mean northern (SAF) and southern (PF) positions of the ACC major fronts (Sallée et al., 2008). The black dotted circle is 50° S. The magenta boxes represent ARC and SWA regions.

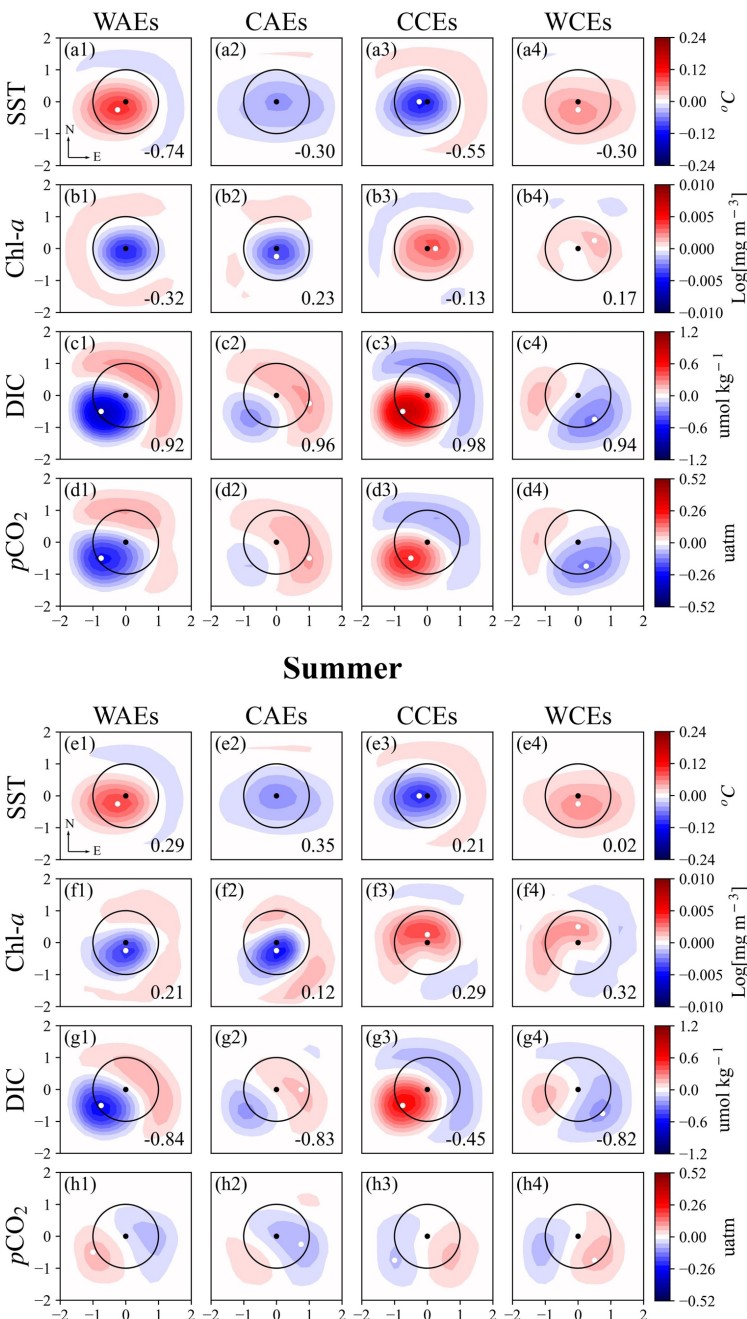

765

**Figure 6.** Eddy-centric composite averages for SST, Chl-*a*, DIC, and $p\mathrm{CO_2}$ anomalies in the SO. On each map, a black dot denotes the eddy center, and a white dot denotes the center location of variables (defined by the location of the extremum value). Contour intervals are every 0.009 °C for SST, every 0.0007 Log[mg m$^{-3}$] for Chl-*a*, every 0.08 umol kg$^{-1}$ for DIC, and every 0.035 uatm for $p\mathrm{CO_2}$. The numbers in the lower right corner are the SSIMs.

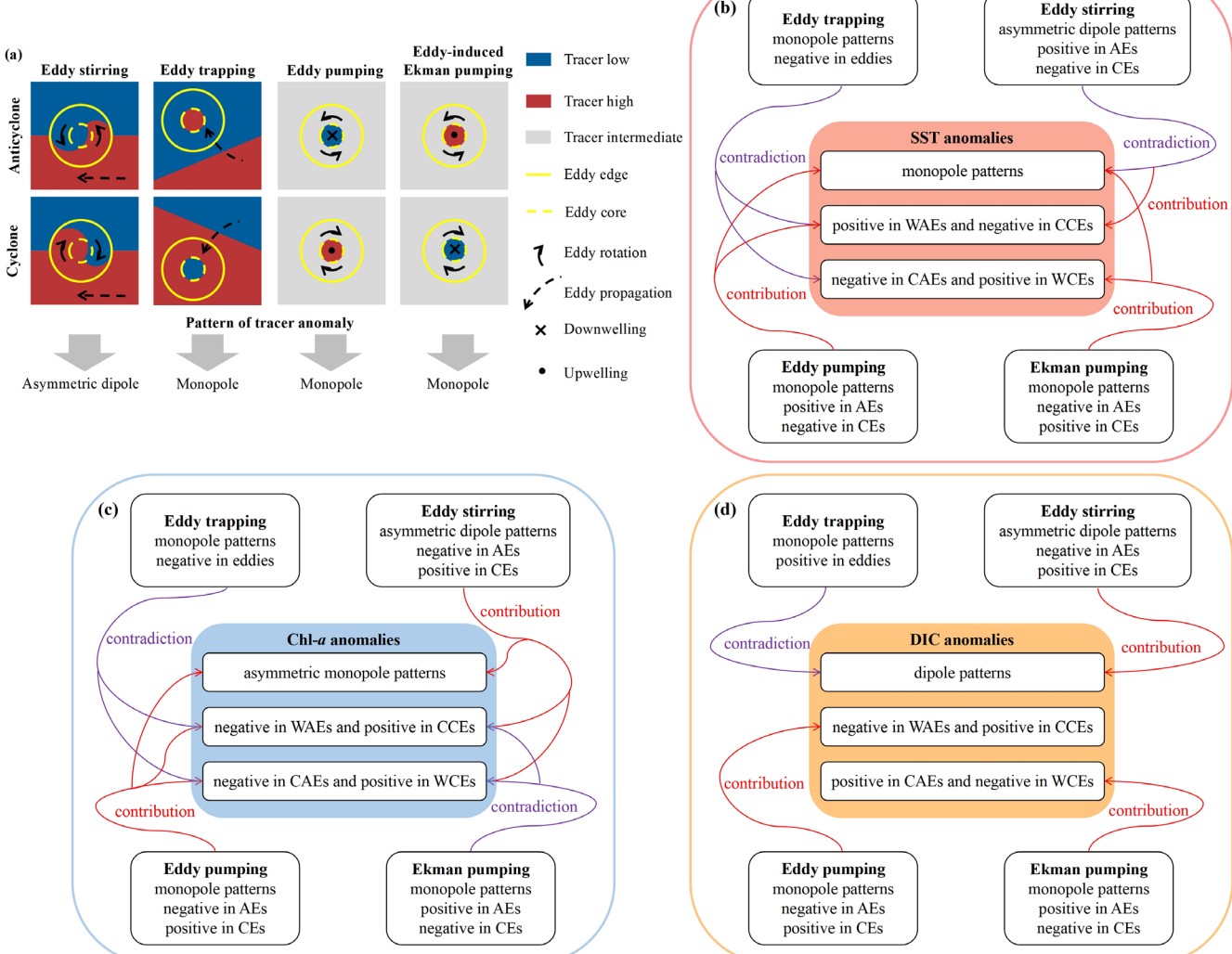

**Figure 7.** (a) Schematic illustrating the mechanisms of how eddies affect physical and biogeochemical parameters in the SO, including eddy stirring, eddy trapping, eddy pumping, and eddy-induced Ekman pumping. The patterns of SST anomalies induced by vertical pumping are opposite to the corresponding patterns shown in this schematic. The figure is inspired by Frenger et al. (2018), Fig. 1. Schematic diagram of the eddy mechanisms influencing (b) SST, (c) Chl-*a*, and (d) DIC anomalies.

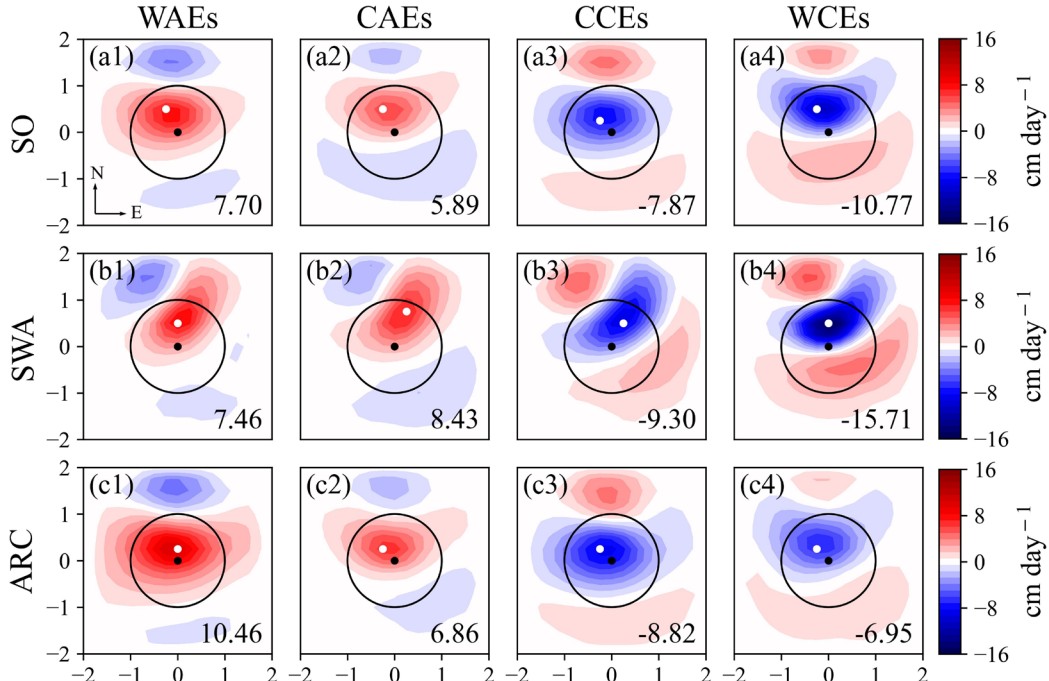

**Figure 8.** Eddy-centric composite averages for eddy-induced Ekman pumping in the SO, SWA, and ARC. On each map, a black dot denotes the eddy center, and a white dot denotes the center location of variables (defined by the location of the extremum value). Contour intervals are 1.067 cm day$^{-1}$. The numbers in the lower right corner are the extremum value.

# Winter

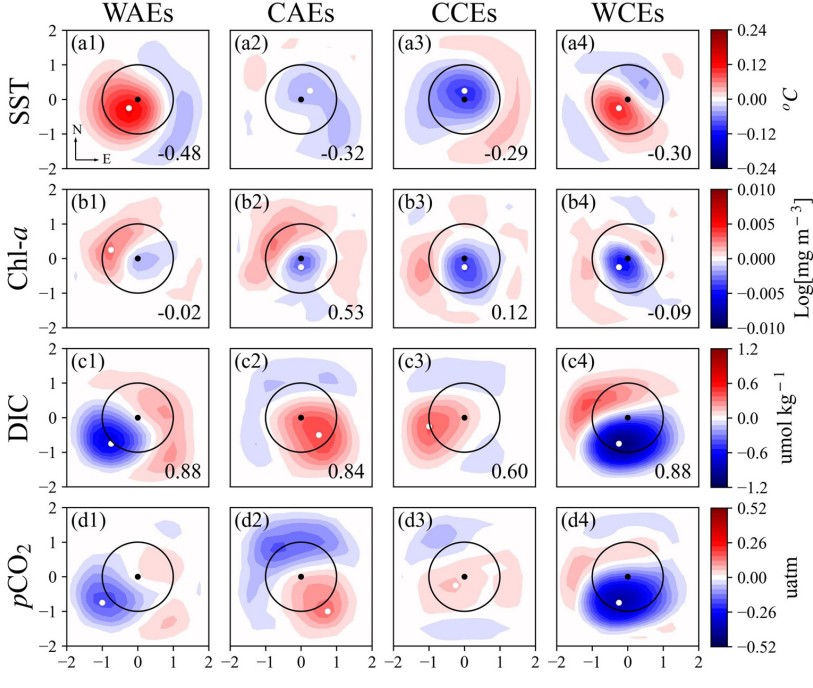

# Summer

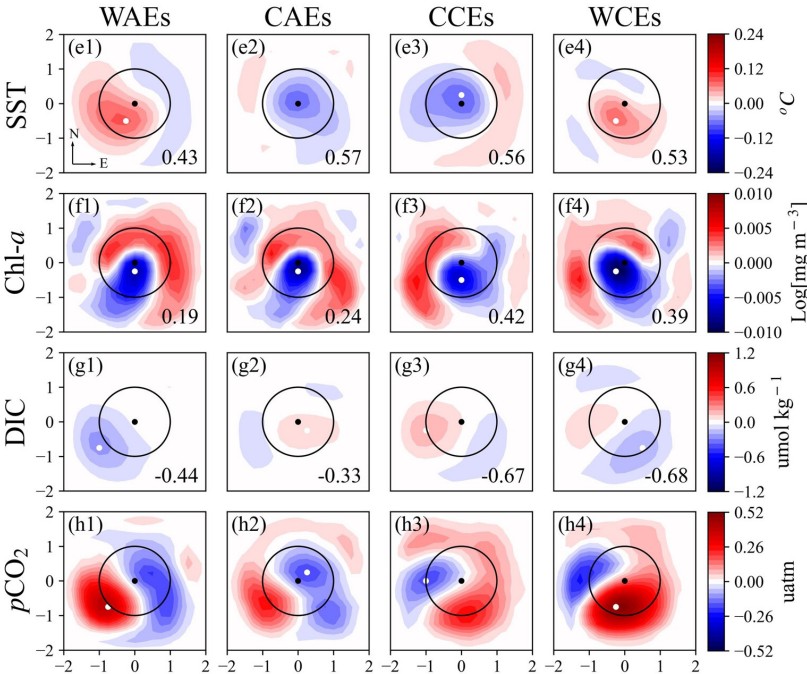

**Figure 9.** Same as Fig. 6 but for the SWA.

**Figure 10.** Same as Fig. 6 but for the ARC. And contour intervals are every 0.133 umol kg$^{-1}$ for DIC and every 0.053 uatm for $p$CO$_2$.

**Table 1.** Significance magnitudes of effects for eddy-driven mechanisms on SST, Chl-*a*, and DIC. A indicates a dominant effect. B represents an effect that contributes to the eddy-induced anomalies but is not the dominant effect. C denotes an effect that is not significant.

| | SST | | Chl-*a* | | DIC | |
|---|---|---|---|---|---|---|
| | "Normal" | "Abnormal" | "Normal" | "Abnormal" | "Normal" | "Abnormal" |
| Eddy trapping | C | C | C | C | C | C |
| Eddy stirring | B | C | A | A | A | A |
| Eddy pumping | A | B | A | A | A | B |
| Eddy-induced Ekman pumping | B | A | B | B | B | A |