# Peer review of "Characteristics of Surface Physical and Biogeochemical Parameters within Mesoscale Eddies in the Southern Ocean"

_Biogeosciences, 2023_

## Author Comment (AC1)

**Response to Reviewer #1**

**Overall impression**

The paper investigates the physical and biogeochemical characteristics of mesoscale eddies at the surface of the Southern Ocean – a region of global importance for heat and carbon exchange and biogeochemical cycles, concurrently a region dominated by eddies. This study involves many novel aspects compared to previous studies and tackles relevant topics for the community. It distinguishes between warm-core anticyclonic eddies (AEs), cold-core AEs, cold-core cyclonic eddies (CEs), and warm-core CEs (termed here as 'normal' and 'abnormal' AEs and CEs). At the same time, the discussion lacks some depth and many aspects of the methods and results/discussion remain unclear. There are also many figures which are only discussed very briefly and could be moved to the Supporting Information. These issues should be addressed before publication.

Response: We would like to thank reviewer 1 for the professional comments and valuable suggestions to improve the manuscript. We hope the answers and information presented here would respond to what was demanded.

**General comments**

1.   From the introduction, it's not entirely clear what's new about this study (one finds the information eventually, but it's quite hidden and only becomes apparent later). What's new about this study compared to previous work should really be the focus of the introduction. E.g., L86: Mention that the Frenger et al. studies investigated the vertical structure, while this study is only considering the surface. The same paragraph (from L86) also reads as if the only difference between the Frenger studies and this study is that this study differentiates between 'normal' and 'abnormal' eddies. However, there are many other differences (surface vs. interior; which parameters are considered; method of eddy detection…).

Response: We appreciate the reviewer's comments regarding the clarity of the introduction and the need to highlight the novelty of our study better. We carefully reviewed the introduction and added a paragraph that explicitly states the key differences between our work and previous studies. The added paragraph reads as follows:

"The role of eddies in modulating surface physical and biogeochemical parameters in the SO is still unclear. Previous studies only focused on the basin-wide impact of eddies on Chl-*a* (Frenger et al., 2015; Dawson et al., 2018; Frenger et al., 2018). To our knowledge, no research yet investigates the basin-scale effects of SO eddies on DIC

and $pCO_2$. Given the potential interactions between different physical and biogeochemical parameters and the importance of the SO in global climate change, biological productivity, and carbon cycling, it is necessary to systematically study the influence of eddies on SST, Chl-$a$, DIC, and $pCO_2$ in the SO. Most importantly, previous research did not distinguish between normal and "abnormal" eddies (Frenger et al., 2015; Dawson et al., 2018; Frenger et al., 2018). "Abnormal" eddies have SST anomalies opposite to normal eddies, which can potentially affect the biogeochemical parameters within eddies. Therefore, when investigating the regulation of air-sea variables induced by mesoscale eddies, it is important to consider the role of "abnormal" eddies, as this can lead to a more accurate estimation of mesoscale eddies' overall impact. In addition, previous work used traditional eddy detection methods based on satellite sea surface height (SSH) data (Chelton et al., 2011; Faghmous et al., 2015). In contrast, the eddy dataset we used is developed by a deep learning (DL) model based on the fusion of SSH and SST data (Liu et al., 2021), which can simultaneously detect eddy locations and distinguish between normal and "abnormal" eddies with great accuracy and efficiency."

2. My biggest concern: McGillicuddy, Gaube, and others have pointed out that eddy-induced Ekman pumping results in the opposite signal compared to regular eddy pumping and that eddy-induced Ekman pumping is usually weaker, but can be significant, especially in regions with large wind stress. Thus, when seeing cold core AEs or warm core CEs, I would assume that there, eddy-induced Ekman pumping dominates. However, in this paper it is framed like a mystery that some AEs have cold cores, and some CEs have warm cores (termed 'abnormal eddies'). Later, the study exactly finds this, at least in the analysis with SST (Section 5.1, esp. L278). Thus, I recommend rephrasing the storyline that cold-core AEs and warm-core CEs are likely to be dominated by eddy-induced Ekman pumping.

Response: We agree and we added a paragraph when mentioning abnormal eddies in the introduction to explain why we need to discuss how the abnormal eddies. The added paragraph reads as follows:

" "Abnormal" eddies may be induced by eddy-induced Ekman pumping (Gaube et al., 2013; Mcgillicuddy, 2015), instability during the eddy decay stage, eddy horizontal entrainment (Sun et al., 2019), and warm/cold background water (Leyba et al., 2017). However, there is still a gap regarding the cause of "abnormal" eddies in the SO."

3. The discussion should be deepened. Currently, Sections 4 and 5 mostly show the results with a lot of figures, and Section 6 (Conclusions) is mostly a summary of these findings. I'm missing a more in-depth discussion of what this now all means and how it matters. One thing to focus on especially is the surprising finding that when considering SST anomalies, eddy-induced Ekman pumping dominates in certain eddies, while for the other variables, different processes dominate. How can the same eddies pump DIC-rich water upwards without pumping cold water up? Once the discussion

has been deepened, the abstract and conclusions can then also mention some more from the discussion. Right now, the abstract and conclusion sections are quite descriptive of the results but don't tell us much about their significance.

Response: We deepened the discussion and revised the manuscript. To address the question that the influence mechanisms of eddies on SST, Chl-*a*, DIC, and $p\text{CO}_2$ are different, we calculated the average gradient of these variables in the SO from 1996 to 2015. The gradients of SST, Chl-*a*, DIC, and $p\text{CO}_2$ are 0.05, 0.11, 0.20, and 0.31, which are normalized prior to calculation. As eddy stirring redistribute physical and biogeochemical parameters spatially through horizontal advection, the larger the horizontal parameter gradient, the stronger the eddy stirring effect (Mcgillicuddy, 2016). The gradient of DIC is four times higher than that of SST, which indicates that eddy stirring will have a stronger effect on DIC than on SST. Thus, the composite DIC anomalies within eddies show dipole patterns, whereas the composite SST anomalies within eddies show monopole patterns. In addition to the different effects of eddy stirring on SST and DIC, both eddy pumping and eddy-induced Ekman pumping have an effect on SST and DIC. For normal eddies, eddy pumping dominates the vertical distribution of variables. Within CCEs, the upwelling with cold, DIC-rich deep water induces negative SST anomalies and positive DIC anomalies, and the reverse is true for WAEs. However, for abnormal eddies, eddy-induced Ekman pumping dominates the vertical distribution of variables. Within WCEs, the downwelling of warm, low-DIC surface waters induces positive SST anomalies and negative DIC anomalies, and the reverse is true for CAEs. Therefore, the influence mechanism of eddies on different variables is not universal, and it varies depending on the inherent properties of each variable and the complex interactions between them and the eddies.

4. Consider moving Fig.1 and 3 and Table 1 and 2 to the Supporting Information, they don't add much new information. Fig. 5 is only discussed with one sentence (L192) and could also move to the SI or be discussed in more depth. Similarly, Fig. 7 is only very briefly touched upon and can move to the SI.

Response: Figs. 1, 3, and 5 and Tables 1 and 2 have been moved to the Supporting Information. We added the description and discussion related to Fig. 7 so that Fig. 7 is not moved.

**Specific comments**

1. It is not immediately clear that the study only focuses on surface properties. This could be added to the title and should be clearer in the abstract and introduction.

Response: We have revised the title, abstract, and introduction to make it clearer that our study solely examines surface properties.

2. L10: add 'horizontal surface' before 'composite'

Response: "horizontal surface" has been added.

3. L73: Are the signals also different when the seasonal signal has been removed? I.e., are the anomalies computed based on mean annual reference values, or on a monthly climatology? I have a feeling that if a monthly climatology is used as a reference, the eddy anomalies might not differ so much anymore by season. This should be mentioned/discussed.

Response: We have modified the method section and added a figure (as shown below) in the Supporting Information to show that we have removed the seasonal signals. The SST and Chl-$a$ anomalies are computed using a 7-90 days band-pass filter to remove the seasonal signal. For DIC and $p$CO$_2$ datasets with the monthly temporal resolution, we subtracted their climatology averages. As shown in Fig. 1 below, there are no significant seasonal variations in eddy-induced SST, Chl-$a$, and DIC anomalies. By contrast, $p$CO$_2$ anomalies were remarkably different between the two seasons, caused by the different dominant effects of SST, Chl-$a$, and DIC, as we discuss in the manuscript.

[Figure]

Figure 1. Variations in monthly mean eddy-induced anomalies, including (a) SST, (b) Chl-*a*, (c) DIC, and (d) *p*CO2 in the SO from 1996 to 2015. Solid lines in different colors denote four kinds of eddies.

4. L119: Be specific that the DIC is gap-filled in this step.

Response: We have explicitly mentioned in the manuscript that the DIC field is gap-filled during this process.

5. Section 2.2 and 2.3: be clearer that those datasets used were created by previous studies. It currently reads ambiguously if this was done during this study or if the data is from previous work.

Response: We have revised the manuscript to make it clear that these datasets were created by previous studies.

6. L127: Is 'This dataset' referring to Landschuezter et al. 2014, or to the Liu et al.

2021 product that is used in this study? If it's referring to Landschuezter: as that product has been used so widely, why did you not use that product? What's the benefit of using Liu et al? If it's referring to Liu et al: rephrase the sentence so that it's clearer (but then some of the references are wrong as they were published before 2021…)

Response: We appreciate the reviewer's comment and apologize for the confusion caused by the ambiguous reference in the manuscript. To clarify, in Line 127, "This dataset" refers to the dataset used in our study, which is from the JMA Ocean $CO_2$ Map dataset, established by Iida et al. (2021). The sentence in the manuscript is as follows: " The $p$CO$_2$ and DIC datasets are from the JMA Ocean $CO_2$ Map dataset with monthly $1° × 1°$ gridded values on the global ocean from 1990 to 2020 (Iida et al., 2021)."

Besides, the reason for citing references published before 2021 is that the initial version of this database was published in 2015 (Iida et al., 2015). These two versions of the $p$CO$_2$ dataset both use the same approach, multiple linear regression (MLR) method. The difference is that the initial one uses sea SST, sea surface salinity (SSS), and Chl-$a$ as independent variables (Iida et al., 2015). By contrast, the new version of the $p$CO$_2$ dataset is reconstructed from the fields of total alkalinity (TA), DIC, SST, and SSS (Iida et al., 2021).

To address the issue of the data source and references, we have emphasized that the dataset is provided by Iida et al. (2021) and cited accurate references related to this dataset.

7. Section 2.3: Mention how the eddy detection method differs from other, more commonly used approaches, such as the AVISO eddy database (newest version: Pegliasco et al. 2021), and why it was preferred. One could have used the AVISO eddies and classified the eddies into normal and abnormal based on their SST signature (e.g., AE with cold SST anomaly is CAE...).

Response: We have added a paragraph in Section 2.3 to highlight the differences between our eddy detection method and the AVISO eddy database, as well as why we choose this eddy dataset. Compared to the AVISO eddy database (Pegliasco et al., 2022), our study utilizes a different eddy detection method (Liu et al., 2021), which uses a deep learning model to fuse satellite SSH and SST data. The reason why we use this method is that deep learning technology has unparalleled learning ability and the capability to model complex nonlinear relationships compared to traditional statistics and machine learning methods (Reichstein et al., 2019). Besides, the method can simultaneously extract SSH features for determining eddy locations and extract SST information to help distinguish between normal and abnormal eddies. As a result, our method achieves great accuracy and much higher efficiency than the traditional method that first detects the eddies and then uses the SST signature to classify them into normal and abnormal eddies. In addition, the method is able to detect eddies in regions where traditional methods may not be effective, such as in regions with weak eddies or regions

with complex oceanic dynamics (Liu et al., 2021). Given its high accuracy and comprehensive information on eddy characteristics, we find this dataset particularly useful for our study.

8.    L144-149: I think this part of the paragraph still belongs to section 2.3.

Response: Thanks for the feedback. This paragraph describes the methodology for obtaining the distribution of various sea surface variables within the eddy, so it should be included in the methodology section 3.1 rather than the data section 2.3. To address this concern, we revised the methodology section title to "Composite Eddy-induced Anomalies" and enhanced the methodology section to provide a more explicit explanation of this point in the paper.

9.    L149-153: Needs a more in-depth description of how the composite eddies were made.

Response: We have added a schematic in the Supporting Information (as shown in Fig. 2 below) and revised the section to provide additional details on the methodology used to create the composite eddies. The revised section reads as follows:

"Finally, we use the eddy-centric composite method to estimate the spatial pattern of the eddy-induced anomalies in sea surface variables. The positions of co-located SST, Chl-$a$, DIC, and $p$CO$_2$ observations are normalized by R, which defines the edge of an eddy as ±1 and the eddy core as 0. This allowed us to construct composite averages from eddies of varying sizes. We then extract data from −2R to 2R to include the interactions between eddies and the surrounding waters and interpolate them onto an evenly spaced 17 by 17 grid to create the surface composite patterns. For daily SST and Chl-$a$, we perform the eddy-centric composite method matching eddies and variables on the same day and calculate the mean value. By contrast, for monthly DIC and $p$CO$_2$, we calculate the eddy-centric composite maps, using all eddies of the same month with DIC and $p$CO$_2$ of that month and calculate the mean value. The composites are not rotated with the background variables gradient, as the large-scale background variables gradients in the SO are oriented north-south. Previous studies have shown that rotating eddies to the large-scale variables gradient in the SO has a negligible impact on the results (Frenger et al., 2015). Therefore, the axes in each figure point north and east."

[Figure]

Figure 2. Schematic of eddy-centric composite method for daily (a) SST and Chl-*a* and monthly (b) DIC and *p*CO$_2$, taking January as an example.

10. L171: Be explicit about how it differs from the method by Gaube et al. 2015.

Response: We have revised the section to clarify that we use the same formula to calculate the total eddy-induced Ekman pumping and the same spatiotemporal filtering method as Gaube et al. (2015). However, we only calculate the total eddy-induced Ekman pumping and do not calculate the individual components such as SST-induced Ekman pumping and current-induced Ekman pumping. Besides, the SSH dataset we use is constructed at a daily temporal resolution, whereas the SSH data used by Gaube et al. (2015) is constructed at 7-day intervals.

11. Section 3.2: Add a reference for the methods to obtain the eddy-induced Ekman pumping.

Response: A reference has been added.

12. L239: Discuss why we want to know how the pattern differs from the *p*CO$_2$ pattern. I would have found it more interesting to see the pattern differences between normal and abnormal eddies, but there could be a reason why you chose this.

Response: We revised this section to explain why we discuss the pattern differences

between $pCO_2$ and other variables. While SST, Chl-$a$, and DIC anomalies within eddies are found to be similar in summer and winter, $pCO_2$ anomalies are significantly different between the two seasons. This is because, in winter, the $pCO_2$ anomalies are dominated by the dominant DIC-driven effect. By contrast, in summer, the $pCO_2$ anomalies are dominated by SST and DIC anomalies in some regions with smaller and larger magnitudes of DIC anomalies, respectively. Due to the opposite anomalies of DIC and SST within the same kind of eddies, the basin-scale effects of SO normal and abnormal eddies on $pCO_2$ have little significant pattern differences in summer. Since we already discussed SST and DIC pattern differences between normal and abnormal eddies, understanding the relationship between $pCO_2$ and other anomalies within the eddies can explain the differences in $pCO_2$ patterns within normal and abnormal eddies.

13. L255: Mention why stirring is not a process (we can see it in the plot, but it needs to be discussed).

Response: In our study, we propose that the meridional and zonal phase shifts in normal eddies are induced by the large-scale background SST gradient and eddy stirring. However, the SST anomalies within abnormal eddies show purely monopoly patterns, which do not reflect the stirring impact. In contrast, processes such as eddy pumping and eddy-induced Ekman pumping have a more significant impact on SST anomalies. Therefore, we do not regard it as a major process regulating the SST anomalies in eddies.

14. Generally: Personally, I would not use the terms 'normal' and 'abnormal', as everything is normal and within the expected physics (when considering eddy-induced Ekman pumping), but this may be a personal choice. Maybe 'regular' and 'unusual' fits better, as warm-core AEs and cold-core CEs are a lot more common than cold-core AEs and warm-core CEs, but I'm nit-picking now.

Response: We understand that everything can be considered normal within the expected physics, therefore, we have chosen to use the term "abnormal" in quotes to indicate that it is a relative term and not an absolute one, and we are referring specifically to departures from the expected behavior. The terms "normal" and "abnormal" are commonly used in the scientific community to describe expected and unexpected phenomena, and using them consistently throughout the paper helps maintain clarity and coherence.

15. Fig. 2: Consider using a sequential colormap. Specify the latitude where the white region starts (65S?). Most of the currents and topographic features are not referred to in the text and can be removed.

Response: A sequential colormap has been used. The currents and topographic features that are not mentioned in the manuscript have been removed.

16. Fig. 4 (and the following figures): Add in the caption what the magenta boxes are.

Response: Captions have been added to all relevant figures.

17. Fig. 6: Why are there some warm spots in cold eddies, and cold spots in warm eddies? By definition, the SST anomalies should be cold in cold eddies, and warm in warm eddies.

Response: Thanks for the comment. We apologize for the confusion caused by our carelessness in using outdated SST anomaly data for CAEs and WCEs in Fig. 6. We have revised the figure by using the correct data. We assure that all other results involving SST anomaly data within eddies are based on the correct data.

18. Fig. 8: Ensure all SSIMs have the same number of decimals.

Response: All SSIMs have been corrected using the same number of decimals.

**Technical corrections**

1. Throughout the document: change biochemical to biogeochemical.

Response: The word biochemical has been changed to biogeochemical.

2. It's a good habit to discuss the findings of this paper in the present tense and refer to previous studies in the past tense. E.g., L9: change to 'we analyze' (instead of 'we analyzed'); same throughout the whole document.

Response: Thanks for pointing it out. We have checked the entire document and made corrections accordingly.

3. L12. I know many studies do this and it is a personal choice, but I dislike sentences with brackets for multiple things. Consider writing it out for each, e.g., 'dominated by DIC anomalies in regions with larger magnitudes of DIC anomalies and dominated by SST anomalies in regions with smaller magnitudes'. Same throughout the whole document.

Response: We have made corrections accordingly.

4. L22: existing (not exiting)

Response: The word exiting has been changed to existing.

5. The font in some figures is very large.

Response: The font in some figures has been reduced appropriately.

**Reference**

Frenger, I., Münnich, M., Gruber, N., and Knutti, R.: Southern Ocean eddy phenomenology, J. Geophys. Res.-Oceans, 120, 7413-7449, https://doi.org/10.1002/2015jc011047, 2015.

Gaube, P., Chelton, D. B., Strutton, P. G., and Behrenfeld, M. J.: Satellite observations of chlorophyll, phytoplankton biomass, and Ekman pumping in nonlinear mesoscale eddies, J. Geophys. Res.-Oceans, 118, 6349-6370, https://doi.org/10.1002/2013JC009027, 2013.

Gaube, P., Chelton, D. B., Samelson, R. M., Schlax, M. G., and O'Neill, L. W.: Satellite Observations of Mesoscale Eddy-Induced Ekman Pumping, J. Phys. Oceanogr., 45, 104-132, https://doi.org/10.1175/jpo-d-14-0032.1, 2015.

Iida, Y., Takatani, Y., Kojima, A., and Ishii, M.: Global trends of ocean $CO_2$ sink and ocean acidification: an observation-based reconstruction of surface ocean inorganic carbon variables, J. Oceanogr., 77, 323-358, https://doi.org/10.1007/s10872-020-00571-5, 2021.

Leyba, I. M., Saraceno, M., and Solman, S. A.: Air-sea heat fluxes associated to mesoscale eddies in the Southwestern Atlantic Ocean and their dependence on different regional conditions, Clim. Dyn., 49, 2491-2501, https://doi.org/10.1007/s00382-016-3460-5, 2017.

Liu, Y., Zheng, Q., and Li, X.: Characteristics of Global Ocean Abnormal Mesoscale Eddies Derived From the Fusion of Sea Surface Height and Temperature Data by Deep Learning, Geophys. Res. Lett., 48, https://doi.org/10.1029/2021gl094772, 2021.

McGillicuddy, D. J.: Formation of Intrathermocline Lenses by Eddy–Wind Interaction, J. Phys. Oceanogr., 45, 606-612, https://doi.org/10.1175/jpo-d-14-0221.1, 2015.

McGillicuddy, D. J.: Mechanisms of Physical-Biological-Biogeochemical Interaction at the Oceanic Mesoscale, Annu. Rev. Mar. Science, 8, 125-159, https://doi.org/10.1146/annurev-marine-010814-015606, 2016.

Pegliasco, C., Delepoulle, A., Mason, E., Morrow, R., Faugère, Y., and Dibarboure, G.: META3.1exp: a new global mesoscale eddy trajectory atlas derived from altimetry, Earth Syst. Sci. Data, 14, 1087-1107, https://doi.org/10.5194/essd-14-1087-2022, 2022.

Reichstein, M., Camps-Valls, G., Stevens, B., Jung, M., Denzler, J., Carvalhais, N., and Prabhat: Deep learning and process understanding for data-driven Earth system science, Nature, 566, 195-204, https://doi.org/10.1038/s41586-019-0912-1, 2019.

Sun, W., Dong, C., Tan, W., and He, Y.: Statistical Characteristics of Cyclonic Warm-Core Eddies and Anticyclonic Cold-Core Eddies in the North Pacific Based on Remote Sensing Data, Remote Sens., 11, 208, 2019.

---

## Author Comment (AC2)

**Response to Reviewer #2**

**Overall impression**

The manuscript analyzed the biochemical influences of mesoscale eddies (including normal and abnormal eddies) in the Southern Ocean, by using machine learning and multi-source marine dataset. The manuscript estimated chlorophyll (Chl) and dissolved inorganic carbon (DIC) contributions to $p$CO$_2$, and found their seasonal variations. These results are interesting and are of vital importance on global biogeochemical cycles and the climate change. However, some description about methods/data are ambiguous and few conclusions need to be further discussed.

Response: We would like to thank reviewer 2 for taking the time to review the manuscript and for its valuable feedback. We acknowledge that the suggestions provided have really helped to improve the quality of this work. We hope the answers and information provided here would respond to what was demanded.

**Major questions:**

1.  There are some other methods to identify abnormal eddies, such as using potential density and directions of geostrophic current. They are supposed to be introduced in the introduction, and point out why authors choose the method of SSTA.

Response: We acknowledge that there are other methods to identify abnormal eddies, such as using potential density and geostrophic current direction (Mcgillicuddy, 2015). We have added this information in the introduction. We also clarified why we chose to use SSTA to distinguish between normal and abnormal eddies. Recent studies have found that abnormal eddies show opposite SSTA signals to normal eddies (Leyba et al., 2017; Liu et al., 2020; Liu et al., 2021; Ni et al., 2021). Compared to potential density, SSTA data can be obtained from satellite remote sensing with higher spatial and temporal resolutions, making it a convenient and reliable data source for identifying eddies (Castellani, 2006; Liu et al., 2021).

Moreover, detecting eddies using directions of geostrophic current is essentially based on SSH features. Our study utilizes the abnormal eddy dataset that Liu et al. (2021) developed, which uses a deep learning model to fuse satellite SSH and SST data. The method can simultaneously extract SSH features for determining eddy locations and extract SST information to help distinguish between normal and abnormal eddies with great accuracy and efficiency. In addition, the method is able to detect eddies in regions where traditional methods may not be effective, such as in regions with weak eddies or regions with complex oceanic dynamics (Liu et al., 2021). Given its high accuracy and comprehensive information on eddy characteristics, we find this dataset to be particularly useful for our study.

2. The methods to derive $pCO_2$ from Chl, SST, DIC and other variables are supposed to be introduced with more descriptions or equations.

Response: We have added the following details to explicitly describe the methods to derive $pCO_2$:

" The $pCO_2$ field is calculated from TA, DIC, SST, and SSS based on seawater $CO_2$ chemistry (Iida et al., 2021). Firstly, the mean rates of regional $pCO_2$ and multiple regressions are used to derive the algorithms of $pCO_2$ expressed empirically as a function of in situ TA, DIC, SST, SSS, and the year. Then, the $pCO_2$ fields that filled both in space (1° × 1°) and time (monthly) are drawn by applying global data sets of TA, DIC, SST, and SSS to the variables in these empirical equations."

3. The method to define and identify abnormal eddies should be introduced in detail even if the authors cited the paper of Liu et al., 2021. Did they identify abnormal eddies according to SSTA>0/SSTA<0 within eddy boundaries/cores? How did they distinguish AEs and CEs just according to SSTA?

Response: We have updated the manuscript to include more detailed information on our methodology for identifying abnormal eddies. We distinguish between normal and abnormal eddies based on the mean SSTA within eddy boundaries. Besides, we distinguish between AEs and CEs based on the SSHA, as AEs (CEs) are usually accompanied by local convergence (divergence), leading to positive (negative) SSHA. Therefore, WAEs are identified according to SSHA >0 and SSTA >0, CAEs are identified according to SSHA >0 and SSTA <0, CCEs are identified according to SSHA <0 and SSTA <0, and WCEs are identified according to SSHA <0 and SSTA >0.

4. The descriptions about eddy dataset and identification are very poor. In line 139, the authors mentioned "the ground truth data set". What's the ground truth data set of eddies? Is it produced by the authors or a public dataset? That's important to the verification.

Response: We have verified in the manuscript that the ground truth dataset of mesoscale eddies used in our study was generated automatically using the SSH-based method proposed by Haller (2005), and the eddy dataset was produced by Liu et al. (2021).

5. How did authors match daily eddy dataset with monthly DIC and $pCO_2$ temporally and spatially when doing composite analyses? Temporally, is eddy at JAN. 31st matched with DIC of JAN. or Feb. data? is DIC data used within eddy boundaries or eddy cores?

Response: We have added a schematic in the Supporting Information and revised the section to provide additional details on the methodology used to create the composite

eddies. The positions of co-located SST, Chl-*a*, DIC, and $pCO_2$ observations are normalized by R, which defines the edge of an eddy as ±1 and the eddy core as 0. This allowed us to construct composite averages from eddies of varying sizes. We then extract data from −2R to 2R to include the interactions between eddies and the surrounding waters and interpolate them onto an evenly spaced 17 by 17 grid to create the surface composite patterns. Therefore, the mean anomalies of SST, Chl-*a*, DIC, and $pCO_2$ are used within eddy boundaries. For daily SST and Chl-*a*, we perform the eddy-centric composite method matching eddies and variables on the same day and calculate the mean value, as shown in the following Fig. 1a. By contrast, for monthly DIC and $pCO_2$, we calculate the eddy-centric composite maps, using all eddies of the same month with DIC and $pCO_2$ of that month and calculate the mean value (Fig. 1b below).

[Figure]

Figure 1. Schematic of eddy-centric composite method for daily (a) SST and Chl-*a* and monthly (b) DIC and $pCO_2$, taking January as an example.

6.   Taking CEs for example, commonly, upwellings are thought to transport cold water to the sea surface, as well as richer nutrients at the same time. Therefore, CEs often show lower SST and higher chlorophyll. The conclusions from Figure 8 show SSTA within abnormal eddies are dominant by Ekman pumping. However, the chlorophyll anomalies of abnormal eddies are attributed to eddy pumping. The conclusions are contradictory to each other. If they are reliable, what's the mechanism leading to contrasting vertical process on SST and chlorophyll respectively? Therefore, discussions of lines 279-280 and 295-296 need more explanations. Besides, line 279 should be "eddy-induced Ekman pumping".

Response: We have revised the manuscript accordingly and deepened the discussion. In response to the question, we further calculated the mean gradient of SST and Chl-*a*, which are normalized prior to calculation. The average gradients of SST and Chl-*a* are found to be 0.05 and 0.11, respectively. The north-south gradients (north is the positive direction) of SST and Chl-*a* are 0.04 and −0.02, respectively. The east-west gradients (east is the positive direction) of SST and Chl-*a* are 0.00 and −0.04, respectively. As eddy stirring redistribute physical and biogeochemical parameters spatially through horizontal advection, the larger the horizontal parameter gradient, the stronger the eddy stirring effect (Mcgillicuddy, 2016). The small gradient of SST leads to a negligible effect of eddy stirring. Within abnormal eddies, the effect of eddy-induced pumping overcomes the effect of eddy pumping, resulting in the opposite SST anomalies in normal and abnormal eddies.

Compared to SST, Chl-*a* has a higher gradient, resulting in a stronger effect of eddy stirring. The gradients of Chl-*a* suggest that the climatological Chl-*a* increases southward and westward. Counterclockwise rotation of AEs in the SO would advect low Chl-*a* from the northeast to the west and high Chl-*a* from the southwest to the east. The reverse is true for CEs. Previous works found that the dipole shapes arising from stirring tend to be asymmetric, with larger anomalies at the leading compared to the trailing side of eddies (Chelton et al., 2011; Frenger et al., 2015; Dawson et al., 2018; Frenger et al., 2018). As the major propagation direction of eddies is westward, the composite Chl-*a* anomalies in AEs/CEs show dominant negative/positive signals due to eddy stirring. Besides, eddy pumping tends to produce Chl-*a* anomalies of the same sign. The common effects of eddy stirring and eddy pumping overcome the effect of eddy-induced Ekman pumping, resulting in similar patterns of Chl-*a* anomalies in normal and abnormal eddies.

However, from Figs. 8b1–b4 and f1–f4 in the manuscript, we can see that the magnitudes of Chl-*a* anomalies within normal eddies are higher than abnormal eddies, which reflects the effect of eddy-induced Ekman pumping. Besides, in some regions with small amplitude, such as the south of ACC and the South Pacific Ocean, we find Chl-*a* anomalies in AEs/CEs are positive/negative (Figs. 6e–h in the manuscript). Such a result may be caused by a more dominant effect of eddy-induced Ekman pumping on Chl-*a*. Overall, eddy stirring and eddy pumping are mainly responsible for the patterns of Chl-*a* anomalies within eddies in the SO, and eddy-induced Ekman pumping attenuates the magnitudes of Chl-*a* anomalies within abnormal eddies.

7. The manuscript is supposed to evaluate the accuracies of abnormal eddy identification method, which can combine with Argo profiles via temperature and potential density. At the same time, it should point out the method improvement in future

Response: We have updated the manuscript to reflect these points. The experiments

showed that the model could accurately identify abnormal eddies in the South China Sea (SCS) and Kuroshio Extension (KE) region (Liu et al., 2021). In addition, Argo floats data also verified the accuracy and validity of the model (Liu et al., 2021). However, we also acknowledge that there is room for improvement in our method. Considering that the changes in SSH, SST, Chl-*a*, and roughness caused by eddies can be recorded by altimeter, infrared, ocean color, and synthetic aperture radar (SAR) remote sensing, respectively. Besides, potential density and temperature recorded by Argo floats can also identify abnormal eddies. In future work, we will combine multiple remote sensing data with Argo profiles to evaluate the accuracies of abnormal eddy identification method.

**Minor questions:**

1. Lines 31-32: Authors point out that eddies have influences on "biochemical parameters". While, the listed references are both about chlorophyll, which is a biological parameter. References about chemical parameters should be introduced.

Response: References about chemical parameters have been added.

2. Lines 37-39: Rotations of eddies are related to the hemisphere. It should illustrate which hemisphere is talked about.

Response: When mentioning the rotations of the eddies, we emphasized that the eddies is in the Southern Hemisphere.

3. Line 56: How about eddy influence on chlorophyll during wintertime with deeper mixing?

Response: We have added information and references about the influence of eddies on Chl-*a* during wintertime with deeper mixing in the introduction. Dufois et al. (2014) suggested that deeper mixed layers could explain long-lived Chl-*a* anomalies in anticyclones of the South Indian Ocean between 20°S and 30°S. Both mixing and eddy-induced Ekman pumping tend to produce Chl-*a* anomalies of the same sign. For instance, shallower mixed layers in cyclonic eddies could result in higher Chl-*a*, while deeper mixed layers in anticyclonic eddies could lead to lower Chl-*a*.

In our study, eddy stirring and eddy pumping are the main modulation processes of normal and abnormal eddies to Chl-*a* in the SO. Composite Chl-*a* anomalies display negative signatures in both WAEs and CAEs and positive signatures in CCEs and WCEs. Therefore, we do not specifically address the influence of eddies on Chl-*a* during wintertime with deeper mixing in our study.

4. Lines 107-108: The expression of OI-SST should be in agreement.

Response: We have corrected the expression of OI-SST.

5.    Line 166: JMA is suggested to be introduced as Japan Meteorological Agency.

Response: We have introduced JMA as "Japan Meteorological Agency" upon its first mention in the manuscript.

6.    Line 178: What are the denominators when calculating eddy frequencies? It should be expressed more clearly.

Response: We have added the definition of eddy frequency. The eddy frequency is the ratio of the number of days eddies appeared to the total number of observation days.

7.    Lines 186-187: The conclusion is true in South America, but not evident in the south of Australia.

Response: We have revised the manuscript to demonstrate the findings explicitly. Based on Figs. 4c and 4f in the manuscript, it can be observed that abnormal eddies have a polarity distribution opposite to that of normal eddies in the continental boundary currents where more CCEs and CAEs occur. However, it should be noted that more WAEs and WCEs occur in the south of Australia.

8.    Line 285: How to understand "eddy trapping has little influence on Chl-*a*"? Please give more descriptions to explain it.

Response: We have added more descriptions to explain why eddy trapping has little influence on Chl-*a*. Nonlinear eddies tend to trap the fluid contained in their interiors (Provenzale, 1999; Mcgillicuddy, 2016). The composition of the trapped fluid is dependent on various factors, including the eddy propagation and the local gradients in physical and biochemical properties. The tracks of long-lived eddies with lifetimes longer than 1 year show that the major propagation direction of eddies is westward, with AEs propagating north and CEs propagating south. Due to the climatological Chl-*a* increasing southward, AEs propagating northward tend to trap high Chl-*a* into northern areas with low Chl-*a*, as shown in Fig. 2 below. On the other hand, CEs propagating southward tend to trap low Chl-*a* into southern areas with high Chl-*a*. Such effect of eddy trapping on Chl-*a* contradicts the actual composite Chl-*a* anomalies over eddies with negative Chl-*a* anomalies in AEs and positive Chl-*a* anomalies in CEs. As a result, we conclude that eddy trapping has little influence on Chl-*a*.

[Figure]

Figure 2. Schematic illustrating the eddy trapping of how AEs and CEs affect Chl-*a*. Red and blue colors represent high and low Chl-*a*, respectively.

9. Lines 303-314: Are those conclusions for summertime still "dominant"? The magnitudes seem similar for summertime.

Response: Yes, there are no significant seasonal variations in eddy-induced SST, Chl-*a*, and DIC anomalies. Therefore, the dominant mechanisms of eddies affecting these variables do not alter by season.

10. Figure 11 is suggested to be shown in wintertime and summer time respectively, based on which Figure 8, Figure 12, and Figure 13 can be better discussed.

Response: We appreciate the suggestion, but given the nature of our findings, we feel that presenting the annual mean is more appropriate for our study. In the original manuscript, we did not discuss the SST, Chl-*a*, and DIC anomalies within the eddies seasonally, as their variations did not exhibit significant seasonal patterns. Therefore, we opted to present the annual mean eddy-induced Ekman pumping in Figure 11, as it provides a comprehensive representation of the differences in variable anomalies between normal and abnormal eddies. We believe that this approach allows for a more convenient and meaningful comparison of the variable anomalies within the eddies.

11. In Figure 4, lines 558-599, the authors mean blue and red colors in the right column. However, blue and red colors are shown in each sub-figure, which is misleading.

Response: We have revised the manuscript to ensure that the blue and red colors are clearly associated with the right column only.

12. Figures 4d and 4e show that abnormal eddies occur along fronts, where eddies are active, and along offshore areas where accuracies of altimeters are low. It is suggested to show ratios of abnormal eddies to normal eddies (WAEs/CAEs, CCEs/WCEs). Will the abnormal eddy signals offshore be amplified offshore? What are the mean depths of clustered abnormal eddies? It should be cautious with eddies shallower than 1000 m.

Response: We appreciate the reviewer's insightful suggestions, and we calculated the ratios of abnormal eddies to normal eddies, as shown in Fig. 3 below. We find more CAEs in the Western Boundary Current (WBC) regions and significant dominance of WCEs in southern Australia. In the southeast of America and Campbell Plateau, with depths shallower than 1000 m (Fig. 4 below), abnormal eddy signals offshore may be amplified offshore due to the low accuracies of altimeters along offshore areas. We further calculate the mean depths of clustered eddies. The mean depths of WAEs, CAEs, CCEs, and WCEs are 4086 m, 3969 m, 4044 m, and 4014 m. All of them are deeper than 1000m. As mentioned in the manuscript, eddies disappear in regions shallower than 2000m because the bottom topography constrains the generation of eddies. Therefore, the amplified abnormal eddy signals in the southeast of America and Campbell Plateau have little influence on the results.

[Figure]

Figure 3. Spatial distribution of eddy polarity dominance in the SO from 1996 to 2015. (a) Ratio of the area occupied by WAEs over the area covered by CAEs. (b) Ratio of the area occupied by CCEs over WCEs. Values >0 in red and <0 in blue mark the dominance of normal and abnormal eddies, respectively. Black solid lines show the mean northern and southern positions of the ACC major fronts. The black dotted circle is 50° S. The magenta boxes represent ARC and SWA regions.

[Figure]

Figure 4. Southern Ocean topography and current. Black solid lines show the mean northern and southern positions of the ACC major fronts. The black dotted circle is 50° S.

13. Figure 6. The abnormal eddies are identified from SST so the SSTA of Figure 6 is regular. The other three parameters are very noisy. The magnitudes of chlorophyll and $pCO_2$ signals induced by abnormal eddies are even higher than normal eddies, which are contrasting with eddy amplitude comparisons. Why?

Response: We have added some explanations to the manuscript to address the reviewer's concern. The distributions of Chl-$a$ anomalies over both normal and abnormal eddies are similar to the eddy amplitude distributions, with stronger negative/positive anomalies within AEs/CEs in regions of higher amplitude. This result indicates the dominant effect of eddy pumping on Chl-$a$. However, in regions of lower amplitude, we find the patterns of Chl-$a$ anomalies are spotty, with average positive/negative Chl-$a$ anomalies in AEs/CEs. Such a result may be caused by a more dominant effect of eddy-induced Ekman pumping on Chl-$a$.

The magnitudes of Chl-$a$ anomalies induced by abnormal eddies are even higher than normal eddies in these regions due to the smaller amplitude and eddy pumping of abnormal eddies than normal eddies. Furthermore, in some regions, such as SWA, the magnitudes of $pCO_2$ anomalies induced by abnormal eddies are higher than normal eddies, which are related to the stronger eddy-induced Ekman pumping of abnormal eddies.

**Reference**

Castellani, M.: Identification of eddies from sea surface temperature maps with neural

networks, Int. J. Remote Sens., 27, 1601-1618, https://doi.org/10.1080/01431160500462170, 2006.

Chelton, D. B., Gaube, P., Schlax, M. G., Early, J. J., and Samelson, R. M.: The Influence of Nonlinear Mesoscale Eddies on Near-Surface Oceanic Chlorophyll, Science, 334, 328-332, https://doi.org/doi:10.1126/science.1208897, 2011.

Dawson, H. R. S., Strutton, P. G., and Gaube, P.: The Unusual Surface Chlorophyll Signatures of Southern Ocean Eddies, J. Geophys. Res.-Oceans, 123, 6053-6069, https://doi.org/10.1029/2017JC013628, 2018.

Dufois, F., Hardman-Mountford, N. J., Greenwood, J., Richardson, A. J., Feng, M., Herbette, S., and Matear, R.: Impact of eddies on surface chlorophyll in the South Indian Ocean, J. Geophys. Res.-Oceans, 119, 8061-8077, https://doi.org/https://doi.org/10.1002/2014JC010164, 2014.

Frenger, I., Münnich, M., and Gruber, N.: Imprint of Southern Ocean mesoscale eddies on chlorophyll, Biogeosciences, 15, 4781-4798, https://doi.org/10.5194/bg-15-4781-2018, 2018.

Frenger, I., Münnich, M., Gruber, N., and Knutti, R.: Southern Ocean eddy phenomenology, J. Geophys. Res.-Oceans, 120, 7413-7449, https://doi.org/10.1002/2015jc011047, 2015.

Haller, G.: An objective definition of a vortex, J. Fluid Mech., 525, 1-26, https://doi.org/10.1017/S0022112004002526, 2005.

Iida, Y., Takatani, Y., Kojima, A., and Ishii, M.: Global trends of ocean CO2 sink and ocean acidification: an observation-based reconstruction of surface ocean inorganic carbon variables, J. Oceanogr., 77, 323-358, https://doi.org/10.1007/s10872-020-00571-5, 2021.

Leyba, I. M., Saraceno, M., and Solman, S. A.: Air-sea heat fluxes associated to mesoscale eddies in the Southwestern Atlantic Ocean and their dependence on different regional conditions, Clim. Dyn., 49, 2491-2501, https://doi.org/10.1007/s00382-016-3460-5, 2017.

Liu, Y., Yu, L., and Chen, G.: Characterization of Sea Surface Temperature and Air‐Sea Heat Flux Anomalies Associated With Mesoscale Eddies in the South China Sea, J. Geophys. Res.-Oceans, 125, https://doi.org/10.1029/2019jc015470, 2020.

Liu, Y., Zheng, Q., and Li, X.: Characteristics of Global Ocean Abnormal Mesoscale Eddies Derived From the Fusion of Sea Surface Height and Temperature Data by Deep Learning, Geophys. Res. Lett., 48, https://doi.org/10.1029/2021gl094772, 2021.

McGillicuddy, D. J.: Formation of Intrathermocline Lenses by Eddy–Wind Interaction, J. Phys. Oceanogr., 45, 606-612, https://doi.org/10.1175/jpo-d-14-0221.1, 2015.

McGillicuddy, D. J.: Mechanisms of Physical-Biological-Biogeochemical Interaction at the Oceanic Mesoscale, Annu. Rev. Mar. Science, 8, 125-159, https://doi.org/10.1146/annurev-marine-010814-015606, 2016.

Ni, Q., Zhai, X., Jiang, X., and Chen, D.: Abundant Cold Anticyclonic Eddies and Warm Cyclonic Eddies in the Global Ocean, J. Phys. Oceanogr., 51, 2793-2806, https://doi.org/10.1175/jpo-d-21-0010.1, 2021.

Provenzale, A.: TRANSPORT BY COHERENT BAROTROPIC VORTICES, Annu. Rev. Fluid Mech., 31, 55-93, https://doi.org/10.1146/annurev.fluid.31.1.55, 1999.

---

## Author Response (AR1)

**Response to reviews on 'Characteristics of Surface Physical and Biogeochemical Parameters within Mesoscale Eddies in the Southern Ocean'**

We thank the editor and both reviewers for their professional comments and constructive suggestions to improve the manuscript. We have addressed all comments and revised the manuscript accordingly. In the following, we address the editor and reviewer's comments point by point. Our response is given in the blue text below.

Yours sincerely,

Qian Liu, on behalf of the co-authors

**Response to Editor**

**General comments**

1. Please include all of the reviewers' suggestions into your revised version. Especially reviewer 1 raised the issues of the introduction not guiding well enough to your study.

Response: We have addressed each comment from the reviewers one by one and incorporated their suggestions into the revised version of the manuscript. The main modifications are shown as follows.

1)**Introduction**
In response to the comments from reviewer 1, the introduction section has been restructured to provide a clear overview of the research objectives and emphasize the novelty of the study. Additionally, we have explicitly stated that eddy-induced Ekman pumping may contribute to the occurrence of "abnormal" eddies in the introduction.

2)**Data and Methodology**
We provided more detailed information regarding the data sources and the methodology for deriving $pCO_2$, identifying "abnormal" eddies, and compositing eddy-induced anomalies.

3)**Discussion**
Compared to the previous response to the reviewers' comments, we have made

significant improvements to the discussion. Specifically, we revealed that in addition to the variation of the same parameter within different eddies, the dominant eddy-driven mechanisms for different parameters within the same kind of eddies also differ. The strength of the eddy stirring effect on different parameters is the major cause due to the different magnitudes of the horizontal parameter gradients.

Additionally, we have revised the abstract and conclusion sections to incorporate more details from the discussion and reflect the significance of our findings.

2. The discussion lacking some depth. My own concerns are in line with this reviewer's and I also would like to see more of a discussion on your findings in comparison with those of McGillicuddy, Gaube and colleagues on eddy induced Ekman pumping.

Response: We have deepened the discussion section and compared our findings with Gaube et al. (2014) regarding eddy-induced Ekman pumping in Section 6, Lines 396–439 (revised manuscript). We added a table, presented below, to illustrate the significance magnitudes of effects for different eddy-driven mechanisms on various parameters.

[revised manuscript text omitted]

3. In addition, every figure needs a thorough explanation in the text.

Response: We have thoroughly reviewed each figure and made revisions according to the suggestions provided by the reviewers and the editor. Additionally, we have ensured that every figure is accompanied by a detailed caption that provides essential information.

1) In response to the comments of reviewer 1, we removed Figs. 1, 3, and 5 and Tables 1 and 2 to the Supporting Information.

2) In Fig. 2, we used a sequential colormap, revised the figure legends to specify where the latitude 65°S is, and removed the currents and topographic features that are not mentioned in the manuscript.

3) We added captions explaining what the magenta boxes represent to Figs. 4, 6, 7, 10.

4) The caption of Fig. 4 has been revised to ensure that the blue and red colors are clearly associated with the right column only.

5) The font size in Figs. 5, 7, and 11 has been appropriately reduced.

Furthermore, in the Results section, we have provided thorough explanations for each figure, making it easier for readers to comprehend the key findings and their relevance to the study.

**Response to Reviewer #1**

**Overall impression**

The paper investigates the physical and biogeochemical characteristics of mesoscale eddies at the surface of the Southern Ocean – a region of global importance for heat and carbon exchange and biogeochemical cycles, concurrently a region dominated by eddies. This study involves many novel aspects compared to previous studies and tackles relevant topics for the community. It distinguishes between warm-core anticyclonic eddies (AEs), cold-core AEs, cold-core cyclonic eddies (CEs), and warm-core CEs (termed here as 'normal' and 'abnormal' AEs and CEs). At the same time, the discussion lacks some depth and many aspects of the methods and results/discussion remain unclear. There are also many figures which are only discussed very briefly and could be moved to the Supporting Information. These issues should be addressed before publication.

Response: We would like to thank reviewer 1 for the professional comments and valuable suggestions to improve the manuscript. We hope the answers and information presented here would respond to what was demanded.

**General comments**

1.  From the introduction, it's not entirely clear what's new about this study (one finds the information eventually, but it's quite hidden and only becomes apparent later). What's new about this study compared to previous work should really be the focus of the introduction. E.g., L86: Mention that the Frenger et al. studies investigated the vertical structure, while this study is only considering the surface. The same paragraph (from L86) also reads as if the only difference between the Frenger studies and this study is that this study differentiates between 'normal' and 'abnormal' eddies. However, there are many other differences (surface vs. interior; which parameters are considered; method of eddy detection…).

Response: Thanks for the suggestion. We have restructured the Introduction section to provide a clear overview of the research objectives and emphasize the novelty of the study. The key differences between our work and previous studies were explicitly stated from several perspectives, including

1) The specific impact of "abnormal" eddies on physical and biogeochemical parameters in the SO remains unclear (Lines 70–73 in the revised manuscript).

2) Previous studies have primarily focused on the basin-wide effects of eddies on Chl-$a$, while investigations into the basin-scale effects of SO eddies on DIC and $p$CO$_2$

are lacking (Lines 73–75 in the revised manuscript).

3) It is necessary to systematically study the influence of eddies on SST, Chl-*a*, DIC, and $p$CO$_2$ in the SO (Lines 75–77 in the revised manuscript).

4) Compared to traditional eddy detection methods, the deep learning model can simultaneously detect eddy locations and distinguish between normal and "abnormal" eddies with great accuracy and efficiency (Lines 79–86 in the revised manuscript).

5) Our study focuses solely on the surface (Lines 1, 10, 78, 91, and 448 in the revised manuscript).

2. My biggest concern: McGillicuddy, Gaube, and others have pointed out that eddy-induced Ekman pumping results in the opposite signal compared to regular eddy pumping and that eddy-induced Ekman pumping is usually weaker, but can be significant, especially in regions with large wind stress. Thus, when seeing cold core AEs or warm core CEs, I would assume that there, eddy-induced Ekman pumping dominates. However, in this paper it is framed like a mystery that some AEs have cold cores, and some CEs have warm cores (termed 'abnormal eddies'). Later, the study exactly finds this, at least in the analysis with SST (Section 5.1, esp. L278). Thus, I recommend rephrasing the storyline that cold-core AEs and warm-core CEs are likely to be dominated by eddy-induced Ekman pumping.

Response: Thanks for the suggestion. We agree that eddy-induced Ekman pumping is a primary generation mechanism of "abnormal" eddies. In addition, other mechanisms are also proposed to induce the formation of "abnormal" eddies. Therefore, we added a review about the generation mechanisms of "abnormal" eddies in the Introduction section, Lines 67–70 (revised manuscript):

"Previous literature proposed that "abnormal" eddies may be induced by eddy-induced Ekman pumping (Gaube et al., 2013; Mcgillicuddy, 2015), instability during the eddy decay stage, eddy horizontal entrainment (Sun et al., 2019), and warm/cold background water (Leyba et al., 2017)."

After rephrasing the introduction, we proved that the formation of "abnormal" eddies is dominated by eddy-induced Ekman pumping in the Results section, making the study storyline more reasonable.

3. The discussion should be deepened. Currently, Sections 4 and 5 mostly show the results with a lot of figures, and Section 6 (Conclusions) is mostly a summary of these findings. I'm missing a more in-depth discussion of what this now all means and how it matters. One thing to focus on especially is the surprising finding that when considering SST anomalies, eddy-induced Ekman pumping dominates in certain eddies,

while for the other variables, different processes dominate. How can the same eddies pump DIC-rich water upwards without pumping cold water up? Once the discussion has been deepened, the abstract and conclusions can then also mention some more from the discussion. Right now, the abstract and conclusion sections are quite descriptive of the results but don't tell us much about their significance.

Response: Thanks for your valuable suggestions. We have deepened the discussion section in Section 6, Lines 396–439 (revised manuscript). The major improvements to the discussion include the following:

1) Comparing the significance magnitudes of effects for eddy trapping, stirring, pumping, and eddy-induced Ekman pumping on SST, Chl-*a*, and DIC (Table 1 in the revised manuscript), we found distinct influence mechanisms of the same kind of eddies on various parameters.

2) Calculating the average horizontal gradients of SST, Chl-*a*, and DIC, we revealed that the different dominant eddy-driven mechanisms for various parameters within the same kind of eddies are the results of the distinct strength of the eddy stirring effect on different parameters due to the different magnitudes of the horizontal parameter gradients.

3) The negligible impact of eddy stirring on SST due to small gradient values results in pronounced monopole patterns within eddies. By contrast, eddy stirring has a stronger impact on Chl-*a* and DIC, resulting in dipole patterns within eddies.

4) Both eddy stirring and eddy pumping contribute to the generation of negative/positive Chl-*a* anomalies within AEs/CEs. The combined effects of eddy stirring and eddy pumping dominate the similar patterns of Chl-*a* anomalies in normal and "abnormal" eddies.

5) Compared with the findings reported by Gaube et al. (2014), we further proved that the effect of eddy-induced Ekman pumping on Chl-*a* is relatively small and contributes to attenuating magnitudes of Chl-*a* anomalies within "abnormal" eddies.

6) Both eddy pumping and eddy-induced Ekman pumping contribute to the variations of SST and DIC. For normal eddies, eddy pumping dominates the vertical distribution of SST and DIC. Within CCEs, the upwelling of cold, DIC-rich deep water induces negative SST anomalies and positive DIC anomalies, whereas the reverse is true for WAEs. However, the influence of eddy-induced Ekman pumping becomes more prominent within "abnormal" eddies. Within WCEs, the downwelling of warm, low-DIC surface waters induces positive SST anomalies and negative DIC anomalies, whereas the reverse is true for CAEs.

Furthermore, the abstract and conclusion sections have also been revised to include

more information from the discussion and reflect the significance of our results.

4.   Consider moving Fig.1 and 3 and Table 1 and 2 to the Supporting Information, they don't add much new information. Fig. 5 is only discussed with one sentence (L192) and could also move to the SI or be discussed in more depth.   Similarly, Fig. 7 is only very briefly touched upon and can move to the SI.

Response: Thanks for the suggestion. In the revised manuscript, Figs. 1, 3, and 5, and Tables 1–2 have been moved to the Supporting Information. Besides, Fig. 7 is retained in the manuscript since it describes the directions of horizontal parameter gradients which are used to evaluate the effects of eddy trapping and stirring on parameters, and a more detailed description and discussion about Fig. 7 are added in Section 5.

**Specific comments**

1.   It is not immediately clear that the study only focuses on surface properties. This could be added to the title and should be clearer in the abstract and introduction.

Response: We have revised the title, abstract, and introduction to make it clearer that our study solely examines surface properties.

2.   L10: add 'horizontal surface' before 'composite'

Response: "horizontal surface" has been added.

3.   L73: Are the signals also different when the seasonal signal has been removed? I.e., are the anomalies computed based on mean annual reference values, or on a monthly climatology? I have a feeling that if a monthly climatology is used as a reference, the eddy anomalies might not differ so much anymore by season. This should be mentioned/discussed.

Response: We have modified the method section and added a figure (as shown below) in the Supporting Information to show that we have removed the seasonal signals. The SST and Chl-$a$ anomalies are computed using a 7–90 days band-pass filter to remove the seasonal signal. For DIC and $p$CO$_2$ datasets with monthly temporal resolution, we have subtracted their climatological averages.

Since we removed the seasonal signals, there are no significant seasonal variations in eddy-induced SST, Chl-$a$, and DIC anomalies, as shown in Fig. 1 below. Moreover, Fig. 8 also shows little variation in composite averages of SST, Chl-$a$, and DIC anomalies within eddies between summer and winter.

Only $p$CO$_2$ anomalies with monthly temporal resolution show remarkable seasonal

variations caused by the different dominant effects of SST, Chl-*a*, and DIC, which has been discussed in the revised manuscript (Lines 364–394).

[Figure]

Figure 1. Variations in monthly mean eddy-induced anomalies, including (a) SST, (b) Chl-*a*, (c) DIC, and (d) $p\mathrm{CO_2}$ in the SO from 1996 to 2015. Solid lines in different colors denote four kinds of eddies.

4. L119: Be specific that the DIC is gap-filled in this step.

Response: Thanks for your suggestion. We have explicitly mentioned in the manuscript that the DIC field is gap-filled during this process.

5. Section 2.2 and 2.3: be clearer that those datasets used were created by previous studies. It currently reads ambiguously if this was done during this study or if the data is from previous work.

Response: We have revised the manuscript to make it clear that these datasets were created by previous studies.

6.  L127: Is 'This dataset' referring to Landschuezter et al. 2014, or to the Liu et al. 2021 product that is used in this study? If it's referring to Landschuezter: as that product has been used so widely, why did you not use that product? What's the benefit of using Liu et al? If it's referring to Liu et al: rephrase the sentence so that it's clearer (but then some of the references are wrong as they were published before 2021…)

Response: We appreciate the reviewer's comment and apologize for the confusion caused by the ambiguous reference in the manuscript. To clarify, in Line 127, "This dataset" refers to the dataset used in our study, which is from the JMA Ocean $CO_2$ Map dataset, established by Iida et al. (2021). The sentence in the revised manuscript (Lines 113–114) is as follows: "The $pCO_2$ and DIC datasets are from the Japan Meteorological Agency (JMA) Ocean $CO_2$ Map dataset with monthly $1° × 1°$ gridded values on the global ocean from 1990 to 2020 (Iida et al., 2021)."

Besides, the reason for citing references published before 2021 is that the initial version of this database was published in 2015 (Iida et al., 2015). These two versions of the $pCO_2$ dataset use the same approach, multiple linear regression (MLR) method. The difference is that the initial one uses sea SST, sea surface salinity (SSS), and Chl-*a* as independent variables (Iida et al., 2015). By contrast, the new version of the $pCO_2$ dataset is reconstructed from the fields of total alkalinity (TA), DIC, SST, and SSS (Iida et al., 2021).

To address the issue of the data source and references, we have emphasized that the dataset is provided by Iida et al. (2021) and cited accurate references related to this dataset.

7.  Section 2.3: Mention how the eddy detection method differs from other, more commonly used approaches, such as the AVISO eddy database (newest version: Pegliasco et al. 2021), and why it was preferred. One could have used the AVISO eddies and classified the eddies into normal and abnormal based on their SST signature (e.g., AE with cold SST anomaly is CAE...).

Response: We have added a paragraph in Section 2.3 to highlight the differences between our eddy detection method and the AVISO eddy database, as well as why we choose this eddy dataset. Compared to the AVISO eddy database (Pegliasco et al., 2022), our study utilizes a different eddy detection method (Liu et al., 2021), which uses a deep learning model to fuse satellite SSH and SST data. The reason why we use this method is that deep learning technology has unparalleled learning ability and the capability to model complex nonlinear relationships compared to traditional statistics and machine learning methods (Reichstein et al., 2019). Besides, the method can simultaneously extract SSH features for determining eddy locations and extract SST information to help distinguish between normal and "abnormal" eddies. As a result, our method achieves great accuracy and much higher efficiency than the traditional method

that first detects the eddies and then uses the SST signature to classify them into normal and "abnormal" eddies. In addition, the method is able to detect eddies in regions where traditional methods may not be effective, such as in regions with weak eddies or regions with complex oceanic dynamics (Liu et al., 2021). Given its high accuracy and comprehensive information on eddy characteristics, we find this dataset particularly useful for our study.

8. L144-149: I think this part of the paragraph still belongs to section 2.3.

Response: Thanks for the suggestion. This paragraph describes the methodology for obtaining the distribution of various sea surface variables within the eddy, so it should be included in the methodology section 3.1 rather than the data section 2.3. To address this concern, we revised the methodology section title to "Composite Eddy-induced Anomalies" and enhanced the methodology section to provide a more explicit explanation of this point in the paper.

9. L149-153: Needs a more in-depth description of how the composite eddies were made.

Response: We have added a schematic in the Supporting Information (as shown in Fig. 2 below) and revised the section to provide additional details on the methodology used to create the composite eddies in Section 3.1, Lines 168–178 (revised manuscript). The revised section reads as follows:

"Finally, we use the eddy-centric composite method to estimate the spatial pattern of the eddy-induced anomalies in sea surface variables. The positions of co-located SST, Chl-$a$, DIC, and $p$CO$_2$ observations are normalized by R, which defines the edge of an eddy as ±1 and the eddy core as 0. This allowed us to construct composite averages from eddies of varying sizes. We then extract data from −2R to 2R to include the interactions between eddies and the surrounding waters and interpolate them onto an evenly spaced 17 by 17 grid to create the surface composite patterns. For daily SST and Chl-$a$, we perform the eddy-centric composite method matching eddies and variables on the same day and calculate the mean value. By contrast, for monthly DIC and $p$CO$_2$, we calculate the eddy-centric composite maps, using all eddies of the same month with DIC and $p$CO$_2$ of that month and calculate the mean value. The composites are not rotated with the background variables gradient, as the large-scale background variables gradients in the SO are oriented north-south. Previous studies have shown that rotating eddies to the large-scale variables gradient in the SO has a negligible impact on the results (Frenger et al., 2015). Therefore, the axes in each figure point north and east."

[Figure]

Figure 2. Schematic of eddy-centric composite method for daily (a) SST and Chl-*a* and monthly (b) DIC and $pCO_2$, taking January as an example.

10. L171: Be explicit about how it differs from the method by Gaube et al. 2015.

Response: We have revised the section to clarify that we use the same formula to calculate the total eddy-induced Ekman pumping and the same spatiotemporal filtering method as Gaube et al. (2015). However, we only calculate the total eddy-induced Ekman pumping and do not calculate the individual components such as SST-induced Ekman pumping and current-induced Ekman pumping. Besides, the SSH dataset we use is constructed at a daily temporal resolution, whereas the SSH data used by Gaube et al. (2015) is constructed at 7-day intervals.

11. Section 3.2: Add a reference for the methods to obtain the eddy-induced Ekman pumping.

Response: A reference (Gaube et al., 2015) has been added.

12. L239: Discuss why we want to know how the pattern differs from the $pCO_2$ pattern. I would have found it more interesting to see the pattern differences between normal and abnormal eddies, but there could be a reason why you chose this.

Response: We revised this section to explain why we discuss the pattern differences

between $pCO_2$ and other variables. While SST, Chl-$a$, and DIC anomalies within eddies are found to be similar in summer and winter, $pCO_2$ anomalies are significantly different between the two seasons. This is because, in winter, the $pCO_2$ anomalies are dominated by the dominant DIC-driven effect. By contrast, in summer, the $pCO_2$ anomalies are dominated by SST and DIC anomalies in some regions with smaller and larger magnitudes of DIC anomalies, respectively. Due to the opposite anomalies of DIC and SST within the same kind of eddies, the basin-scale effects of SO normal and "abnormal" eddies on $pCO_2$ have little significant pattern differences in summer. Since we already discussed SST and DIC pattern differences between normal and "abnormal" eddies, understanding the relationship between $pCO_2$ and other anomalies within the eddies can explain the differences in $pCO_2$ patterns within normal and "abnormal" eddies.

13. L255: Mention why stirring is not a process (we can see it in the plot, but it needs to be discussed).

Response: In Section 5.1, Lines 300–304 (revised manuscript), we proposed that the meridional and zonal phase shifts in normal eddies are induced by the large-scale background SST gradient and eddy stirring. However, the SST anomalies within abnormal eddies show purely monopoly patterns, which do not reflect the stirring impact. In contrast, processes such as eddy pumping and eddy-induced Ekman pumping have a more significant impact on SST anomalies. Therefore, we did not regard it as a major process regulating the SST anomalies in eddies.

14. Generally: Personally, I would not use the terms 'normal' and 'abnormal', as everything is normal and within the expected physics (when considering eddy-induced Ekman pumping), but this may be a personal choice. Maybe 'regular' and 'unusual' fits better, as warm-core AEs and cold-core CEs are a lot more common than cold-core AEs and warm-core CEs, but I'm nit-picking now.

Response: We understand that everything can be considered normal within the expected physics, therefore, we have chosen to use the term "abnormal" in quotes to indicate that it is a relative term and not an absolute one, and we are referring specifically to departures from the expected behavior. The terms "normal" and "abnormal" are commonly used in the scientific community to describe expected and unexpected phenomena, and using them consistently throughout the paper helps maintain clarity and coherence.

15. Fig. 2: Consider using a sequential colormap. Specify the latitude where the white region starts (65S?). Most of the currents and topographic features are not referred to in the text and can be removed.

Response: A sequential colormap has been used in Fig. 2. The figure legends have been revised to specify where the latitude 65°S is. The currents and topographic features that

are not mentioned in the manuscript have been removed.

16. Fig. 4 (and the following figures): Add in the caption what the magenta boxes are.

6) Response: We added captions explaining what the magenta boxes represent to Figs. 4, 6, 7, 10.

17. Fig. 6: Why are there some warm spots in cold eddies, and cold spots in warm eddies? By definition, the SST anomalies should be cold in cold eddies, and warm in warm eddies.

Response: Thanks for the comment. We apologize for the confusion caused by our carelessness in using outdated SST anomaly data for CAEs and WCEs in Fig. 6. We have revised the figure using the correct data. We assure that all other results involving SST anomaly data within eddies are based on the correct data.

18. Fig. 8: Ensure all SSIMs have the same number of decimals.

Response: All SSIMs have been corrected using the same number of decimals.

**Technical corrections**

1. Throughout the document: change biochemical to biogeochemical.

Response: The word biochemical has been changed to biogeochemical.

2. It's a good habit to discuss the findings of this paper in the present tense and refer to previous studies in the past tense. E.g., L9: change to 'we analyze' (instead of 'we analyzed'); same throughout the whole document.

Response: Thanks for pointing it out. We have checked the entire document and made corrections accordingly.

3. L12. I know many studies do this and it is a personal choice, but I dislike sentences with brackets for multiple things. Consider writing it out for each, e.g., 'dominated by DIC anomalies in regions with larger magnitudes of DIC anomalies and dominated by SST anomalies in regions with smaller magnitudes. Same throughout the whole document.

Response: We have made corrections accordingly.

4. L22: existing (not exiting)

Response: The word exiting has been changed to existing.

5.  The font in some figures is very large.

Response: The font size in Figs. 5, 7, and 11 has been appropriately reduced.

**Response to Reviewer #2**

**Overall impression**

The manuscript analyzed the biochemical influences of mesoscale eddies (including normal and abnormal eddies) in the Southern Ocean, by using machine learning and multi-source marine dataset. The manuscript estimated chlorophyll (Chl) and dissolved inorganic carbon (DIC) contributions to $p$CO$_2$, and found their seasonal variations. These results are interesting and are of vital importance on global biogeochemical cycles and the climate change. However, some description about methods/data are ambiguous and few conclusions need to be further discussed.

Response: We would like to thank reviewer 2 for taking the time to review the manuscript and for its valuable feedback. We acknowledge that the suggestions provided have really helped to improve the quality of this work. We hope the answers and information provided here would respond to what was demanded.

**Major questions:**

1.  There are some other methods to identify abnormal eddies, such as using potential density and directions of geostrophic current. They are supposed to be introduced in the introduction, and point out why authors choose the method of SSTA.

Response: We acknowledge that there are other methods to identify abnormal eddies, such as using potential density and geostrophic current direction (Mcgillicuddy, 2015). We added this information in the introduction section and clarified why we used SSTA to distinguish between normal and abnormal eddies (Lines 82–86 in the revised manuscript).

Recent studies have found that abnormal eddies show opposite SSTA signals to normal eddies (Leyba et al., 2017; Liu et al., 2020; Liu et al., 2021; Ni et al., 2021). Compared to potential density, SSTA data can be obtained from satellite remote sensing with higher spatial and temporal resolutions, making it a convenient and reliable data source for identifying eddies (Castellani, 2006; Liu et al., 2021).

Moreover, detecting eddies using directions of geostrophic current is essentially based on SSH features. Our study utilizes the abnormal eddy dataset that Liu et al. (2021) developed, which uses a deep learning model to fuse satellite SSH and SST data. The method can simultaneously extract SSH features for determining eddy locations and extract SST information to help distinguish between normal and abnormal eddies with great accuracy and efficiency. In addition, the method is able to detect eddies in regions where traditional methods may not be effective, such as in regions with weak eddies or regions with complex oceanic dynamics (Liu et al., 2021). Given its high accuracy and

comprehensive information on eddy characteristics, we find this dataset particularly useful for our study.

2.  The methods to derive $pCO_2$ from Chl, SST, DIC and other variables are supposed to be introduced with more descriptions or equations.

Response: We have added the following details to explicitly describe the methods to derive $pCO_2$ in Lines 123–126:

"The $pCO_2$ field is calculated from TA, DIC, SST, and SSS based on seawater $CO_2$ chemistry (Iida et al., 2021). Firstly, the mean rates of regional $pCO_2$ and multiple regressions are used to derive the algorithms of $pCO_2$ expressed empirically as a function of in situ TA, DIC, SST, SSS, and the year. Then, the $pCO_2$ fields that filled both in space ($1° \times 1°$) and time (monthly) are drawn by applying global data sets of TA, DIC, SST, and SSS to the variables in these empirical equations."

3.  The method to define and identify abnormal eddies should be introduced in detail even if the authors cited the paper of Liu et al., 2021. Did they identify abnormal eddies according to SSTA>0/SSTA<0 within eddy boundaries/cores? How did they distinguish AEs and CEs just according to SSTA?

Response: We have updated the manuscript to include more detailed information on our methodology for identifying abnormal eddies (Lines 133–141 in the revised manuscript).

We distinguished between normal and abnormal eddies based on the mean SSTA within eddy boundaries. Besides, we distinguished between AEs and CEs based on the SSHA, as AEs (CEs) are usually accompanied by local convergence (divergence), leading to positive (negative) SSHA. Specifically, WAEs are identified according to SSHA >0 and SSTA >0, CAEs are identified according to SSHA >0 and SSTA <0, CCEs are identified according to SSHA <0 and SSTA <0, and WCEs are identified according to SSHA <0 and SSTA >0.

4.  The descriptions about eddy dataset and identification are very poor. In line 139, the authors mentioned "the ground truth data set". What's the ground truth data set of eddies? Is it produced by the authors or a public dataset? That's important to the verification.

Response: Thanks for the suggestions. We have verified in the manuscript that the ground truth dataset of mesoscale eddies used in our study was generated automatically using the SSH-based method proposed by Haller (2005), and the eddy dataset was produced by Liu et al. (2021).

5.  How did authors match daily eddy dataset with monthly DIC and $pCO_2$ temporally

and spatially when doing composite analyses? Temporally, is eddy at JAN. 31st matched with DIC of JAN. or Feb. data? is DIC data used within eddy boundaries or eddy cores?

Response: We have added a schematic in the Supporting Information and revised the section to provide additional details on the methodology used to create the composite eddies (Lines 161–178 in the revised manuscript). The positions of co-located SST, Chl-$a$, DIC, and $p$CO$_2$ observations were normalized by R, which defines the edge of an eddy as ±1 and the eddy core as 0. This allowed us to construct composite averages from eddies of varying sizes. We then extracted data from −2R to 2R to include the interactions between eddies and the surrounding waters and interpolated them onto an evenly spaced 17 by 17 grid to create the surface composite patterns. Therefore, the mean anomalies of SST, Chl-$a$, DIC, and $p$CO$_2$ are used within eddy boundaries.

For daily SST and Chl-$a$, we performed the eddy-centric composite method matching eddies and variables on the same day and calculated the mean value, as shown in the Fig. S2a. By contrast, for monthly DIC and $p$CO$_2$, we calculated the eddy-centric composite maps, using all eddies of the same month with DIC and $p$CO$_2$ of that month and calculate the mean value (Fig. S2b in the revised manuscript).

6.    Taking CEs for example, commonly, upwellings are thought to transport cold water to the sea surface, as well as richer nutrients at the same time. Therefore, CEs often show lower SST and higher chlorophyll. The conclusions from Figure 8 show SSTA within abnormal eddies are dominant by Ekman pumping. However, the chlorophyll anomalies of abnormal eddies are attributed to eddy pumping. The conclusions are contradictory to each other. If they are reliable, what's the mechanism leading to contrasting vertical process on SST and chlorophyll respectively? Therefore, discussions of lines 279-280 and 295-296 need more explanations. Besides, line 279 should be "eddy-induced Ekman pumping".

Response: We have revised the Results section accordingly and deepened the discussion.

In response to the question, we further calculated the mean gradient of SST and Chl-$a$ and revealed that the different dominant eddy-driven mechanisms for SST and Chl-$a$ within the same kind of eddies are the results of the distinct strength of the eddy stirring effect due to the different magnitudes of the horizontal parameter gradients.

The average gradients of SST and Chl-$a$ are found to be 0.03 and 0.08, respectively. The north-south gradients (north is the positive direction) of SST and Chl-$a$ are 0.04 and −0.02, respectively. The east-west gradients (east is the positive direction) of SST and Chl-$a$ are 0.00 and −0.04, respectively. As eddy stirring redistribute physical and biogeochemical parameters spatially through horizontal advection, the larger the horizontal parameter gradient, the stronger the eddy stirring effect (Mcgillicuddy, 2016).

The small gradient of SST leads to a negligible effect of eddy stirring. Within normal eddies, eddy pumping dominates the vertical heat advection, resulting in positive and negative SST anomalies in WAEs and CCEs, respectively. However, within "abnormal" eddies, the effect of eddy-induced Ekman pumping becomes more prominent, resulting in negative and positive SST anomalies in CAEs and WCEs, respectively.

Compared to SST, Chl-*a* has a higher gradient, resulting in a stronger effect of eddy stirring. The gradients of Chl-*a* suggest that the climatological Chl-*a* increases southward and westward. Counterclockwise rotation of AEs in the SO would advect low Chl-*a* from the northeast to the west and high Chl-*a* from the southwest to the east. The reverse is true for CEs. Previous works found that the dipole shapes arising from stirring tend to be asymmetric, with larger anomalies at the leading compared to the trailing side of eddies (Chelton et al., 2011; Frenger et al., 2015; Dawson et al., 2018; Frenger et al., 2018). As the major propagation direction of eddies is westward, the composite Chl-*a* anomalies in AEs/CEs show dominant negative/positive signals due to eddy stirring. Besides, eddy pumping tends to produce Chl-*a* anomalies of the same sign. The common effects of eddy stirring and eddy pumping overcome the effect of eddy-induced Ekman pumping, resulting in similar patterns of Chl-*a* anomalies in normal and "abnormal" eddies.

However, from Figs. 8b1–b4 and f1–f4 in the manuscript, we can see that the magnitudes of Chl-*a* anomalies within normal eddies are higher than "abnormal" eddies, which reflects the effect of eddy-induced Ekman pumping. Besides, in some regions with small amplitude, such as the south of ACC and the South Pacific Ocean, we find Chl-*a* anomalies in AEs/CEs are positive/negative (Figs. 6e–h in the manuscript). Such a result may be caused by a more dominant effect of eddy-induced Ekman pumping on Chl-*a*. Overall, eddy stirring and eddy pumping are mainly responsible for the patterns of Chl-*a* anomalies within eddies in the SO, and eddy-induced Ekman pumping attenuates the magnitudes of Chl-*a* anomalies within "abnormal" eddies.

7. The manuscript is supposed to evaluate the accuracies of abnormal eddy identification method, which can combine with Argo profiles via temperature and potential density. At the same time, it should point out the method improvement in future

Response: We have updated the manuscript to address these points. The experiments showed that the model could accurately identify "abnormal" eddies in the South China Sea (SCS) and Kuroshio Extension (KE) region (Liu et al., 2021). In addition, Argo floats data also verified the accuracy and validity of the model (Liu et al., 2021).

However, we also acknowledge that there is room for improvement in our method. Considering that the changes in SSH, SST, Chl-*a*, and roughness caused by eddies can be recorded by altimeter, infrared, ocean color, and synthetic aperture radar (SAR) remote sensing, respectively, and potential density and temperature recorded by Argo

floats can also identify "abnormal" eddies, in future work, we will combine multiple remote sensing data with Argo profiles to evaluate the accuracies of "abnormal" eddy identification method.

**Minor questions:**

1.  Lines 31-32: Authors point out that eddies have influences on "biochemical parameters". While, the listed references are both about chlorophyll, which is a biological parameter. References about chemical parameters should be introduced.

Response: References about chemical parameters have been added.

2.  Lines 37-39: Rotations of eddies are related to the hemisphere. It should illustrate which hemisphere is talked about.

Response: When mentioning the rotations of the eddies, we emphasized that the eddies is in the Southern Hemisphere.

3.  Line 56: How about eddy influence on chlorophyll during wintertime with deeper mixing?

Response: Thanks for your suggestions. We added information and references about the influence of eddies on Chl-*a* during wintertime with deeper mixing in AEs in the introduction.

Both mixing and eddy-induced Ekman pumping tend to produce Chl-*a* anomalies of the same sign in the nutrient-limited SO. For instance, shallower mixed layers in cyclonic eddies could result in lower Chl-*a*, while deeper mixed layers in anticyclonic eddies could lead to higher Chl-*a*. Dufois et al. (2014) suggested that in the South Indian Ocean between 20°S and 30°S, deeper mixing in winter AEs can elevate nutrient supply, while shallower mixing in CEs can reduce it, which could explain stronger positive Chl-*a* anomalies in AEs than in CEs. Dawson et al. (2018) indicated that the deepening of winter and early spring mixed layers in anticyclones and shallowing of mixed layers in cyclones are the main drivers of the positive Chl-*a* anomalies in AEs and negative Chl-*a* anomalies in CEs in summer and autumn between the Subtropical Front and the Polar Front. Such seasonal lag effect is due to deeper mixed layers in winter and spring that enhance light limitation, reducing the biological effect in AEs (Song et al., 2016). The remained nutrients in the mixed layer of AEs could sustain higher phytoplankton levels in summer when light limitation is alleviated, leading to positive Chl-*a* anomalies in summer and autumn. By contrast, the shallow mixing in CEs makes phytoplankton communities begin to consume nutrients earlier, leading to negative Chl-*a* anomalies in summer and autumn. Therefore, the influence of mixing within eddies on Chl-*a* is seasonal.

In our study, the Chl-*a* anomalies within eddies show litter seasonal variation, which contradicts the seasonal effect of mixing. Therefore, the seasonal modulation of the mixed layer is not discussed in our study. Instead, eddy stirring and eddy pumping are the main modulation processes of normal and "abnormal" eddies to Chl-*a* in the SO. Composite Chl-*a* anomalies display negative signatures in both WAEs and CAEs and positive signatures in CCEs and WCEs. However, we find the magnitudes of Chl-a anomalies within "abnormal" eddies are smaller than normal eddies, which associates with the effect of eddy-induce Ekman pumping.

4. Lines 107-108: The expression of OI-SST should be in agreement.

Response: We have corrected the expression of OI-SST.

5. Line 166: JMA is suggested to be introduced as Japan Meteorological Agency.

Response: We have introduced JMA as "Japan Meteorological Agency" upon its first mention in the manuscript.

6. Line 178: What are the denominators when calculating eddy frequencies? It should be expressed more clearly.

Response: We have added the definition of eddy frequency. The eddy frequency is the ratio of the number of days eddies appeared to the total number of observation days.

7. Lines 186-187: The conclusion is true in South America, but not evident in the south of Australia.

Response: We have revised the manuscript to demonstrate the findings explicitly. Based on Figs. 4c and 4f in the manuscript, it can be observed that "abnormal" eddies have a polarity distribution opposite to that of normal eddies in the continental boundary currents where more CCEs and CAEs occur. However, it should be noted that more WAEs and WCEs occur in the south of Australia.

8. Line 285: How to understand "eddy trapping has little influence on Chl-*a*"? Please give more descriptions to explain it.

Response: We have added more descriptions to explain why eddy trapping has little influence on Chl-*a*. Nonlinear eddies tend to trap the fluid contained in their interiors (Provenzale, 1999; Mcgillicuddy, 2016). The composition of the trapped fluid is dependent on various factors, including the eddy propagation and the local gradients in physical and biochemical properties. The tracks of long-lived eddies with lifetimes longer than 1 year show that the major propagation direction of eddies is westward, with AEs propagating north and CEs propagating south. Due to the climatological Chl-*a* increasing southward, AEs propagating northward tend to trap high Chl-*a* into

northern areas with low Chl-*a*, as shown in Fig. 3 below. On the other hand, CEs propagating southward tend to trap low Chl-*a* into southern areas with high Chl-*a*. Such effect of eddy trapping on Chl-*a* contradicts the actual composite Chl-*a* anomalies over eddies with negative Chl-*a* anomalies in AEs and positive Chl-*a* anomalies in CEs. As a result, we conclude that eddy trapping has little influence on Chl-*a*.

[Figure]

Figure 3. Schematic illustrating the eddy trapping of how AEs and CEs affect Chl-*a*. Red and blue colors represent high and low Chl-*a*, respectively.

9. Lines 303-314: Are those conclusions for summertime still "dominant"? The magnitudes seem similar for summertime.

Response: Yes, there are no significant seasonal variations in eddy-induced SST, Chl-*a*, and DIC anomalies. Therefore, the dominant mechanisms of eddies affecting these variables do not alter by season.

10. Figure 11 is suggested to be shown in wintertime and summer time respectively, based on which Figure 8, Figure 12, and Figure 13 can be better discussed.

Response: We appreciate the suggestion, but given the nature of our findings, presenting the annual mean is more appropriate for our study. In the original manuscript, we did not discuss the SST, Chl-*a*, and DIC anomalies within the eddies seasonally, as their variations did not exhibit significant seasonal patterns. Therefore, we opted to present the annual mean eddy-induced Ekman pumping in Figure 11, as it provides a comprehensive representation of the differences in variable anomalies between normal and "abnormal" eddies. We believe that this approach allows for a more convenient and meaningful comparison of the variable anomalies within the eddies.

11. In Figure 4, lines 558-599, the authors mean blue and red colors in the right column. However, blue and red colors are shown in each sub-figure, which is misleading.

Response: We have revised the manuscript to ensure that the blue and red colors are clearly associated with the right column only.

12. Figures 4d and 4e show that abnormal eddies occur along fronts, where eddies are active, and along offshore areas where accuracies of altimeters are low. It is suggested to show ratios of abnormal eddies to normal eddies (WAEs/CAEs, CCEs/WCEs). Will the abnormal eddy signals offshore be amplified offshore? What are the mean depths of clustered abnormal eddies? It should be cautious with eddies shallower than 1000 m.

Response: We appreciate the reviewer's insightful suggestions, and we calculated the ratios of "abnormal" eddies to normal eddies, as shown in Fig. 4 below. We find more CAEs in the Western Boundary Current (WBC) regions and significant dominance of WCEs in southern Australia. In the southeast of America and Campbell Plateau, with depths shallower than 1000 m (Fig. 5 below), "abnormal" eddy signals offshore may be amplified offshore due to the low accuracies of altimeters along offshore areas. We further calculate the mean depths of clustered eddies. The mean depths of WAEs, CAEs, CCEs, and WCEs are 4086 m, 3969 m, 4044 m, and 4014 m. All of them are deeper than 1000m. As mentioned in the manuscript, eddies disappear in regions shallower than 2000m because the bottom topography constrains the generation of eddies. Therefore, the amplified "abnormal" eddy signals in the southeast of America and Campbell Plateau have little influence on the results.

[Figure]

Figure 4. Spatial distribution of eddy polarity dominance in the SO from 1996 to 2015. (a) Ratio of the area occupied by WAEs over the area covered by CAEs. (b) Ratio of the area occupied by CCEs over WCEs. Values >0 in red and <0 in blue mark the dominance of normal and "abnormal" eddies, respectively. Black solid lines show the mean northern and southern positions of the ACC major fronts. The black dotted circle is 50° S. The magenta boxes represent ARC and SWA regions.

[Figure]

Figure 5. Southern Ocean topography and current. Black solid lines show the mean northern and southern positions of the ACC major fronts. The black dotted circle is 50° S.

13. Figure 6. The abnormal eddies are identified from SST so the SSTA of Figure 6 is regular. The other three parameters are very noisy. The magnitudes of chlorophyll and $p$CO$_2$ signals induced by abnormal eddies are even higher than normal eddies, which are contrasting with eddy amplitude comparisons. Why?

Response: We have added some explanations to the manuscript to address the reviewer's concern. The distributions of Chl-$a$ anomalies over both normal and "abnormal" eddies are similar to the eddy amplitude distributions, with stronger negative/positive anomalies within AEs/CEs in regions of higher amplitude. This result indicates the dominant effect of eddy pumping on Chl-$a$. However, in regions of lower amplitude, we find the patterns of Chl-$a$ anomalies are spotty, with average positive/negative Chl-$a$ anomalies in AEs/CEs. Such a result may be caused by a more dominant effect of eddy-induced Ekman pumping on Chl-$a$.

The magnitudes of Chl-$a$ anomalies induced by "abnormal" eddies are even higher than normal eddies in these regions due to the smaller amplitude and eddy pumping of "abnormal" eddies than normal eddies. Furthermore, in some regions, such as SWA, the magnitudes of $p$CO$_2$ anomalies induced by "abnormal" eddies are higher than normal eddies, which are related to the stronger eddy-induced Ekman pumping of "abnormal" eddies.

**Reference**

[revised manuscript text omitted]

---

## Author Response (AR2)

**Overall impression**

The manuscript aims at biochemical influences of normal and abnormal eddies in the Southern Ocean (SO), discussing influences of different eddy mechanism on sea surface temperature (SST), chlorophyll-a (Chl-*a*), dissolved inorganic carbon (DIC) and their contributions to $p$CO$_2$. The topic is of vital importance on eddy contributions in the global biogeochemical cycles. Many results are interesting and the figures are displayed very clearly. While, some conclusions still should be rethinking and the writing could be improved. The manuscript would be more suitable for publication after a major revision.

Response: We would like to thank the reviewer for the professional comments and valuable suggestions to improve the manuscript. We hope the answers and information presented here would respond to what was demanded.

**Major questions:**

1. The manuscript needs to highlight its innovation and scientific meanings. Although the authors stress eddy's importance on the biogeochemical cycle, what are the contributions to future scientific studies and what are the scientific meanings?

Response: Thanks for your suggestion. In the manuscript, we stated four significant scientific meanings, including

1) Considering the distinct role of "abnormal" eddies in modulating physical and biogeochemical parameters enhances the precision of estimating mesoscale eddy impacts.

2) The spatial distribution of eddy-induced Chl-*a* anomalies indicates the potential for localized hotspots of productivity and nutrient supply within eddies (Figs. 3e–h in the manuscript).

3) Eddy impacts on DIC distributions highlight their role in transporting carbon-rich waters, notably affecting regional carbon budgets and oceanic carbon uptake.

4) Understanding the complexity of eddy-driven processes in the SO is vital for accurately simulating and predicting the biogeochemical dynamics of the SO.

For future scientific studies, we acknowledge two key limitations that warrant consideration.

1) First, our study focuses solely on the surface ocean, potentially overlooking subsurface Chl-*a* maxima (Cornec et al., 2021). Eddy-induced effects on phytoplankton growth are likely more prominent in the lower euphotic zone and could manifest less prominently at the surface (Mcgillicuddy et al., 2007; Siegel et al., 2011). The development of oceanic autonomous observation platforms,

especially biogeochemical Argo (BGC-Argo) floats, can help characterize the vertical structure of Chl-*a* and nutrients, improving our understanding of the physical-biological interactions.

2) Furthermore, we may underestimate the overall impact of SO eddies on physical and biogeochemical parameters due to unaccounted effects of smaller mesoscale features and submesoscale processes near eddy boundaries, such as submesoscale secondary circulations and small-scale turbulent mixing (Ning et al., 2021; Wang et al., 2021). Submesoscale processes support vertical velocities of up to 10–100 m day$^{-1}$, an order of magnitude larger than those induced by mesoscale eddies (Klein and Lapeyre, 2009). Expanding our investigation to smaller scales can enrich our understanding of eddy-driven processes.

2. Many expressions and conclusions need solid support.

(1) The most concern, the authors state that the eddies in the SO is mainly westward. Considering the large number of eastward eddies in ACC, statistics about westward/eastward and northward/southward eddy propagation should be clarified in the whole SO, rather than just focusing on eddies with lifetimes longer than 1 year (Figure 7).

Response: Thanks for your valuable feedback. In response to your suggestion, we have incorporated statistics that encompass a broader range of eddy lifetimes, including both short-lived and long-lived eddies, to represent eddy propagation directions accurately. Table 1, presented below, illustrates that regardless of the lifespan, both AEs and CEs propagate primarily westward and northward. By contrast, AEs and CEs living longer than 1 year propagate primarily northward and southward, respectively, corresponding with the intrinsic meridional propagation of eddies (Cushman-Roisin and Beckers, 2011). Frenger et al. (2015) reported that only partial eddies follow this intrinsic meridional propagation in the SO, owing to the strong overcompensation by the background meridional deflections of the mean current. Figure 7 in the manuscript shows that between 30°S and the ACC, the major propagation direction of eddies is westward, with AEs propagating north and CEs propagating south. However, most eddies in the ACC influence area propagate eastward, with AEs propagating south and CEs propagating north. These results are similar to those reported by Dawson et al. (2018).

According to the fact that more AEs and CEs propagate westward and northward, we added a paragraph in Section 4.1 to illustrate the eddy propagation directions. We also revised the descriptions in Section 5 regarding the influence of eddy trapping on SST, Chl-*a*, and DIC anomalies as follows:

1) We revised the sentences in lines 294–299 as follows:
   "Table S3 shows that the predominant propagation direction of eddies is westward and northward (Fig. 3). According to the southward decreasing SST, northward propagating eddies would trap cold water and result in negative SST anomalies. However, this process contradicts the positive SST anomalies within WAEs and WCEs, indicating the weak effect of eddy trapping on SST."

2) We revised the sentences in lines 324–328 as follows:

"Due to the climatological Chl-*a* increasing southward (Figs. 5b1–b3), eddies propagating northward tend to trap high Chl-*a* into northern areas with low Chl-*a*. Likewise, due to the climatological Chl-*a* increasing westward, eddies propagating westward tend to trap low Chl-*a* into western areas with high Chl-*a*. However, the effect of eddy trapping on Chl-*a* cannot explain the opposite Chl-*a* anomalies between AEs and CEs (Figs. 6b1, b2, f1, and f2). Consequently, it can be inferred that the role of eddy trapping in influencing Chl-*a* distributions is limited."

3) We revised the sentences in lines 350–352 as follows:
"Under the condition of southward increasing DIC (Figs. 5c1–c3), eddies propagating northward tend to trap high DIC. Thus, the effect of eddy trapping may contribute to the positive signals of DIC anomalies within eddies."

Table 1. Number of AEs and CEs moving westward/eastward and northward/southward, including the overall eddies and eddies with lifetimes longer than 1 year.

|  | Eastward | Westward | Northward | Southward |
|---|---|---|---|---|
| AEs | 7,924,626 | 9,261,954 | 9,266,102 | 7,920,478 |
| CEs | 8,387,806 | 9,300,955 | 9,824,357 | 7,864,404 |
| AEs (>1 year) | 44,184 | 323,242 | 294,536 | 72,890 |
| CEs (>1 year) | 97,955 | 294,167 | 140,362 | 251,760 |

(2) The authors use a number of averages to illustrate positive/negative signals induced by normal/abnormal eddies. Accounting for many noises (as shown in Figure 5), the standard error is suggested to be added to prove the reliability of the averages.

Response: Thank you for your insightful feedback. We calculated the averages and standard error of SST, Chl-*a*, DIC, and $p$CO$_2$ anomalies within WAEs, CAEs, CCEs, and WCEs, as shown in Table 2 and Fig. 1 below. The averages are consistent with the dominant signals of the anomaly patterns within eddies (Fig. 5 in the manuscript). The small standard error proves the reliability of the averages. These results strengthen the robustness of our conclusions and ensure a more accurate representation of the uncertainties associated with our results. We have added Table 2 and Fig. 1 (presented below) in supplementary.

Table 2. Averages and standard error (in parentheses) of SST, Chl-*a*, DIC, and $p$CO$_2$ anomalies within WAEs, CAEs, CCEs, and WCEs.

|  | WAEs | CAEs | CCEs | WCEs |
|---|---|---|---|---|
| SST | 0.03898 (0.00007) | -0.02697 (0.00011) | -0.04284 (0.00007) | 0.02828 (0.00010) |
| Chl-*a* | -0.00125 (0.00001) | -0.00101 (0.00002) | 0.00136 (0.00001) | 0.00054 (0.00002) |
| DIC | -0.18258 (0.00062) | 0.02228 (0.00120) | 0.21712 (0.00066) | -0.07598 (0.00106) |
| $p$CO$_2$ | -0.0323 (0.00061) | -0.01046 (0.00098) | 0.01498 (0.00060) | -0.01383 (0.00090) |

[Figure]

**Figure 1.** Averages (bars) and standard error (error bars) of SST, Chl-*a*, DIC, and $p$CO$_2$ anomalies within WAEs, CCEs, CAEs, and WCEs.

(3) For some results, the authors should cite references or give figures or significance to support the statement. For example, lines 236-237, "The amplitude and Chl-*a* anomalies are negatively correlated in subtropical waters north of the ACC and positively correlated along the ACC", needs evidence. Please revise the similar problems in the whole manuscript.

Response: Thank you for your suggestions. We have conducted a comprehensive review of the manuscript, ensuring that each result is adequately supported through appropriate citations or relevant figures or significance. We have undertaken the following revisions:

1) Lines 236–237: We calculated the correlation coefficient between Chl-*a* anomalies and eddy amplitude along latitudes. According to Fig. 2 below, we revised the sentences in lines 236–237 as follows:
   "As shown in Fig. S4, the correlation coefficients between amplitude and Chl-*a* anomalies have larger magnitudes in subtropical waters, with negative values in WAEs and CAEs and positive values in CCEs and WCEs. This result illustrates that in subtropical regions with higher amplitudes, such as BMC, ARC, and Tasman Sea, WAEs and CAEs induced stronger negative Chl-*a* anomalies, while CCEs and WCEs induced stronger positive Chl-*a* anomalies."

[Figure]

**Figure 2.** Correlation coefficients between Chl-*a* anomalies and eddy amplitudes along latitudes. The correlation coefficients range from -1 to 1, where -1 and 1 indicate perfect negative and positive linear correlations, respectively, and 0 signifies no linear correlation. Solid lines in different colors denote four kinds of eddies.

2) Lines 301–303: We added figures and revised the sentences as follows:
   "Specifically, WAEs rotating counterclockwise through the SST gradient would advect warmer water from the north to the southeast, leading to positive extremums slightly shifting westward and poleward relative to the cores (Figs. 6a1, e1). Conversely, CCEs rotating clockwise through the SST gradient would advect cooler water from the south to the northwest, leading to negative extremums slightly shifting westward and equatorward relative to the cores (Figs. 6a3, e3)."

3) Lines 333–334: We added a table and figures to support the statement:
   "As the major propagation direction of eddies is westward (Table S3), the composite Chl-*a* anomalies in AEs/CEs show dominant negative/positive signals due to eddy stirring (Figs. 6b1–b4 and f1–f4)."

4) Lines 380–382: We added figures to support the statement:
   "Likewise, the $p$CO$_2$ anomalies over eddies are determined by the DIC anomalies in winter, which is also associated with the higher magnitudes of DIC anomalies in winter compared to summer (Figs. 6c1–c4 and g1–g4)."

5) Lines 443–444: We added a table to support the statement:
   "Specifically, in the SWA dominated by "abnormal" eddies, the contributions of "abnormal" eddies to $p$CO$_2$ are opposite to normal eddies and are about twice as high as normal eddies (Table S5)."

3.  The eddy mechanisms for SST/Chl-*a*/DIC analyzes need more thinking.
The authors try to explain the SST, Chl-*a*, and DIC anomalies affected by normal and abnormal eddies via eddy pumping/Ekman pumping/eddy tripping/eddy stirring. I got lost in section 5 and many times feel hard to understand how the authors obtained the conclusion. Some times their figures don't support their conclusions, and sometimes the conclusions are not solid.

Response: Thanks for your valuable suggestions. To clarity of our analysis on the eddy mechanisms influencing on SST, Chl-*a*, and DIC anomalies, we added a schematic diagram showcasing the results and conclusions in Section 5, as shown below.

Furthermore, we have thoroughly assessed the alignment between our figures and the corresponding explanations to ensure that they consistently support our conclusions.

We also conducted more in-depth analyses to validate our findings and conclusions.

[Figure]

Figure 3. (a) Schematic illustrating the mechanisms of how eddies affect physical and biogeochemical parameters in the SO, including eddy stirring, eddy trapping, eddy pumping, and eddy-induced Ekman pumping. The patterns of SST anomalies induced by vertical pumping are opposite to the corresponding patterns shown in this schematic. The figure is inspired by Frenger et al. (2018), Fig. 1. Schematic diagram of the eddy mechanisms influencing (a) SST, (b) Chl-*a*, and (c) DIC anomalies.

**Minor questions:**

1.  The authors point out that amplitudes of abnormal eddies are smaller than normal eddies, so that the eddy pumping of abnormal eddies are weaker. While looking at table S1, is the difference of amplitude significant enough to induce opposite results of normal and abnormal eddies?

Response: Thanks for your feedback. Figures. 3a–d in the manuscript illustrate that in the regions with larger amplitude, the magnitudes of SST anomalies within normal eddies are higher and those within "abnormal" eddies are lower. We further compared the quantitative relationship between SST anomalies and amplitudes over normal and "abnormal" eddies, as shown in Fig. 4 below. The SST anomalies in WAEs and CAEs are positively correlated with amplitudes, while the SST anomalies in CCEs and WCEs are negatively correlated with amplitudes. These findings indicate a positive correlation between amplitude and eddy pumping, as eddy pumping within AEs/CEs corresponds to downwelling/upwelling, inducing positive/negative SST anomalies. The smaller the

amplitude of "abnormal" eddies, the weaker the eddy pumping, making them more susceptible to the influence of eddy-induced Ekman pumping. This results in opposite SST anomalies within "abnormal" eddies compared to normal eddies. Therefore, the negative/positive SST anomalies with high magnitudes in CAEs/WCEs in the low-amplitude regions characterize the final composite maps of SST anomalies within "abnormal" eddies (Figs. 5a2, a4, e2, and e4 in the manuscript).

[Figure]

**Figure 4.** The mean SST anomaly within (a) normal eddies and (b) "abnormal" eddies as a function of eddy amplitude in the SO. Dots denote the values averaged at the binned amplitude intervals of 2 cm. Solid lines denote the regression lines obtained from least squares fitting with S being the slope and R the correlation coefficient. Solid lines in different colors denote four kinds of eddies.

2. Some descriptions about data processing are very detailed in the manuscript, such as lines 171 to 175. It is suggested to be moved in supplementary.

Response: Thanks for your suggestion. The descriptions you mentioned have been moved in supplementary.

3. A table could be added to supplementary to better show information of spatial/temporal resolution and filtering methods.

Response: Thanks for your suggestion. We have added a table (as shown below) in supplementary to presents information about spatial and temporal resolutions and filtering methods employed.

Table 3. Spatial and temporal resolutions and filtering methods of SST, Chl-*a*, DIC, and $pCO_2$.

| | Temporal resolution | Spatial resolution | Temporal filter | Spatial filter |
|---|---|---|---|---|
| SST | daily | 0.25° × 0.25° | 7-90 days band-pass filter | |
| Chl-*a* | daily | 4 km × 4 km | | spatial high-pass filtering with 6° × 6° |
| DIC | monthly | 1° × 1° | subtracting the climatological averages | |
| *p*CO₂ | monthly | 1° × 1° | | |

4.  As for mechanisms for sea surface temperature and chlorophyll anomalies induced by abnormal eddies, the authors conclude that negative SST anomalies within CAEs are caused by eddy-induced Ekman pumping, while their negative Chl-*a* anomalies are due to eddy pumping. How can eddies modulate sea waters via two different vertical mechanisms? Although the authors explain that Chl-*a* is also affected by eddy stirring and show gradients of Chl-*a* background, the results shown in Figure 5 are not in agreement with their conclusions. Eddy stirring is supposed to induce dipole patterns, while Figure 5 shows monopoles (such as 5b2, 5f2). As a result, the authors should think more about the mechanisms.

Response: Thank you for your insightful questions and comments regarding our paper. Firstly, to address your concerns about the monopole Chl-*a* anomalies within winter eddies, as shown in Figs. 5b1–b4 in the manuscript, we calculated the seasonal average gradients of Chl-*a* in the SO. In winter, the north-south gradient of Chl-*a* is −0.01 (north is the positive direction), and the east-west gradient of Chl-*a* is −0.01 (east is the positive direction). However, in summer, the north-south gradient of Chl-*a* is −0.02, and the east-west gradient of Chl-*a* is −0.05. Compared to the summer Chl-*a* anomalies (Figs. 5f1–f4 in the manuscript), the smaller gradients of winter Chl-*a* weaken the impacts of eddy stirring, diminishing the dipole patterns of Chl-*a* anomalies within winter eddies (Figs. 5b1–b4 in the manuscript).

Although the dipole patterns of winter Chl-*a* anomalies are not obvious, we can still find the impacts of eddy stirring on Chl-*a*, that is, the meridional and zonal shifts of Chl-*a* anomalies extremums (Figs. 5b2–b4 and f1–f4 in the manuscript), which are proposed to be induced by the large-scale background Chl-*a* gradient and eddy stirring (Hausmann and Czaja, 2012; Villas Bôas et al., 2015). For meridional shifts, AEs rotating counterclockwise through the southward increasing Chl-*a* gradient would induce negative extremums slightly shifting poleward relative to the cores (Figs. 5b2, f1, and f2 in the manuscript). The reverse is true for CEs (Figs. 5b3, b4, f3, and f4 in the manuscript). For zonal shifts, AEs rotating counterclockwise through the westward increasing Chl-*a* gradient would induce negative extremums slightly shifting westward

relative to the cores (Figs. 5b2, f1, and f2 in the manuscript). The reverse is true for CEs (Figs. 5b3, b4, f3, and f4 in the manuscript).

Furthermore, Frenger et al. (2018) have demonstrated that eddy-induced Chl-*a* anomalies in the SO primarily stem from stirring, but they did not account for the impacts of vertical pumping induced by eddies. They proposed that lateral entrainment diminishes the dipole component of the Chl-*a* anomalies, resulting in predominantly monopole Chl-*a* anomaly patterns rather than dipole patterns.

In addition to the horizontal redistribution of Chl-*a* anomalies, the major limitation of marine Chl-*a* is the insufficient supplement of nutrients from depth into the euphotic zone (Mahadevan, 2016). The transport of nutrients enriched in deep seawater is mainly controlled by eddy pumping. By contrast, the variations of SST and DIC anomalies are prone to be influenced by heat and carbon exchange at the ocean-atmosphere interface (Gaube et al., 2015; Song et al., 2016), making them susceptible to eddy-induced Ekman pumping. Consequently, Chl-*a* anomalies in normal and "abnormal" eddies show similar patterns and signals, whereas SST and DIC anomalies in normal and "abnormal" eddies show opposite signals.

We have revised the eddy influencing mechanisms on Chl-*a* anomalies and deepened the discussion.

5. Line 339-341, it's a little hard for the readers to follow the authors' thinking. Please make the expression clearer, such as "the more evident Ekman pumping mechanism of abnormal eddies resisting eddy pumping and leads to lower Chl-*a* magnitude within abnormal eddies than normal eddies". Other similar problems in the manuscript could be also improved.

Response: Thanks for your suggestions. We have reviewed the entire manuscript to revise these problems. The specific modifications are shown below:

1) We revised the sentences in lines 339–341 as follows:
   "However, the more evident Ekman pumping mechanism of "abnormal" eddies resists eddy pumping and leads to lower Chl-*a* magnitude within "abnormal" eddies than normal eddies (Figs. 6b1–b4 and f1–f4)."

2) We revised the sentences in lines 423–425 as follows:
   "However, the magnitudes of Chl-*a* anomalies within "abnormal" eddies are lower than normal eddies, which is related to the more pronounced impact of "abnormal" eddies in counteracting eddy pumping through the mechanism of Ekman pumping."

6. Line 358-361. How did the authors get the conclusion? Please add references or show their own results. A similar problem also occurs in other expressions. Please read the manuscript seriously and improve them.

Response: Thanks for your advice. We have thoroughly reviewed the manuscript to address these problems. The specific modifications are shown below:

1) Lines 358–361: We added figures to show the results. The revised sentences are shown as follows:

"Moreover, the Ekman pumping caused by WAEs is stronger than that caused by CAEs (Figs. 8a1, a2), resulting in stronger positive DIC anomalies within WAEs than CAEs (Figs. 6c1, c2, g1, and g2). Similarly, the Ekman pumping caused by WCEs is stronger than that caused by CCEs (Figs. 8a3, a4), resulting in stronger negative DIC anomalies within WCEs than CCEs (Figs. 6c3, c4, g3, and g4)."

2) Lines 435–439: We added figures and revised the sentences as follows:
"In winter, the dominant DIC-driven effect leads to negative $pCO_2$ anomalies in WAEs and WCEs and positive anomalies in CAEs and CCEs (Figs. 6d1–d4). However, in summer, the $pCO_2$ anomalies are dominated by the combined effects of SST, Chl-$a$, and DIC (Figs. 6h1–h4). Notably, the $pCO_2$ anomalies within eddies are dominated by SST anomalies in the summer SWA, with smaller magnitudes of DIC anomalies (Fig. 9). In contrast, the $pCO_2$ anomalies within eddies are dominated by DIC anomalies in the ARC, with larger magnitudes of DIC anomalies (Fig. 10)."

7. Line 364, missing space.

Response: Revised.

8. Line 398-399, how did the authors evaluate the significance of different mechanisms? Are there any quantitative criteria?

Response: Thanks for your feedback. Regarding the evaluation of the significance of different mechanisms, our analysis focused on qualitative rather than quantitative assessment. By comparing the patterns and signals of SST, Chl-$a$, and DIC with the effects of eddy trapping, stirring, pumping, and eddy-induced Ekman pumping, we were able to identify the relative importance of each mechanism. While we acknowledge the value of quantitative criteria, our study aimed to provide insights into the dominant mechanisms through a qualitative analysis of the observed patterns and signals.

9. The writing in Figure 4a1 is suggested in Figure 4a1.

Response: Revised.

**Reference**

Cornec, M., Laxenaire, R., Speich, S., and Claustre, H.: Impact of Mesoscale Eddies on Deep Chlorophyll Maxima, Geophys. Res. Lett., 48, e2021GL093470, https://doi.org/https://doi.org/10.1029/2021GL093470, 2021.

Cushman-Roisin, B. and Beckers, J.-M.: Chapter 18 - Fronts, Jets and Vortices, in: International Geophysics, edited by: Cushman-Roisin, B., and Beckers, J.-M., Academic Press, 589-623, https://doi.org/https://doi.org/10.1016/B978-0-12-088759-0.00018-3, 2011.

Frenger, I., Münnich, M., and Gruber, N.: Imprint of Southern Ocean mesoscale eddies on chlorophyll, Biogeosciences, 15, 4781-4798, https://doi.org/10.5194/bg-15-4781-2018, 2018.

Frenger, I., Münnich, M., Gruber, N., and Knutti, R.: Southern Ocean eddy phenomenology, J. Geophys. Res.: Oceans, 120, 7413-7449, https://doi.org/10.1002/2015jc011047, 2015.

Gaube, P., Chelton, D. B., Samelson, R. M., Schlax, M. G., and O'Neill, L. W.: Satellite Observations of Mesoscale Eddy-Induced Ekman Pumping, J. Phys. Oceanogr., 45, 104-132, https://doi.org/10.1175/jpo-d-14-0032.1, 2015.

Hausmann, U. and Czaja, A.: The observed signature of mesoscale eddies in sea surface temperature and the associated heat transport, Deep-Sea Research Part I-Oceanographic Research Papers 70, 60-72, https://doi.org/10.1016/j.dsr.2012.08.005, 2012.

Klein, P. and Lapeyre, G.: The Oceanic Vertical Pump Induced by Mesoscale and Submesoscale Turbulence, Annu. Rev. Mar. Science, 1, 351-375, https://doi.org/10.1146/annurev.marine.010908.163704, 2009.

Mahadevan, A.: The Impact of Submesoscale Physics on Primary Productivity of Plankton, Annu. Rev. Mar. Science, 8, 161-184, https://doi.org/10.1146/annurev-marine-010814-015912, 2016.

McGillicuddy, D. J., Anderson, L. A., Bates, N. R., Bibby, T., Buesseler, K. O., Carlson, C. A., Davis, C. S., Ewart, C., Falkowski, P. G., Goldthwait, S. A., Hansell, D. A., Jenkins, W. J., Johnson, R., Kosnyrev, V. K., Ledwell, J. R., Li, Q. P., Siegel, D. A., and Steinberg, D. K.: Eddy/Wind Interactions Stimulate Extraordinary Mid-Ocean Plankton Blooms, Science, 316, 1021-1026, https://doi.org/doi:10.1126/science.1136256, 2007.

Ning, J., Chen, K., and Gaube, P.: Diverse Variability of Surface Chlorophyll During the Evolution of Gulf Stream Rings, Geophys. Res. Lett., 48, e2020GL091461, https://doi.org/https://doi.org/10.1029/2020GL091461, 2021.

Siegel, D. A., Peterson, P., McGillicuddy Jr., D. J., Maritorena, S., and Nelson, N. B.: Bio-optical footprints created by mesoscale eddies in the Sargasso Sea, Geophys. Res. Lett., 38, https://doi.org/10.1029/2011GL047660, 2011.

Song, H., Marshall, J., Munro, D. R., Dutkiewicz, S., Sweeney, C., McGillicuddy, D. J., and Hausmann, U.: Mesoscale modulation of air-sea $CO_2$ flux in Drake Passage, J. Geophys. Res.: Oceans, 121, 6635-6649, https://doi.org/10.1002/2016jc011714, 2016.

Villas Bôas, A. B., Sato, O. T., Chaigneau, A., and Castelão, G. P.: The signature of mesoscale eddies on the air‐sea turbulent heat fluxes in the South Atlantic Ocean, Geophys. Res. Lett., 42, 1856-1862, https://doi.org/10.1002/2015gl063105, 2015.

Wang, T., Chai, F., Xing, X., Ning, J., Jiang, W., and Riser, S. C.: Influence of multi-scale dynamics on the vertical nitrate distribution around the Kuroshio Extension: An investigation based on BGC-Argo and satellite data, Prog. Oceanogr., 193, 102543, https://doi.org/https://doi.org/10.1016/j.pocean.2021.102543, 2021.

---

## Author Response (AR3)

**Overall impression**

The paper investigates the physical and biogeochemical characteristics of mesoscale eddies at the surface of the Southern Ocean – a region of global importance for heat and carbon exchange and biogeochemical cycles, concurrently a region dominated by eddies. This study involves many novel aspects compared to previous studies and tackles relevant topics for the community. It distinguishes between warm-core anticyclonic eddies (AEs), cold-core AEs, cold-core cyclonic eddies (CEs), and warm-core CEs (termed here as 'normal' and 'abnormal' AEs and CEs).

The authors have done a great job in improving the manuscript. I find that the introduction is a lot clearer now, and generally, the text flows a lot better. I also like that the discussion is deeper now, and the figures that are kept in the main text are analyzed at sufficient depth. My other suggestions were addressed adequately too. I just have a few minor suggestions which should be addressed before publication.

Response: We would like to thank the reviewer for the positive comments and valuable suggestions to improve the manuscript. We hope the answers and information presented here would respond to what was demanded.

**General comment**

1.    As eddy-induced Ekman pumping should have a larger effect in high-wind regions, it would be great to check if there are more 'abnormal' eddies in high-wind areas than elsewhere. If not, then how do we explain it? Please add a few sentences about this in the discussion/conclusion section.

Response: We appreciate your insightful suggestion. We calculated the number of "abnormal" eddies occurrence in each $1° × 1°$ latitude-longitude bin over the analyzed period 1996–2015 in the SO. Then, we averaged the eddy number at the binned wind speed intervals of 1 m s$^{-1}$, as shown in Fig. 1 below. In low-wind regions, specifically with wind speeds less than 6 m s$^{-1}$, the occurrence of "abnormal" eddies is scarce. However, as wind speed progressively increases, we observe a corresponding increase in the number of "abnormal" eddies. Considering that eddy-induced Ekman pumping is expected to exert a more pronounced influence in high-wind regions, this result indicates the effect of eddy-induced Ekman pumping on the generation of "abnormal" eddies. We have incorporated this result into the discussion/conclusion section.

[Figure]

**Figure 1.** The mean number of "abnormal" eddies as a function of wind speed in the SO. The blue bars denote the number of "abnormal" eddies occurrence in each $1° \times 1°$ latitude-longitude bin over the analyzed period 1996–2015, which is averaged at the binned wind speed intervals of $1 \text{ m s}^{-1}$. The black error bars denote the standard error. The green line denotes the regression line obtained from least squares fitting with S being the slope and R the correlation coefficient.

**Specific comments**

1.  L11: I suggest putting 'normal' in inverted commas too (not just 'abnormal'). Check in the whole document.

Response: Revised.

2.  L45: "The variation of $p\text{CO}_2$ was found to be positively correlated with SST and DIC but negatively correlated with Chl-*a*… Therefore, the variation of $p\text{CO}_2$ within the eddies will be complex and necessitates discussion based on seasons and regions." This is not a necessary conclusion. Positive and negative correlations don't say anything about seasons and regions. Rephrase.

Response: Thanks for your valuable feedback. We revised the sentences in lines 45–48 as follows:

"The variation of $p\text{CO}_2$ was found to be positively correlated with SST and DIC but negatively correlated with Chl-*a* (Chen et al., 2007; Landschützer et al., 2015; Song et al., 2016; Fay et al., 2018; Jersild and Ito, 2020; Iida et al., 2021). The competing

seasonal cycles of SST, Chl-*a*, and DIC would induce the seasonal variability of $pCO_2$ and the seasonal variation of $pCO_2$ within the eddies varies in different regions (Chen et al., 2007; Frenger et al., 2013; Jiang et al., 2014; Munro et al., 2015; Song et al., 2016; Jones et al., 2017; Jersild and Ito, 2020). Therefore, the variation of $pCO_2$ within the eddies will be complex and necessitates discussion based on seasons and regions."

3.   L.49: 'recent studies'… 'Mcgillicuddy et al., 2007' 2007 is not that recent.

Response: Thanks for your suggestion. We replaced "recent studies" with "previous studies".

4.   L151: 'Besides, our method achieves great accuracy and much higher efficiency than the traditional method that first detects the eddies and then uses the SST signature to classify them into normal and "abnormal" eddies.' Add a reference to this statement.

Response: Revised.

5.   L158: 'in future work, we will combine multiple remote sensing data with Argo profiles to evaluate the accuracies of "abnormal" eddy identification method'. Discuss why you didn't do it in this study.

Response: Thanks for your feedback. There are several reasons why we did not incorporate this approach in the current study:

1) Data Availability and Access: The availability and access to high-quality remote sensing data and Argo profiles can vary based on geographic regions and time periods. Obtaining and processing such datasets can be a time-consuming and resource-intensive task.

2) Resource and Time Constraints: The uneven distribution of Argo floats in time and space can impact the precision and accuracy of using Argo for eddy identification. Most satellite data, on the other hand, can only monitor surface ocean information. Although it is possible to employ deep learning techniques to combine Argo and satellite data to infer three-dimensional ocean temperature and salinity with spatiotemporal continuity, this approach demands a substantial amount of time and resources for implementation and validation, which could extend the timeline of the study beyond its intended scope.

3) Suitability of the Current Approach: Our study aimed to establish the fundamental relationships and mechanisms between eddies and the oceanographic parameters of interest. The existing data and methods are sufficient to address our primary research questions.

6.   L209: 'from 1996 to 2015, an average of 1991 eddies were identified daily' This

number is a lot larger than the ~1 million eddies that Frenger et al. 2015 detected in the S.O between 1997 and 2010 (~ 200 daily eddies on average). Are we sure they are all robust features?

Response: Thanks for your feedback. The discrepancy in the number of identified eddies between our study and Frenger et al. (2015) is primarily due to the differences in the data resolution and detection criteria.

1) Data resolution: The eddies detected by Frenger et al. (2015) are weekly, constrained by the weekly SSHA data they used. By contrast, the eddy dataset we used is daily, which is detected by daily SSHA and SSTA data. Therefore, the eddies identified by Frenger et al. (2015) are roughly seven times fewer than ours within the same period.

2) Lifespan limitation: Frenger et al. (2015) detected eddies with a minimum lifespan of four weeks. By contrast, we had no limitation on eddies' lifespan, as we believe that short-lived eddies are also significant.

In addition, we discarded eddies with amplitudes < 2 cm and radii < 35 km due to the limitations of the resolution capability of the SSHA data (Ducet et al., 2000), which helps to increase the robustness of eddy features. Besides, the spatial distributions of eddies are consistent with the observations reported by Frenger et al. (2015) and Dawson et al. (2018), substantiating the reliability of the eddy dataset we utilized.

7. L498: It's always good to end the paper with a strong statement. Maybe swap the last two paragraphs around.

Response: Thanks for your suggestion. We deleted the last two paragraphs.

8. Fig.1-5 captions: Name which of the fronts they are and their reference.

Response: Thanks for your suggestion. We revised the related sentences in Figs. 1–5 and S3 as follows:

"Black solid lines show the mean northern (SAF) and southern (PF) positions of the ACC major fronts (Sallée et al., 2008)"

**Technical corrections**

1. L275: Replace 'extremums' with 'extrema' (check whole document)

Response: Revised

**Reference**

Chen, F., Cai, W.-J., Benitez-Nelson, C., and Wang, Y.: Sea surface $p\mathrm{CO_2}$-SST relationships across a cold-core cyclonic eddy: Implications for understanding regional variability and air-sea gas exchange, Geophys. Res. Lett., 34, https://doi.org/10.1029/2006gl028058, 2007.

Dawson, H. R. S., Strutton, P. G., and Gaube, P.: The Unusual Surface Chlorophyll Signatures of Southern Ocean Eddies, J. Geophys. Res.: Oceans, 123, 6053-6069, https://doi.org/10.1029/2017JC013628, 2018.

Ducet, N., Le Traon, P. Y., and Reverdin, G.: Global high-resolution mapping of ocean circulation from TOPEX/Poseidon and ERS-1 and -2, J. Geophys. Res.: Oceans, 105, 19477-19498, https://doi.org/10.1029/2000JC900063, 2000.

Fay, A. R., Lovenduski, N. S., McKinley, G. A., Munro, D. R., Sweeney, C., Gray, A. R., Landschützer, P., Stephens, B. B., Takahashi, T., and Williams, N.: Utilizing the Drake Passage Time-series to understand variability and change in subpolar Southern Ocean $p\mathrm{CO_2}$, Biogeosciences, 15, 3841-3855, https://doi.org/10.5194/bg-15-3841-2018, 2018.

Frenger, I., Gruber, N., Knutti, R., and Münnich, M.: Imprint of Southern Ocean eddies on winds, clouds and rainfall, Nat. Geosci., 6, 608-612, https://doi.org/10.1038/ngeo1863, 2013.

Frenger, I., Münnich, M., Gruber, N., and Knutti, R.: Southern Ocean eddy phenomenology, J. Geophys. Res.: Oceans, 120, 7413-7449, https://doi.org/10.1002/2015jc011047, 2015.

Iida, Y., Takatani, Y., Kojima, A., and Ishii, M.: Global trends of ocean $\mathrm{CO_2}$ sink and ocean acidification: an observation-based reconstruction of surface ocean inorganic carbon variables, J. Oceanogr., 77, 323-358, https://doi.org/10.1007/s10872-020-00571-5, 2021.

Jersild, A. and Ito, T.: Physical and Biological Controls of the Drake Passage $p\mathrm{CO_2}$ Variability, Global Biogeochem. Cycles, 34, https://doi.org/10.1029/2020gb006644, 2020.

Jiang, C., Gille, S. T., Sprintall, J., and Sweeney, C.: Drake Passage Oceanic $p\mathrm{CO_2}$: Evaluating CMIP5 Coupled Carbon–Climate Models Using in situ Observations, J. Clim., 27, 76-100, https://doi.org/10.1175/jcli-d-12-00571.1, 2014.

Jones, E. M., Hoppema, M., Strass, V., Hauck, J., Salt, L., Ossebaar, S., Klaas, C., van Heuven, S. M. A. C., Wolf-Gladrow, D., Stöven, T., and de Baar, H. J. W.: Mesoscale features create hotspots of carbon uptake in the Antarctic Circumpolar Current, Deep Sea Res. Part II, 138, 39-51, https://doi.org/10.1016/j.dsr2.2015.10.006, 2017.

Landschützer, P., Gruber, N., Haumann, F. A., Rödenbeck, C., Bakker, D. C. E., Heuven, S. v., Hoppema, M., Metzl, N., Sweeney, C., Takahashi, T., Tilbrook, B., and Wanninkhof, R.: The reinvigoration of the Southern Ocean carbon sink, Science, 349, 1221-1224, https://doi.org/doi:10.1126/science.aab2620, 2015.

Munro, D. R., Lovenduski, N. S., Stephens, B. B., Newberger, T., Arrigo, K. R., Takahashi, T., Quay, P. D., Sprintall, J., Freeman, N. M., and Sweeney, C.: Estimates

of net community production in the Southern Ocean determined from time series observations (2002–2011) of nutrients, dissolved inorganic carbon, and surface ocean $p$CO$_2$ in Drake Passage, Deep Sea Res. Part II, 114, 49-63, https://doi.org/10.1016/j.dsr2.2014.12.014, 2015.

Sallée, J. B., Speer, K., and Morrow, R.: Response of the Antarctic Circumpolar Current to Atmospheric Variability, J. Clim., 21, 3020-3039, https://doi.org/10.1175/2007JCLI1702.1, 2008.

Song, H., Marshall, J., Munro, D. R., Dutkiewicz, S., Sweeney, C., McGillicuddy, D. J., and Hausmann, U.: Mesoscale modulation of air-sea CO$_2$ flux in Drake Passage, J. Geophys. Res.: Oceans, 121, 6635-6649, https://doi.org/10.1002/2016jc011714, 2016.